# Generalizing Nonlinear ICA Beyond Structural Sparsity

**Yujia Zheng**[1], **Kun Zhang**[1,2]
[1] Carnegie Mellon University
[2] Mohamed bin Zayed University of Artificial Intelligence
{yujiazh, kunz1}@cmu.edu

## Abstract

Nonlinear independent component analysis (ICA) aims to uncover the true latent sources from their observable nonlinear mixtures. Despite its significance, the identifiability of nonlinear ICA is known to be impossible without additional assumptions. Recent advances have proposed conditions on the connective structure from sources to observed variables, known as *Structural Sparsity*, to achieve identifiability in an unsupervised manner. However, the sparsity constraint may not hold universally for all sources in practice. Furthermore, the assumptions of bijectivity of the mixing process and independence among all sources, which arise from the setting of ICA, may also be violated in many real-world scenarios. To address these limitations and generalize nonlinear ICA, we propose a set of new identifiability results in the general settings of undercompleteness, partial sparsity and source dependence, and flexible grouping structures. Specifically, we prove identifiability when there are more observed variables than sources (undercomplete), and when certain sparsity and/or source independence assumptions are not met for some changing sources. Moreover, we show that even in cases with flexible grouping structures (e.g., part of the sources can be divided into irreducible independent groups with various sizes), appropriate identifiability results can also be established. Theoretical claims are supported empirically on both synthetic and real-world datasets.

## 1 Introduction

The unveiling of the true generating process of observations is fundamental to scientific discovery. Nonlinear independent component analysis (ICA) provides a statistical framework that represents a set of observed variables $\mathbf{x}$ as a nonlinear mixture of independent latent sources $\mathbf{s}$, i.e., $\mathbf{x} = \mathbf{f}(\mathbf{s})$. Unlike linear ICA (Comon, 1994), the mixing function $\mathbf{f}$ can be an unknown nonlinear function, thus generalizing the theory to more real-world tasks. However, the identifiability of nonlinear ICA has been a long-standing problem for decades. The main obstacle is that, without additional assumptions, there exist infinite spurious solutions returning independent variables that are mixtures of the true sources (Hyvärinen and Pajunen, 1999). In the context of machine learning, this makes the theoretical analysis of unsupervised learning of disentangled representations difficult (Locatello et al., 2019).

To overcome this challenge, recent work has introduced the auxiliary variable $\mathbf{u}$, and assumed that all sources are conditionally independent given $\mathbf{u}$. Most of these methods require auxiliary variables to be observable, such as class labels and domain indices (Hyvärinen and Morioka, 2016; Hyvärinen et al., 2019; Khemakhem et al., 2020a; Sorrenson et al., 2020; Lachapelle et al., 2022; Lachapelle and Lacoste-Julien, 2022), with the exceptions being those for time series (Hyvärinen and Morioka, 2017; Hälvä et al., 2021; Yao et al., 2021, 2022). While the use of the auxiliary variable $\mathbf{u}$ allows for the identifiability of nonlinear ICA with mild restrictions on the mixing process, it also necessitates a large number of distinct values of $\mathbf{u}$, which can be difficult to obtain in tasks with insufficient side informa-

tion. Moreover, since these results assume that all sources are dependent on $\mathbf{u}$, they cannot accommodate a subset of sources with invariant distributions (e.g., content may not change with different styles).

Another possible direction is to impose appropriate conditions on the mixing process, but limited results are available in the literature. For example, it has been shown that conformal maps are identifiable up to specific indeterminacies (Hyvärinen and Pajunen, 1999; Buchholz et al., 2022). Moreover, Taleb and Jutten (1999) identify the latent sources when the mixing process is a component-wise nonlinear function added to a linear mixture. These methods do not rely on conditional independence given the auxiliary variable and thus achieve the identifiability in a fully unsupervised setting. At the same time, the requirement of above-mentioned classes of the mixing function, such as conformal maps and post-nonlinear models, restricts the applicability of the results in another way. For instance, according to Liouville's theorem (Monge, 1850), conformal maps in Euclidean spaces of dimensions higher than two are Möbius transformations, which appear to be overly restrictive for most data-generating processes. As an alternative, Zheng et al. (2022) prove that, under the assumption of *Structural Sparsity*, the true sources can be identified up to trivial indeterminacies. Since the proposed condition is on the connective structure from sources to observed variables, i.e., the support of the Jacobian matrix of the mixing function, it does not require the mixing function to be of any specific algebraic form. Thus, *Structural Sparsity* may serve as one of the first general principles for the identifiability of nonlinear ICA in a fully unsupervised setting.

While being a potential solution to the identifiability of nonlinear ICA without side information, the assumption of *Structural Sparsity* has its limitations from a pragmatic viewpoint. The most obvious one arises from the fact that it may fail in a number of situations where the generating processes are heavily entangled. Although the principle of simplicity may be a general rule in nature, it is intuitively possible that *Structural Sparsity* does not apply to at least a subset of sources, such as one or a few speakers in a crowded room. Unfortunately, Zheng et al. (2022) require *Structural Sparsity* to hold for all sources in order to provide any identifiability guarantee. Therefore, it would be desirable in practice to provide weaker notions of identifiability, such as the ability to identify a subset of sources to a trivial degree of uncertainty, in cases of partial sparsity.

In addition to partial sparsity, identifiability with *Structural Sparsity* also fails with the undercompleteness (more observed variables than sources) and/or partial source dependence (potential dependence among some hidden sources). These limitations are not unique to the sparsity assumption, but rather a result of the traditional setting of ICA, where the numbers of the sources and observed variables must be equal and dependencies among sources are not allowed. However, both situations are quite common in practice. One may easily have millions of pixels (observed variables) but only dozens of hidden concepts (sources) in a picture, constituting an undercomplete case that cannot be handled by previous results. Meanwhile, dependencies among some variables are also prevalent in tasks such as computational biology (Cardoso, 1998; Theis, 2006). The alternative assumption of conditional independence given auxiliary variables may still be overly restrictive if applied universally to all sources. For the identifiability of nonlinear ICA to truly benefit scientific discovery in a wider range of scenarios, these methodological limitations should be properly addressed.

Aiming to generalize nonlinear ICA with *Structural Sparsity*, we first present a set of new identifiability results to address these fundamental challenges of undercompleteness, partial sparsity, and source dependence. We show that, under the assumption of *Structural Sparsity* and without auxiliary variables, latent sources can be identified from their nonlinear mixtures up to a component-wise invertible transformation and a permutation, even when there are more observed variables than sources (Thm. 3.1). Moreover, if the assumption of sparsity and/or source independence does not hold for some changing sources, we provide partial identifiability results, showing that the remaining sources can still be identified up to the same trivial indeterminacy (Thm. 4.1, Thm. 4.2). Furthermore, in the cases with flexible grouping structures (e.g., part of the sources can be grouped into irreducible independent subgroupings with various sizes, such as mixtures of signals with various dimensions), certain types of identifiability are also guaranteed with auxiliary variables (Thm. 4.3, Thm. 4.4). Therefore, we establish, to the best of our knowledge, one of the first general frameworks for uncovering latent variables with appropriate identifiability guarantees in a principled manner. The theoretical claims are validated empirically through our experiments and many previous works involving disentanglement.

## 2 Preliminaries

The data-generating process of nonlinear ICA is as follows:

$$p_{\mathbf{s}}(\mathbf{s}) = \prod_{i=1}^{n} p_{\mathbf{s}_i}(\mathbf{s}_i), \tag{1}$$

$$\mathbf{x} = \mathbf{f}(\mathbf{s}), \tag{2}$$

where $\mathbf{s} = (\mathbf{s}_1, \ldots, \mathbf{s}_n) \in \mathcal{S} \subseteq \mathbb{R}^n$ is a latent vector representing the independent sources, and $\mathbf{x} = (\mathbf{x}_1, \ldots, \mathbf{x}_m) \in \mathcal{X} \subseteq \mathbb{R}^m$ denotes the observed random vector. The mixing function $\mathbf{f}$ is assumed to be smooth in the sense that its second-order derivatives exist. The primary objective of ICA is to establish *identifiable* models, i.e., the sources $\mathbf{s}$ are identifiable (recoverable) up to certain indeterminacies by learning an estimated mixing function $\hat{\mathbf{f}} : \hat{\mathcal{S}} \to \mathcal{X}$ with assumptions identical to the generating process (Comon, 1994). Different from most ICA results where $m = n$ and $\mathbf{f} : \mathcal{S} \to \mathcal{X}$ must be linear, we allow $m > n$ (i.e., undercompleteness) and $\mathbf{f}$ to be a general nonlinear function, therefore extending the previous setting. Thus, we relax the previous assumption on the invertibility of $\mathbf{f}$, only necessitating it to be injective and its Jacobian to be of full column rank. Furthermore, we denote $p_{s_i}$ as the marginal probability density function (PDF) of the $i$-th source $s_i$ and $p_{\mathbf{s}}$ as the joint PDF of the random vector $\mathbf{s}$. Moreover, we introduce some additional technical notations as follows:

**Definition 2.1.** Given a subset $\mathcal{A} \subseteq \{1, \ldots, n\}$, the subspace $\mathbb{R}_{\mathcal{A}}^n$ is defined as

$$\mathbb{R}_{\mathcal{A}}^n := \{z \in \mathbb{R}^n \mid i \notin \mathcal{A} \implies z_i = 0\},$$

where $z_i$ is the $i$-th element of the vector $z$.

That is, $\mathbb{R}_{\mathcal{A}}^n$ denotes the subspace of $\mathbb{R}^n$ specified by an index set $\mathcal{A}$. Furthermore, we define the support of a matrix as follows:

**Definition 2.2.** The support of a matrix $\mathbf{M} \in \mathbb{R}^{m \times n}$ is defined as

$$\mathrm{supp}(\mathbf{M}) := \{(i, j) \mid \mathbf{M}_{i,j} \neq 0\}.$$

With a slight abuse of notation, we reuse $\mathrm{supp}(\cdot)$ to denote the support of a matrix-valued function:

**Definition 2.3.** The support of a function $\mathbf{M} : \Theta \to \mathbb{R}^{m \times n}$ is defined as

$$\mathrm{supp}(\mathbf{M}(\Theta)) := \{(i, j) \mid \exists \theta \in \Theta, \mathbf{M}(\theta)_{i,j} \neq 0\}.$$

For brevity, we denote $\mathcal{F}$ and $\hat{\mathcal{F}}$ as the support of the Jacobian $\mathbf{J_f}(\mathbf{s})$ and $\mathbf{J}_{\hat{\mathbf{f}}}(\hat{\mathbf{s}})$, respectively. Additionally, $\mathcal{T}$ refers to a set of matrices with the same support of $\mathbf{T}(\mathbf{s})$ in $\mathbf{J}_{\hat{\mathbf{f}}}(\hat{\mathbf{s}}) = \mathbf{J_f}(\mathbf{s})\mathbf{T}(\mathbf{s})$, where $\mathbf{T}(\mathbf{s})$ is a matrix-valued function. Throughout this work, for any matrix $\mathbf{M}$, we use $\mathbf{M}_{i,:}$ to denote its $i$-th row, and $\mathbf{M}_{:,j}$ to denote its $j$-th column. For any set of indices $\mathcal{B} \subset \{1, \ldots, m\} \times \{1, \ldots, n\}$, analogously, we have $\mathcal{B}_{i,:} := \{j \mid (i, j) \in \mathcal{B}\}$ and $\mathcal{B}_{:,j} := \{i \mid (i, j) \in \mathcal{B}\}$.

## 3 Identifiability with undercompleteness

We first present the result on removing one of the major assumptions in ICA, i.e., the number of observed variables $m$ must be equal to that of hidden sources $n$. We prove that, in the undercomplete case ($m > n$), sources can be identified up to a trivial indeterminacy under *Structural Sparsity*.

**Theorem 3.1.** *Let the observed data be a large enough sample generated by an undercomplete nonlinear ICA model as defined in Eqs. (1) and (2). Suppose the following assumptions hold:*

    *i. For each $i \in \{1, \ldots, n\}$, there exist $\{\mathbf{s}^{(\ell)}\}_{\ell=1}^{|\mathcal{F}_{i,:}|}$ and a matrix $\mathrm{T} \in \mathcal{T}$ s.t. $\mathrm{span}\{\mathbf{J_f}(\mathbf{s}^{(\ell)})_{i,:}\}_{\ell=1}^{|\mathcal{F}_{i,:}|} = \mathbb{R}_{\mathcal{F}_{i,:}}^n$ and $\left[\mathbf{J_f}(\mathbf{s}^{(\ell)})\mathrm{T}\right]_{i,:} \in \mathbb{R}_{\hat{\mathcal{F}}_{i,:}}^n$.*

    *ii. (Structural Sparsity) For each $k \in \{1, \ldots, n\}$, there exists $\mathcal{C}_k$ s.t. $\bigcap_{i \in \mathcal{C}_k} \mathcal{F}_{i,:} = \{k\}$.*

*Then $\mathbf{s}$ is identifiable up to an element-wise invertible transformation and a permutation.*

The proof is included in Appx. A.1, of which part of the conditions and techniques are based on (Zheng et al., 2022). It is noteworthy that, same as previous work, we also need to add a sparsity

regularization on the learned Jacobian during the estimation so that $|\hat{\mathcal{F}}| \leq |\mathcal{F}|$, which is required for all sparsity-based identifications throughout the paper and we only emphasize here for brevity.

Assumption i avoids some pathological conditions (e.g., samples are from very limited sub-populations that only span a degenerate subspace) and is typically satisfied asymptotically. The first part implies that there are at least $|\mathcal{F}_{i,:}|$ observed samples spanning the support space, which is almost always satisfied asymptotically. The second part is also relatively mild. Note that $\mathcal{T}$ refers to a set of matrices with the same support of $\mathbf{T}(\mathbf{s})$ in $\mathbf{J}_{\hat{\mathbf{f}}}(\hat{\mathbf{s}}) = \mathbf{J}_{\mathbf{f}}(\mathbf{s})\mathbf{T}(\mathbf{s})$ and $\mathbf{J}_{\hat{\mathbf{f}}}(\hat{\mathbf{s}})_{i,:} \in \mathbb{R}^n_{\hat{\mathcal{F}}_{i,:}}$. Since we only necessitate the existence of one matrix $\mathbf{T} \in \mathcal{T}$ in the entire space, even in rare cases where these two matrices do not share the same non-zero coordinates due to non-generic canceling between specific values of elements, there is almost always an existence of a matrix $\mathbf{T} \in \mathcal{T}$ fulfilling the assumption.

Assumption ii, i.e., *Structural Sparsity*, originates from (Zheng et al., 2022). Intriguingly, compared to the original bijective setting considered by (Zheng et al., 2022), this assumption is much more likely to be satisfied in the undercomplete case. The key reason is that it only necessitates the existence of a subset of observed variables whose intersection uniquely identifies the target source variable. For instance, regarding $\mathbf{s}_1$ in Fig. 1, there exist $\mathbf{x}_1$ and $\mathbf{x}_4$ s.t. the intersection of their parents is only $\mathbf{s}_1$. In principle, the size of this set can be quite small (e.g., one or two). Hence, it is very likely to be satisfied when there is a sufficient number of observed

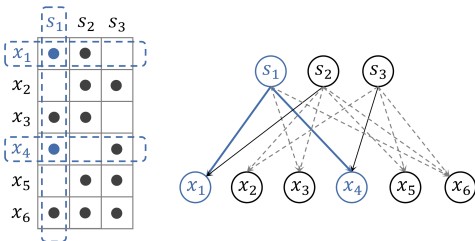

Figure 1: The structural sparsity assumption in the undercomplete case, where the matrix represents $\mathrm{supp}(\mathbf{J}_{\mathbf{f}}(\mathbf{s}))$.

variables (e.g., millions of pixels for images), which has also been verified empirically in our experiments (e.g., Fig. 4 in Sec. 5). Additionally, in some tasks, we might even construct or select observations in a data-centric manner to satisfy this assumption. Without the previous constraint of bijectivity, structural sparsity can truly be applied in a much broader range of practical scenarios.

By proving the identifiability in the undercomplete case, we remove the previous assumption of bijectivity on the mixing function $\mathbf{f}$ and thus generalizing the theory to more application scenarios. It is worth noting that, while some work has provided results without assuming bijectivity (Khemakhem et al., 2020a), they rely on extra information from many distinct values of auxiliary variables. Differently, we do not need any auxiliary variables and follow a fully unsupervised setting; Zheng et al. (2022); Kivva et al. (2022) explore the undercompleteness without any auxiliary variable. However, Zheng et al. (2022) only remove the rotational indeterminacy in the nonlinear case and Kivva et al. (2022) assume Gaussian mixture priors, while we provide the full identifiability result without distributional assumptions. At the same time, as elaborated above, the assumption of *Structural Sparsity* has been significantly weakened in the undercomplete case considered by our theorem. Moreover, identifiability with undercompleteness is also essential if assumptions are partially violated w.r.t. a subset of sources, of which the intuition is verified by theoretical results introduced in the following sections.

## 4   Identifiability with partial sparsity and source dependence

Under the condition of *Structural Sparsity*, we show the identifiability of undercomplete ICA with general nonlinear functions (Thm. 3.1). While this removes the restriction of bijectivity between sources and observed variables, it remains uncertain as to whether *Structural Sparsity* holds for all sources in a universal way. At the same time, even in the scenarios that *Structural Sparsity* may not be universally satisfied for all sources, it is still valuable to consider its potential to hold true for a subset of sources. This type of partial sparsity may often be the case in practical scenarios, as illustrated by our experiments (e.g., Fig. 5 in Sec. 5). However, the corresponding partial identifiability, i.e., the theoretical guarantee for the identification of the subset of sources satisfying *Structural Sparsity*, is not achieved by (Zheng et al., 2022). In fact, as long as one or a few sources do not meet the assumption of *Structural Sparsity*, the previous work is unable to provide any identifiability guarantees.

Furthermore, in addition to the universal sparsity, the statistical independence between sources is another fundamental assumption. This assumption arises from the original setting of ICA and has been adopted in most related works. However, in many real-world scenarios, requiring *all* sources to be mutually independent might be impractical, and there are likely to be a subset of sources that

are dependent in some way. For example, the frequency and duration of smoking, as well as the type of tobacco products used, are all interrelated factors that contribute to the development of lung cancer. Without any identifiability guarantees in the case where there exist any dependent sources, it is quite restrictive for nonlinear ICA, and even its undercomplete extension, to successfully deal with real problems in practice. Therefore, similar to the partial sparsity case, it is highly desirable that alternative theoretical results, i.e., identifiability for independent sources, can be guaranteed even with the existence of dependent sources.

To deal with these remaining challenges in the considered undercomplete case, we further relax other assumptions with additional information. To start with, we provide an identifiability result when *Structural Sparsity* holds true for only a subset of sources, thus alleviating the obstacle of partial sparsity. Moreover, we relax the mutual independence assumption and allow for some changing sources to be dependent on each other. To this end, we partition the sources into two parts $\mathbf{s} = [\mathbf{s}_I, \mathbf{s}_D]$, where variables in $\mathbf{s}_I$ are mutually independent, but those in $\mathbf{s}_D$ do not need to be. Let $\mathbf{s}_I$ and $\mathbf{s}_D$ correspond to variables in $\mathbf{s}$ with indices $\{1, \ldots, n_I\}$ and $\{n_{I+1}, \ldots, n\}$, respectively. That is, $\mathbf{s}_I = (s_1, \ldots, s_{n_I}) \in \mathcal{S}_I \subseteq \mathbb{R}^{n_I}$ and $\mathbf{s}_D = (s_{n_I+1}, \ldots, s_n) \in \mathcal{S}_D \subseteq \mathbb{R}^{n_D}$. We denote the $i$-th scalar element in a vector, say $\mathbf{s}$, as $s_i$. For sources in $\mathbf{s}_D$, they do not need to be mutually independent as long as they are dependent on a variable $\mathbf{u}$, i.e.,

$$p_{\mathbf{s}|\mathbf{u}}(\mathbf{s}|\mathbf{u}) = p_{\mathbf{s}_D|\mathbf{u}}(\mathbf{s}_D|\mathbf{u}) \prod_{i=1}^{n_i} p_{s_i}(s_i). \tag{3}$$

It is noteworthy that we allow arbitrary relations between sources in $\mathbf{s}_D$. These sources might be grouped into several subspaces or actually be mutually independent, but we do not need to obtain this information as prior knowledge. This is essential since the exact dependence structures, or even the number of dependent variables, are usually unknown in practice. By keeping this type of uncertainty for both partial sparsity and/or partial source dependence, one can be more confident in applying the theoretical advancements of nonlinear ICA in various tasks. We prove the identifiability for this arguably more flexible scenario as the following theorems:

**Theorem 4.1.** *Let the observed data be a large enough sample generated by an undercomplete nonlinear ICA model defined in Eqs. (2) and (3). Suppose the following assumptions hold:*

    *i. There exist $n_D + 1$ distinct values of $\mathbf{u}$, i.e., $\mathbf{u}_j$ with $j \in \{0, 1, \ldots, n_D\}$, s.t. the $n_D$ vectors $\mathbf{w}(\mathbf{s}_D, \mathbf{u}, i)$ with $i \in \{n_I + 1, \ldots, n\}$ are linearly independent, where vector $\mathbf{w}(\mathbf{s}_D, \mathbf{u}, i)$ is defined as follows:*

$$\mathbf{w}(\mathbf{s}_D, \mathbf{u}, i) = \left( \frac{\partial \left( \log p(\mathbf{s}_D|\mathbf{u}_1) - \log(p(\mathbf{s}_D|\mathbf{u}_0)) \right)}{\partial s_i}, \ldots, \frac{\partial \left( \log p(\mathbf{s}_D|\mathbf{u}_{n_D}) - \log(p(\mathbf{s}_D|\mathbf{u}_0)) \right)}{\partial s_i} \right).$$

    *ii. There exist $\mathbf{u}_1, \mathbf{u}_2 \in \mathbf{u}$, s.t., for any set $A_\mathbf{s} \subseteq \mathcal{S}$ with non-zero probability measure and cannot be expressed as $B_{\mathbf{s}_I} \times \mathbf{s}_D$ for any $B_{\mathbf{s}_I} \subset \mathcal{S}_I$, we have*

$$\int_{\mathbf{s} \in A_\mathbf{s}} p_{\mathbf{s}|\mathbf{u}}(\mathbf{s} \mid \mathbf{u}_1) \, d\mathbf{s} \neq \int_{\mathbf{s} \in A_\mathbf{s}} p_{\mathbf{s}|\mathbf{u}}(\mathbf{s} \mid \mathbf{u}_2) \, d\mathbf{s}.$$

*Then $\mathbf{s}_D$ is identifiable up to an subspace-wise invertible transformation.*

Thm. 4.1 ensures the subspace-wise identifiability of $\mathbf{s}_D$, i.e., the estimated subspace $\hat{\mathbf{s}}_D$ contains all and only information from $\mathbf{s}_D$. This implies that we can disentangle and extract the invariant part of the latents, beneficial for tasks like domain adaptation where recovering each individual source might not be necessary as long as the subspace that is invariant across domains can be disentangled. Furthermore, we also prove the component-wise identifiability as follows:

**Theorem 4.2.** *In addition to assumptions in Thm. 4.1, suppose the following assumptions hold:*

    *i. For each $i \in \{1, \ldots, n_I\}$, there exist $\{\mathbf{s}^{(\ell)}\}_{\ell=1}^{|\mathcal{F}_{i,:n_I}|}$ and a matrix $\mathrm{T} \in \mathcal{T}$ s.t. $\mathrm{span}\{\mathbf{J_f}(\mathbf{s}^{(\ell)})_{i,:n_I}\}_{\ell=1}^{|\mathcal{F}_{i,:n_I}|} = \mathbb{R}_{\mathcal{F}_{i,:n_I}}^{n_I}$ and $[\mathbf{J_f}(\mathbf{s}^{(\ell)})\mathrm{T}]_{i,:n_I} \in \mathbb{R}_{\hat{\mathcal{F}}_{i,:n_I}}^{n_I}$.*

    *ii. (Structural Sparsity) For all $k \in \{1, \ldots, n_I\}$, there exists $\mathcal{C}_k$ s.t. $\bigcap_{i \in \mathcal{C}_k} \mathcal{F}_{i,:n_I} = \{k\}$.*

*Then $\mathbf{s}_I$ is identifiable up to an element-wise invertible transformation and a permutation.*

The proofs are presented in Appx. A.2 and Appx. A.3. We tackle the challenge of partial sparsity and dependence by necessitating *Structural Sparsity* and independence only on a subset of sources in $\mathbf{s}_I$ and prove that these sources can be identified up to trivial indeterminacies. For the remaining sources in $\mathbf{s}_D$, they only need to be dependent on an auxiliary variable $\mathbf{u}$ without necessitating conditional independence among sources or distributional assumption. This extends previous models that assume all sources to be conditionally independent given $\mathbf{u}$ (Hyvärinen et al., 2019; Lachapelle et al., 2022) or require the conditional distribution of the sources to be of a specific form (Khemakhem et al., 2020b).

The assumption on $p(\mathbf{s}_D|\mathbf{u})$ in Thm. 4.1 indicates that the auxiliary variable $\mathbf{u}$ should have a sufficiently diverse impact on sources without independence assumption (i.e., $\mathbf{s}_D$). It follows a similar spirit to the standard assumption of variability (Hyvärinen et al., 2019) but we further relax it. Specifically, we only need $n_D + 1$ values of $\mathbf{u}$ for the identifiability of sources in $\mathbf{s}_I$. This is intuitively reasonable since the *fewer changes* (smaller $n_D$) a system has, the *easier* (fewer required values, i.e., $n_D + 1$) that *a larger part of it* (larger $n_I$, i.e., $n - n_D$) is identifiable. In contrast, most previous works require all sources to be dependent on an auxiliary variable $\mathbf{u}$ with $2n+1$ distinct values of $\mathbf{u}$: no identifiability for any subset of sources can be provided if there exists any degree of violations, either on the number of sources dependent on $\mathbf{u}$ or the number of values of $\mathbf{u}$. This limits the application of these results to ideal scenarios where all sources are influenced by the same auxiliary variable with sufficient changes without any type of compromise. In practice, however, it is often the case that only a subset of sources benefit from the additional information provided by auxiliary variables, different auxiliary variables may affect different sources, or auxiliary variables do not contain sufficient information. Assumption ii in Thm. 4.1 is originally from (Kong et al., 2022) and also necessitate the presence of change. Intuitively, the chance of having a subset $A_\mathbf{s}$ on which all domain distributions have an equal probability measure is very slim, which has been verified empirically in (Kong et al., 2022). For both theorems, we consider the more challenging undercomplete case, for which the related identifiability results are lacking in the literature. Additionally, unlike previous works assuming specific distributions of sources such as exponential families, we do not have similar distributional assumptions on the sources.

## 4.1 Results with flexible grouping structures

If we further have access to the dependence structure among variables in $\mathbf{s}_D$, additional identifiability results for these sources may also be established. For example, consider the setting that $\mathbf{s}_D = (s_{n_I+1}, \ldots, s_n)$ can be decomposed to $d$ irreducible independent subspaces $\{\mathbf{s}_{c_1}, \ldots, \mathbf{s}_{c_d}\}$, of which each is a multi-dimensional vector consisting multiple sources. We denote the $j$-th consecutive $d$-dimensional vector ($j$-th subspace) in $\mathbf{s}$ as $\mathbf{s}_{c_j} = (s_{(j-1)d+1}, \ldots, s_{jd}) = (s_{c_j(l)}, \ldots, s_{c_j(h)})$, where $s_{c_j(l)}$ and $s_{c_j(h)}$ are the first and the last sources in $\mathbf{s}_{c_j}$, respectively. Then we have

$$p_{\mathbf{s}|\mathbf{u}}(\mathbf{s}|\mathbf{u}) = \prod_{i=1}^{n_i} p_{s_i}(s_i) \prod_{j=c_1}^{c_d} p_{\mathbf{s}_{c_j}|\mathbf{u}}(\mathbf{s}_{c_j}|\mathbf{u}). \tag{4}$$

This is similar to Independent Subspace Analysis (ISA) (Hyvärinen and Hoyer, 2000; Theis, 2006) but we allow only a subset of sources as a composition of (conditionally) independent subspaces instead of all, which formalizes the tasks of blind source separation or uncovering latent variable models with mixtures of both high-dimensional and one-dimensional signals. The considered general setting essentially covers ICA and ISA as special cases: if $n_I = n$, it is consistent with the ICA problem; if $n_I = 0$, all sources can be decomposed into irreducible independent subspaces, and thus it becomes an ISA problem. The identifiability result under this setting is shown in the following theorem with its proof provided in Appx. A.4:

**Theorem 4.3.** *Let the observed data be a large enough sample generated from an undercomplete nonlinear ICA model as defined in Eqs. (2) and (4). Suppose the following assumptions hold:*

    *i. For each $i \in \{1, \ldots, n_I\}$, there exist $\{\mathbf{s}^{(\ell)}\}_{\ell=1}^{|\mathcal{F}_{i,:n_I}|}$ and a matrix $\mathrm{T} \in \mathcal{T}$ s.t. $\mathrm{span}\{\mathbf{J_f}(\mathbf{s}^{(\ell)})_{i,:n_I}\}_{\ell=1}^{|\mathcal{F}_{i,:n_I}|} = \mathbb{R}^{n_I}_{\mathcal{F}_{i,:n_I}}$ and $\left[\mathbf{J_f}(\mathbf{s}^{(\ell)})\mathrm{T}\right]_{i,:n_I} \in \mathbb{R}^{n_I}_{\hat{\mathcal{F}}_{i,:n_I}}$.*

    *ii. There exist $2n_D + 1$ values of $\mathbf{u}$, i.e., $\mathbf{u}_i$ with $i \in \{0, 1, \ldots, 2n_D\}$, s.t. the $2n_D$ vectors $\mathbf{w}(\mathbf{s}_D, \mathbf{u}_i) - \mathbf{w}(\mathbf{s}_D, \mathbf{u}_0)$ with $i \in \{1, \ldots, 2n_D\}$ are linearly independent, where vector $\mathbf{w}(\mathbf{s}_D, \mathbf{u}_i)$ is defined as follows:*

$$\mathbf{w}(\mathbf{s}_D, \mathbf{u}_i) = (\mathbf{v}(\mathbf{s}_{c_1}, \mathbf{u}_i), \cdots, \mathbf{v}(\mathbf{s}_{c_d}, \mathbf{u}_i), \mathbf{v}'(\mathbf{s}_{c_1}, \mathbf{u}_i), \cdots, \mathbf{v}'(\mathbf{s}_{c_d}, \mathbf{u}_i)),$$

*where*

$$\mathbf{v}(\mathbf{s}_{c_j}, \mathbf{u}_i) = \Big( \frac{\partial \log p(\mathbf{s}_{c_j}|\mathbf{u}_i)}{\partial s_{c_j^{(l)}}}, \cdots, \frac{\partial \log p(\mathbf{s}_{c_j}|\mathbf{u}_i)}{\partial s_{c_j^{(h)}}} \Big),$$

$$\mathbf{v}'(\mathbf{s}_{c_j}, \mathbf{u}_i) = \Big( \frac{\partial^2 \log p(\mathbf{s}_{c_j}|\mathbf{u}_i)}{(\partial s_{c_j^{(l)}})^2}, \cdots, \frac{\partial^2 \log p(\mathbf{s}_{c_j}|\mathbf{u}_i)}{(\partial s_{c_j^{(h)}})^2} \Big).$$

iii. *There exist* $\mathbf{u}_1, \mathbf{u}_2 \in \mathbf{u}$*, s.t., for any set* $A_\mathbf{s} \subseteq \mathcal{S}$ *with nonzero probability measure and cannot be expressed as* $B_{\mathbf{s}_I} \times \mathcal{S}_D$ *for any* $B_{\mathbf{s}_I} \subset \mathcal{S}_I$*, we have*

$$\int_{\mathbf{s} \in A_\mathbf{s}} p_{\mathbf{s}|\mathbf{u}}\left(\mathbf{s} \mid \mathbf{u}_1\right) d\mathbf{s} \neq \int_{\mathbf{s} \in A_\mathbf{s}} p_{\mathbf{s}|\mathbf{u}}\left(\mathbf{s} \mid \mathbf{u}_2\right) d\mathbf{s}.$$

iv. *(Structural Sparsity) For all* $k \in \{1, \dots, n_I\}$*, there exists* $\mathcal{C}_k$ *s.t.* $\bigcap_{i \in \mathcal{C}_k} \mathcal{F}_{i,:n_I} = \{k\}$*.*

*Then* $\mathbf{s}_I$ *is identifiable up to an element-wise invertible transformation and a permutation, and* $\mathbf{s}_D$ *is identifiable up to a subspace-wise invertible transformation and a subspace-wise permutation.*

All assumptions align with the same principles as those elaborated in the theorems proposed above and have been adapted to cater to the flexible grouping structure. Specifically, in addition to the identifiability of sources in $\mathbf{s}_I$, we prove that we can also identify sources in $\mathbf{s}_D$ up to an indeterminacy that, for each $c_i \in \{c_1, \dots, c_d\}$, there exists an invertible transformation $\mathbf{h}_{c_i}$ s.t. $\mathbf{h}_{c_i}(\mathbf{s}_{c_i}) = \hat{\mathbf{s}}_{c_i}$, which is analogous to the previous element-wise indeterminacy. Consequently, even when dealing with mixtures of high and one-dimensional sources, like in the case of multi-modal data, we can still recover the hidden generating process to some extent. Based on the aforementioned theoretical results, which consider undercompleteness, partial sparsity, and partial source dependence, Thm. 4.3 further generalizes the identifiability of nonlinear ICA by relaxing the dimensionality constraint of the latent generating factors.

In this vein, it is natural to consider another dependence structure, i.e., sources in $s_D$ are not marginally but conditionally independent given an auxiliary variable $\mathbf{u}$. This is similar to the assumption made in most previous works on identifiable nonlinear ICA with surrogate information, which assume that all sources are conditionally independent of each other given the auxiliary variable. However, our setting is more flexible in the sense that we do not assume all sources to be influenced by the auxiliary variable. Specifically, sources in $\mathbf{s}_I$ are mutually independent as in the original ICA setting, while only sources in $\mathbf{s}_D$ have access to the side information from the conditional independence given $\mathbf{u}$, i.e.,

$$p_{\mathbf{s}|\mathbf{u}}(\mathbf{s}|\mathbf{u}) = \prod_{i=1}^{n_I} p_{s_i}(s_i) \prod_{j=n_I+1}^{n} p_{s_j|\mathbf{u}}(s_j|\mathbf{u}). \tag{5}$$

The identifiability result for all sources ($\mathbf{s}_I$ and $\mathbf{s}_D$) is as follows with proof in Appx. A.5:

**Theorem 4.4.** *Let the observed data be a large enough sample generated from an undercomplete nonlinear ICA model as defined in Eqs. (2) and (5), suppose the following assumptions hold:*

i. *For each* $i \in \{1, \dots, n_I\}$*, there exist* $\{\mathbf{s}^{(\ell)}\}_{\ell=1}^{|\mathcal{F}_{i,:n_I}|}$ *and a matrix* $\mathrm{T} \in \mathcal{T}$ *s.t.* $\mathrm{span}\{\mathbf{J_f}(\mathbf{s}^{(\ell)})_{i,:n_I}\}_{\ell=1}^{|\mathcal{F}_{i,:n_I}|} = \mathbb{R}^{n_I}_{\mathcal{F}_{i,:n_I}}$ *and* $\big[\mathbf{J_f}(\mathbf{s}^{(\ell)})\mathrm{T}\big]_{i,:n_I} \in \mathbb{R}^{n_I}_{\hat{\mathcal{F}}_{i,:n_I}}$*.*

ii. *There exist* $2n_D + 1$ *values of* $\mathbf{u}$*, i.e.,* $\mathbf{u}_i$ *with* $i \in \{0, 1, \dots, 2n_D\}$*, s.t. the* $2n_D$ *vectors* $\mathbf{w}(\mathbf{s}_D, \mathbf{u}_i) - \mathbf{w}(\mathbf{s}_D, \mathbf{u}_0)$ *with* $i \in \{1, \dots, 2n_D\}$ *are linearly independent, where vector* $\mathbf{w}(\mathbf{s}_D, \mathbf{u})$ *is defined as follows:*

$$\mathbf{w}(\mathbf{s}_D, \mathbf{u}_i) = (\mathbf{v}(\mathbf{s}_D, \mathbf{u}_i), \mathbf{v}'(\mathbf{s}_D, \mathbf{u}_i)),$$

*where*

$$\mathbf{v}(\mathbf{s}_D, \mathbf{u}_i) = \Big( \frac{\partial \log p(s_{n_I+1}|\mathbf{u}_i)}{\partial s_{n_I+1}}, \cdots, \frac{\partial \log p(s_n|\mathbf{u}_i)}{\partial s_n} \Big),$$

$$\mathbf{v}'(\mathbf{s}_D, \mathbf{u}_i) = \Big( \frac{\partial^2 \log p(s_{n_I+1}|\mathbf{u}_i)}{(\partial s_{n_I+1})^2}, \cdots, \frac{\partial^2 \log p(s_n|\mathbf{u}_i)}{(\partial s_n)^2} \Big).$$

iii. *There exist* $\mathbf{u}_1, \mathbf{u}_2 \in \mathbf{u}$*, s.t., for any set* $A_\mathbf{s} \subseteq \mathcal{S}$ *with nonzero probability measure and cannot be expressed as* $B_{\mathbf{s}_I} \times \mathcal{S}_D$ *for any* $B_{\mathbf{s}_I} \subset \mathcal{S}_I$*, we have*

$$\int_{\mathbf{s} \in A_\mathbf{s}} p_{\mathbf{s}|\mathbf{u}}\left(\mathbf{s} \mid \mathbf{u}_1\right) d\mathbf{s} \neq \int_{\mathbf{s} \in A_\mathbf{s}} p_{\mathbf{s}|\mathbf{u}}\left(\mathbf{s} \mid \mathbf{u}_2\right) d\mathbf{s}.$$

*iv. (Structural Sparsity) For all $k \in \{1, \dots, n_I\}$, there exists $\mathcal{C}_k$ s.t. $\bigcap_{i \in \mathcal{C}_k} \mathcal{F}_{i,:n_I} = \{k\}$.*

*Then $\mathbf{s}$ is identifiable up to an element-wise invertible transformation and a permutation.*

With different assumptions on different sets of sources, one could view this theorem as an expansion of both previous theoretical findings that impose distributional constraints on sources with auxiliary variables (e.g., (Hyvärinen et al., 2019)) and those that constrain the mixing function with *Structural Sparsity* (Zheng et al., 2022). This is particularly helpful in the context of self-supervised learning (Von Kügelgen et al., 2021) or transfer learning (Kong et al., 2022), where latent representations are modeled as a changing part and an invariant part. In (Kong et al., 2022), the component-wise identifiability for variables changing across domains (i.e., $\mathbf{s}_D$ with multiple values of $\mathbf{u}$ in our setting) are provided but not those in the invariant part (i.e., $\mathbf{s}_I$ in our setting). With the help of Thm. 4.4, we can show identifiability up to an element-wise invertible transformation and a permutation for each source, regardless of whether it changes across domains or not, which may help some related tasks where full identifiability is necessary. Furthermore, for causal reasoning or disentanglement with observational time-series data, our theorem benefits the identifiability of temporal processes involving instantaneous relations. Previous works in that area can only deal with time-delayed/changing influences, as they rely on the global conditional independence of all sources given the changing time index as the auxiliary variable (Hyvärinen and Morioka, 2016, 2017; Yao et al., 2022). Our theorem, on the other hand, provides the added ability to identify unconditional sources, thanks to the partially satisfied sparsity assumption, and thus aids the uncovering of latent processes with instantaneous relations.

## 5  Experiments

In order to validate the proposed identifibaility results, we conduct experiments using both simulated data and real-world images. It is noteworthy that there has been extensive research that has empirically verified that deep latent variable models are likely to be identifiable in complex scenarios, particularly in the disentanglement task (Kumar et al., 2017; Klys et al., 2018; Locatello et al., 2018; Rubenstein et al., 2018; Chen et al., 2018; Burgess et al., 2018; Duan et al., 2020; Falck et al., 2021; Carbonneau et al., 2022). While we are not sure of the exact inductive biases or side information that are available during the real-world application, which has been proved to be necessary (Hyvärinen and Pajunen, 1999; Locatello et al., 2019), the empirical success of these methods sheds light on the possibility of identification in the general settings considered in this work.

**Setup.** For settings with the auxiliary variable $\mathbf{u}$, $\mathbf{u}$ is always available during estimation and we consider the dataset as $\mathcal{D} = \left\{ \left( \mathbf{x}^{(1)}, \mathbf{u}^{(1)} \right), \dots, \left( \mathbf{x}^{(N)}, \mathbf{u}^{(N)} \right) \right\}$, where $N$ is the sample size and $\mathbf{u}^{(i)}$ is the value of $\mathbf{u}$ (or class label) corresponding to the data point $\mathbf{x}^{(i)}$. Given the estimated model $\hat{f}$ parameterized by $\theta$, similar to (Sorrenson et al., 2020), we consider a regularized maximum-likelihood approach for the required sparsity regularization during estimation with the objective function as: $\mathcal{L}(\theta) = \mathbb{E}_{(\mathbf{x},\mathbf{u}) \in \mathcal{D}} \left[ \log p_{\hat{\mathbf{f}}^{-1}}(\mathbf{x}|\mathbf{u}) - \lambda \mathbf{R} \right]$, where $\lambda \in [0, 1]$ is a regularization parameter and $\mathbf{R}$ is the regularization term on the Jacobian of the estimated mixing function, i.e., $\mathbf{J}_{\hat{f}}$. Based on our experimental results (Fig. 8 in Appx. B.2), we adopt the minimax concave penalty (MCP) (Zhang, 2010) as the regularization term. For settings without the auxiliary variable, we remove the access of $\mathbf{u}$ and follow the same objective function in (Zheng et al., 2022). We train a General Incompressible-flow Network (GIN) (Sorrenson et al., 2020), which is a flow-based generative model, to maximize the objective function $\mathcal{L}(\cdot)$. Following (Sorrenson et al., 2020), where necessary, we concatenate the latent sources with independent Gaussian noises to meet the dimensionality requirements. All results are from 20 trials with random seeds. Additional details of the experimental setup are included in Appx. B.

**Ablation study.** We perform an ablation study to verify the necessity of the proposed assumptions. Specifically, we focus on the following models corresponding to different assumptions: *(UCSS)* The assumption of Structural Sparsity, as well as other assumptions in the undercomplete case (Thm. 3.1), are satisfied; *(Mixed)* The assumption of Structural Sparsity with undercompleteness and the required dependence structure among sources influenced by the auxiliary variable, as well as other assumptions in the partial sparsity and dependence case (Thm. 4.4), are satisfied; *(Base)* The vanilla baseline in the undercomplete case, where the assumption of Structural Sparsity is not satisfied compared to *UCSS*. The datasets are generated according to the required assumptions, the details of which are included in Appx. B. All experiments are conducted in the undercomplete case, where the number of observed variables is twice the number of sources. For datasets that contain both sources in $\mathbf{s}_I$ and $\mathbf{s}_D$ (*Mixed* case), we set half as $\mathbf{s}_I$ and the other half as $\mathbf{s}_D$, and the minimum required

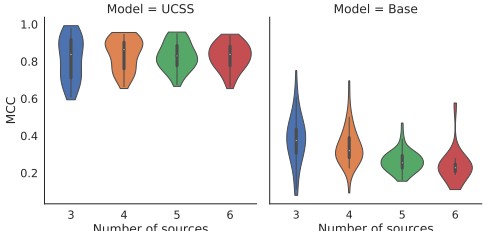

Figure 2: MCC of *UCSS* w.r.t. different number of sources.

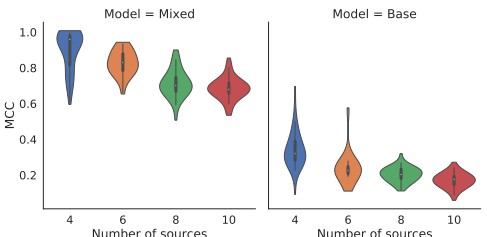

Figure 3: MCC of *Mixed* w.r.t. different number of sources.

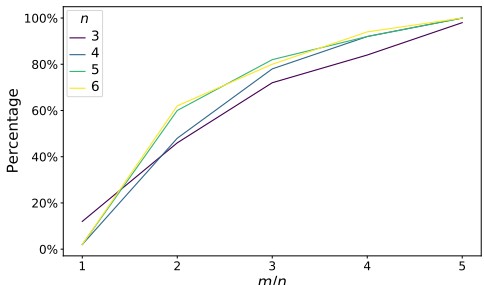

Figure 4: Percentage of random structures satisfying Structural Sparsity w.r.t. different degree of undercompleteness (i.e., $m/n$).

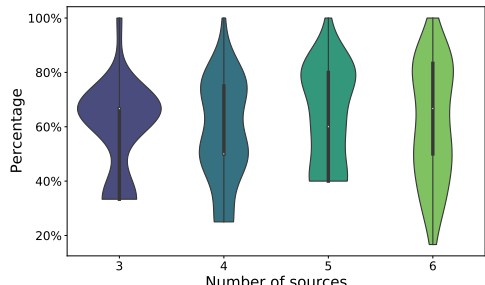

Figure 5: Percentage of sources satisfying *Structural Sparsity* w.r.t. different numbers of sources in the bijective setting ($m/n = 1$).

numbers of required distinct values have been assigned to the auxiliary variable **u**. Following previous works (Hyvärinen and Morioka, 2016; Lachapelle et al., 2022), we use the mean correlation coefficient (MCC) between the true sources and the estimated ones as the evaluation metric.

Results for each model are summarised in Fig. 2 and Fig. 3. It can be observed that when the proposed assumptions are met (*UCSS* and *Mixed*), our models achieve higher MCCs than *Base*. This indicates that it is indeed possible to identify sources from nonlinear mixtures up to trivial indeterminacy in the general settings with undercompleteness, partial sparsity, and partial source dependence. Additionally, we conduct experiments with different numbers of sources $n$ to evaluate the stability of the identification. Our results show that both models consistently outperform *Base* across all values of $n$, further supporting the theoretical claims.

**Undercomplete Structural Sparsity.**    As previously discussed, the assumption of *Structural Sparsity* (Assumption ii in Thm. 3.1) is far more plausible in an undercomplete setting considered in our theory (i.e., the number of observed variables $m$ is larger than the number of sources $n$) than in the more restrictive bijective scenario required in (Zheng et al., 2022) (i.e., the numbers are equal, $m = n$). Consequently, extending the identifiability with structural sparsity from a bijective to an undercomplete setting significantly broadens its applicability in real-world contexts. In order to validate the necessity of the proposed generalization empirically, we construct several experiments studying the *Structural Sparsity* assumption in the undercomplete case. We consider different numbers of sources $n$ with different degrees of undercompleteness ($m/n$, where $m$ is the number of observed variables). For each setting, we generate 50 random matrices where each entry is independently determined with an equal probability to be either zero or non-zero. The results of the percentages of matrices satisfying the assumption of *Structural Sparsity* are presented in Fig. 4. We could observe that there exists a significant gap on the percentages between the cases where $m/n = 1$, i.e., the bijective setting, and the undercomplete settings where $m/n > 1$. Thus, it is clear that the assumption is much more likely to hold true when we have more observed variables than sources. Furthermore, when the degree of undercompleteness increases, the percentage of cases satisfying structural converges to 1. This further suggests that the assumption will almost always hold with a sufficient degree of undercompleness, which is rather common in practice. For instance, a photo can easily have millions of pixels (observed variables) but only a dozen of hidden concepts (sources).

**Partial Structural Sparsity.**    Moreover, as previously noted, it is not uncommon for *Structural Sparsity* to be violated for a subset of sources. For instance, certain sources (such as high-decibel sound sources) may exert influence over all observed variables (microphones). Nonetheless, the

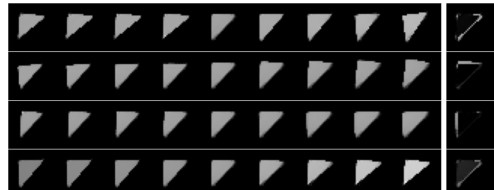

Figure 6: Results on Triangles. The rows may correspond to rotation, height, width, and brightness, respectively.

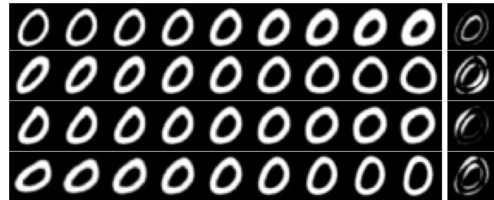

Figure 7: Results on EMNIST. The rows may correspond to line thickness, angle, upper width, and height, respectively.

prior study (Zheng et al., 2022) necessitates that the sparsity assumption holds true for all sources, providing no identifiability assurance in cases of any degree of violation. To confront this practical obstacle, we propose Thm. 4.1 and Thm. 4.2 to demonstrate that the remaining sources (i.e., $n_I$ sources) can still be identified even when the sparsity assumption does not universally hold for some sources. These results can also be motivated empirically. For instance, from Fig. 4, one may find that *Structural Sparsity* does not likely to hold for all sources when $m/n = 1$, which is the bijective setting considered in (Zheng et al., 2022). However, as discussed above, if a subset of sources satisfies the assumption, at least the identifiability for these sources could be guaranteed by our proposed theorems (Thm. 4.1 and Thm. 4.2) under certain conditions. To illustrate this, we conduct experiments in the bijective setting ($m/n = 1$) and report the percentage of sources satisfying *Structural Sparsity* in Fig. 4. We consider datasets with different number of sources and generate 50 random matrices for each of these. Each entry is independently determined with an equal probability to be either zero or non-zero. Combining results from both Fig. 4 and Fig. 5, we observe that, even in scenarios where *Structural Sparsity* is rarely satisfied for all sources ($m/n = 1$ in Fig. 4), it is almost always satisfied for a significant fraction of sources (Fig. 5). Consequently, our generalization also proves helpful even within the confines of the earlier bijective setting.

**Image datasets.** To study how reasonable the proposed theories are w.r.t. the practical generating process of observational data in complex scenarios, we conduct experiments on "Triangles" (Yang et al., 2022) and EMNIST (Cohen et al., 2017) datasets. The "Triangles" dataset consists of $60,000$ synthetic $28 \times 28$ images of triangles, which are generated from 4 factors: rotation, height, width, and brightness. By fixing the number of pixels (observed variables) and generating factors (sources), we can guarantee that the images are generated according to an undercomplete process, although the exact generating process is still unknown (e.g., a pixel could be (indirectly) influenced by multiple factors in a complicated way). For the real-world dataset, EMNIST contains $240,000$ $28 \times 28$ images of handwritten digits and is a larger version of the classical MNIST dataset. Although we do not know the exact number of sources, it is highly possible that it is smaller than the number of pixels (784). We present the identified sources with the top four standard deviations (SDs) from both datasets in Fig. 6 and Fig. 7. In both figures, each row represents a source identified by our model, with it varying from $-4$ to $+4$ SDs to illustrate its influence. The rightmost column is a heat map given by the absolute pixel difference between $-1$ and $+1$ SDs. By observing the identified sources with the top four standard deviations, one could find that it is possible to identify semantically meaningful attributes from practical image datasets, which further suggests the potential of our theory in real-world scenarios. Additional results are available in Appx. B.2.

## 6 Conclusion

We establish a set of new identifiability results of nonlinear ICA in general settings with undercompleteness, partial sparsity and source dependence, and flexible grouping structures, thereby extending the identifiabilty theory to a wide range of real-world scenarios. Specifically, we prove the identifiability when there are more observed variables than underlying sources, and when sparsity and/or independence are not met for a subset of sources. Moreover, by leveraging various dependence structures among sources, further identifiability guarantees can also be obtained. Theoretical results have been validated through a combination of extensive previous studies and our own experiments, which involve both synthetic and real-world datasets. Future work includes adopting the theoretical framework for related tasks, such as disentanglement, transfer learning, and causal discovery. Furthermore, the proposed identifiability guarantees on generalized latent variable models bolster our confidence in uncovering hidden truths across diverse real-world settings in scientific discovery. We have only explored the visual disentanglement task, and the lack of other applications is a limitation of this work.

## Acknowledgements

We are grateful to everyone involved in the anonymous reviewing process for their insightful feedback. This project is partially supported by NSF Grant 2229881, the National Institutes of Health (NIH) under Contract R01HL159805, a grant from Apple Inc., a grant from KDDI Research Inc., and generous gifts from Salesforce Inc., Microsoft Research, and Amazon Research.

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

# Appendix

## Table of Contents

## A   Proofs

### A.1   Proof of Theorem 3.1

**Theorem 3.1.** *Let the observed data be a large enough sample generated by an undercomplete nonlinear ICA model as defined in Eqs. (1) and (2). Suppose the following assumptions hold:*

> *i. For each $i \in \{1, \ldots, n\}$, there exist $\{\mathbf{s}^{(\ell)}\}_{\ell=1}^{|\mathcal{F}_{i,:}|}$ and a matrix $\mathrm{T} \in \mathcal{T}$ s.t. $\mathrm{span}\{\mathbf{J_f}(\mathbf{s}^{(\ell)})_{i,:}\}_{\ell=1}^{|\mathcal{F}_{i,:}|} = \mathbb{R}^n_{\mathcal{F}_{i,:}}$ and $\left[\mathbf{J_f}(\mathbf{s}^{(\ell)})\mathrm{T}\right]_{i,:} \in \mathbb{R}^n_{\hat{\mathcal{F}}_{i,:}}$.*

> *ii. (Structural Sparsity) For each $k \in \{1, \ldots, n\}$, there exists $\mathcal{C}_k$ s.t. $\bigcap_{i \in \mathcal{C}_k} \mathcal{F}_{i,:} = \{k\}$.*

*Then $\mathbf{s}$ is identifiable up to an element-wise invertible transformation and a permutation.*

*Proof.* Let $\mathbf{h} : \mathbf{s} \to \hat{\mathbf{s}}$ denotes the transformation between the true sources and estimated sources. We can apply the chain rule repeatedly to get:

$$\begin{aligned} \mathbf{J_f}(\mathbf{s}) &= \mathbf{J_{\hat{f} \circ h}}(\mathbf{s}) \\ &= \mathbf{J_{\hat{f}}}(\hat{\mathbf{s}})\mathbf{J_h}(\mathbf{s}). \end{aligned} \tag{6}$$

Since $\mathbf{J_{\hat{f}}}(\hat{\mathbf{s}})$ and $\mathbf{J_f}(\mathbf{s})$ both possess full column rank, $\mathbf{J_h}(\mathbf{s})$ should have a non-zero determinant. From this, we can deduce by incorporating the inverse of $\mathbf{h}$:

$$\mathbf{J_{\hat{f}}}(\hat{\mathbf{s}}) = \mathbf{J_f}(\mathbf{s})\mathbf{J_h}(\mathbf{s})^{-1}. \tag{7}$$

Our objective here is to demonstrate that the function $\mathbf{h}$ is a composition of a permutation and a component-wise invertible transformation. Let $\mathbf{D}(\mathbf{s})$ denote a diagonal matrix and $\mathbf{P}$ denote a permutation matrix, our goal can be rewritten as demonstrating that $\mathbf{J_h}(\mathbf{s})^{-1} = \mathbf{D}(\mathbf{s})\mathbf{P}$. This leads us to demonstrate that:

$$\mathbf{J_{\hat{f}}}(\hat{\mathbf{s}}) = \mathbf{J_f}(\mathbf{s})\mathbf{D}(\mathbf{s})\mathbf{P}. \tag{8}$$

Further, we can express:

$$\mathbf{J_{\hat{f}}}(\hat{\mathbf{s}}) = \mathbf{J_f}(\mathbf{s})\mathbf{T}(\mathbf{s}), \tag{9}$$

where $\mathbf{T}(\mathbf{s}) \in \mathbb{R}^{n \times n}$ is a square matrix. Here, we define $\mathcal{F}$ as the support of $\mathbf{J_f}(\mathbf{s})$, $\hat{\mathcal{F}}$ as the support of $\mathbf{J_{\hat{f}}}(\hat{\mathbf{s}})$ and $\mathcal{T}$ as a set of matrices with the same support of $\mathbf{T}(\mathbf{s})$. Furthermore, $\mathrm{T} \in \mathcal{T}$ is a matrix with the same support as $\mathbf{T}(\mathbf{s})$. Based on Assumption i, we have:

$$\mathrm{span}\{\mathbf{J_f}(\mathbf{s}^{(\ell)})_{i,:}\}_{\ell=1}^{|\mathcal{F}_{i,:}|} = \mathbb{R}^n_{\mathcal{F}_{i,:}}. \tag{10}$$

Given that the set $\{\mathbf{J_f}(\mathbf{s}^{(\ell)})_{i,:}\}_{\ell=1}^{|\mathcal{F}_{i,:}|}$ forms a basis of $\mathbb{R}^n_{\mathcal{F}_{i,:}}$, we can express any vector in this space as a linear combination of these basis vectors. In particular, for any $j_0 \in \mathcal{F}_{i,:}$, the one-hot vector $e_{j_0} \in \mathbb{R}^n_{\mathcal{F}_{i,:}}$ can be written as

$$e_{j_0} = \sum_{\ell \in \mathcal{F}_{i,:}} \alpha_\ell \mathbf{J_f}(\mathbf{s}^{(\ell)})_{i,:}, \tag{11}$$

where $\alpha_\ell$ denotes the respective coefficient.

With this in mind, we can find the transformation of $e_{j_0}$ under $\mathrm{T}$ as

$$\mathrm{T}_{j_0,:} = e_{j_0}\mathrm{T} = \sum_{\ell \in \mathcal{F}_{i,:}} \alpha_\ell \mathbf{J_f}(\mathbf{s}^{(\ell)})_{i,:}\mathrm{T}. \tag{12}$$

According to Assumption i, each term in the above summation belongs to the space $\mathbb{R}^n_{\hat{\mathcal{F}}_{i,:}}$. Therefore, $\mathrm{T}_{j_0,:}$ itself resides in $\mathbb{R}^n_{\hat{\mathcal{F}}_{i,:}}$, i.e., $\mathrm{T}_{j_0,:} \in \mathbb{R}^n_{\hat{\mathcal{F}}_{i,:}}$. Thus

$$\forall j \in \mathcal{F}_{i,:}, \ \mathrm{T}_{j,:} \in \mathbb{R}^n_{\hat{\mathcal{F}}_{i,:}}. \tag{13}$$

Then the connections between these supports can be established according to Defn. 2.3

$$\forall (i,j) \in \mathcal{F}, \{i\} \times \mathcal{T}_{j,:} \subset \hat{\mathcal{F}}. \tag{14}$$

It is noteworthy that a similar strategy to derive Eq. 14 has been applied in (Zheng et al., 2022) and part of the proof technique is inspired by that work. In contrast to the proof by Zheng et al. (2022), which assumes the invertibility of $f$, we only necessitate its injectivity. This distinction allows for the inclusion of undercomplete cases.

Since $\mathbf{J_f}(\mathbf{s}^{(\ell)})$ and $\mathbf{J_{\hat{f}}}(\hat{\mathbf{s}}^{(\ell)})$ have full column rank $n$, $\mathbf{T}(\mathbf{s}^{(\ell)})$ must have a non-zero determinant. Otherwise, it would follow that the rank of $\mathbf{T}(\mathbf{s}^{(\ell)})$ is less than $n$, which would imply a contradiction that $\mathbf{J_{\hat{f}}}(\hat{\mathbf{s}}^{(\ell)}) = \mathbf{J_f}(\mathbf{s}^{(\ell)})\mathbf{T}(\mathbf{s}^{(\ell)})$ has a column rank less than $n$. Representing the determinant of the matrix $\mathbf{T}(\mathbf{s}^{(\ell)})$ as its Leibniz formula yields

$$\det(\mathbf{T}(\mathbf{s}^{(\ell)})) = \sum_{\sigma \in \mathcal{S}_n} \left( \mathrm{sgn}(\sigma) \prod_{i=1}^n \mathbf{T}(\mathbf{s}^{(\ell)})_{i,\sigma(i)} \right) \neq 0, \tag{15}$$

where $\mathcal{S}_n$ is the set of $n$-permutations. Thus, there is at least one term in the sum that is non-zero, i.e.,

$$\exists \sigma \in \mathcal{S}_n, \ \forall i \in \{1, \ldots, n\}, \ \mathrm{sgn}(\sigma) \prod_{i=1}^n \mathbf{T}(\mathbf{s}^{(\ell)})_{i,\sigma(i)} \neq 0, \tag{16}$$

which is equivalent to

$$\exists \sigma \in \mathcal{S}_n, \ \forall i \in \{1, \ldots, n\}, \ \mathbf{T}(\mathbf{s}^{(\ell)})_{i,\sigma(i)} \neq 0. \tag{17}$$

Then we can conclude that this $\sigma$ is in the support of $\mathbf{T}(\mathbf{s})$ since $\mathbf{s}^{(\ell)} \in \mathbf{s}$. Therefore, it follows that

$$\forall j \in \{1, \ldots, n\}, \ \sigma(j) \in \mathcal{T}_{j,:}. \tag{18}$$

Together with Eq. (14), we have

$$\forall (i,j) \in \mathcal{F}, (i, \sigma(j)) \in \{i\} \times \mathcal{T}_{j,:} \subset \hat{\mathcal{F}}. \tag{19}$$

Denote

$$\sigma(\mathcal{F}) = \{(i, \sigma(j)) \mid (i,j) \in \mathcal{F}\}. \tag{20}$$

Then we have

$$\sigma(\mathcal{F}) \subset \hat{\mathcal{F}}. \tag{21}$$

Because of the sparsity regularization on the estimated Jacobian, we further have

$$|\hat{\mathcal{F}}| \leq |\mathcal{F}| = |\sigma(\mathcal{F})|. \tag{22}$$

Combining this with Eq. (21), we derive

$$\sigma(\mathcal{F}) = \hat{\mathcal{F}}. \tag{23}$$

Suppose $\mathbf{T}(\mathbf{s}) \neq \mathbf{D}(\mathbf{s})\mathbf{P}$, then

$$\exists j_1 \neq j_2, \ \mathcal{T}_{j_1,:} \cap \mathcal{T}_{j_2,:} \neq \emptyset. \tag{24}$$

Additionally, consider $j_3 \in \{1, \ldots, n\}$ for which

$$\sigma(j_3) \in \mathcal{T}_{j_1,:} \cap \mathcal{T}_{j_2,:}. \tag{25}$$

Since $j_1 \neq j_2$, we can assume $j_3 \neq j_1$ without loss of generality. A similar strategy has been used previously in (Lachapelle et al., 2022; Zheng et al., 2022). Based on Assumption ii, there exists $\mathcal{C}_{j_1} \ni j_1$ such that $\bigcap_{i \in \mathcal{C}_{j_1}} \mathcal{F}_{i,:} = \{j_1\}$. Because

$$j_3 \notin \{j_1\} = \bigcap_{i \in \mathcal{C}_{j_1}} \mathcal{F}_{i,:}, \tag{26}$$

there must exists $i_3 \in \mathcal{C}_{j_1}$ such that

$$j_3 \notin \mathcal{F}_{i_3,:}. \tag{27}$$

Since $j_1 \in \mathcal{F}_{i_3,:}$, it follows that $(i_3, j_1) \in \mathcal{F}$. Therefore, according to Eq. (14), we have

$$\{i_3\} \times \mathcal{T}_{j_1,:} \subset \hat{\mathcal{F}}. \tag{28}$$

Notice that $\sigma(j_3) \in \mathcal{T}_{j_1,:} \cap \mathcal{T}_{j_2,:}$ implies

$$(i_3, \sigma(j_3)) \in \{i_3\} \times \mathcal{T}_{j_1,:}. \tag{29}$$

Then by Eqs. (28) and (29), we have

$$(i_3, \sigma(j_3)) \in \hat{\mathcal{F}}. \tag{30}$$

This further implies $(i_3, j_3) \in \mathcal{F}$ by Eq. (20) and (23), which contradicts Eq. (27). Therefore, we have proven by contradiction that $\mathbf{T}(\mathbf{s}) = \mathbf{D}(\mathbf{s})\mathbf{P}$. By replacing $\mathbf{T}(\mathbf{s})$ with $\mathbf{D}(\mathbf{s})\mathbf{P}$ in Eq. (9), we obtain Eq. (8), which is the goal. □

### A.2 Proof of Theorem 4.1

**Theorem 4.1.** *Let the observed data be a large enough sample generated by an undercomplete nonlinear ICA model defined in Eqs. (2) and (3). Suppose the following assumptions hold:*

    i. *There exist $n_D + 1$ distinct values of $\mathbf{u}$, i.e., $\mathbf{u}_j$ with $j \in \{0, 1, \ldots, n_D\}$, s.t. the $n_D$ vectors $\mathbf{w}(\mathbf{s}_D, \mathbf{u}, i)$ with $i \in \{n_I + 1, ..., n\}$ are linearly independent, where vector $\mathbf{w}(\mathbf{s}_D, \mathbf{u}, i)$ is defined as follows:*

$$\mathbf{w}(\mathbf{s}_D, \mathbf{u}, i) = \left( \frac{\partial \left( \log p(\mathbf{s}_D|\mathbf{u}_1) - \log(p(\mathbf{s}_D|\mathbf{u}_0)) \right)}{\partial s_i}, \ldots, \frac{\partial \left( \log p(\mathbf{s}_D|\mathbf{u}_{n_D}) - \log(p(\mathbf{s}_D|\mathbf{u}_0)) \right)}{\partial s_i} \right).$$

    ii. *There exist $\mathbf{u}_1, \mathbf{u}_2 \in \mathbf{u}$, s.t., for any set $A_\mathbf{s} \subseteq \mathcal{S}$ with non-zero probability measure and cannot be expressed as $B_{\mathbf{s}_I} \times \mathbf{s}_D$ for any $B_{\mathbf{s}_I} \subset \mathcal{S}_I$, we have*

$$\int_{\mathbf{s} \in A_\mathbf{s}} p_{\mathbf{s}|\mathbf{u}} (\mathbf{s} \mid \mathbf{u}_1) \, d\mathbf{s} \neq \int_{\mathbf{s} \in A_\mathbf{s}} p_{\mathbf{s}|\mathbf{u}} (\mathbf{s} \mid \mathbf{u}_2) \, d\mathbf{s}.$$

*Then $\mathbf{s}_D$ is identifiable up to an subspace-wise invertible transformation.*

*Proof.* Let $h : \mathbf{s} \to \hat{\mathbf{s}}$ denotes the transformation between the true and estimated sources. By using chain rule repeatedly, we have

$$\begin{aligned} \mathbf{J}_\mathbf{f}(\mathbf{s}) &= \mathbf{J}_{\hat{\mathbf{f}} \circ \mathbf{h}}(\mathbf{s}) \\ &= \mathbf{J}_{\hat{\mathbf{f}}}(\hat{\mathbf{s}}) \mathbf{J}_\mathbf{h}(\mathbf{s}). \end{aligned} \tag{31}$$

Since $\mathbf{J}_{\hat{\mathbf{f}}}(\hat{\mathbf{s}})$ and $\mathbf{J}_\mathbf{f}(\mathbf{s})$ both possess full column rank, $\mathbf{J}_\mathbf{h}(\hat{\mathbf{s}})$ should have a non-zero determinant. Thus, $\mathbf{J}_\mathbf{h}(\mathbf{s})$ must be invertible and have a non-zero determinant. Otherwise, one of them would not be of full column rank, which leads to a contradiction.

Applying the change of variable rule, we have

$$p_{\mathbf{s}|\mathbf{u}}(\mathbf{s}|\mathbf{u})| \det(\mathbf{J}_{\mathbf{h}^{-1}}(\hat{\mathbf{s}}))| = p_{\hat{\mathbf{s}}}(\hat{\mathbf{s}}|\mathbf{u}). \tag{32}$$

Taking the logarithm on both sides yields

$$\log p_{\mathbf{s}|\mathbf{u}}(\mathbf{s}|\mathbf{u}) + \log|\det(\mathbf{J}_{\mathbf{h}^{-1}}(\hat{\mathbf{s}}))| = \log p_{\hat{\mathbf{s}}|\mathbf{u}}(\hat{\mathbf{s}}|\mathbf{u}). \tag{33}$$

Note that according to the model defined in Eqs. (2) and (3), the joint densities can be factorized as

$$p_{\mathbf{s}|\mathbf{u}}(\mathbf{s}|\mathbf{u}) = p_{\mathbf{s}_D|\mathbf{u}}(\mathbf{s}_D|\mathbf{u}) \prod_{i=1}^{n_I} p_{s_i}(s_i),$$

$$p_{\hat{\mathbf{s}}|\mathbf{u}}(\hat{\mathbf{s}}|\mathbf{u}) = p_{\hat{\mathbf{s}}_D|\mathbf{u}}(\hat{\mathbf{s}}_D|\mathbf{u}) \prod_{i=1}^{n_I} p_{\hat{s}_i}(\hat{s}_i). \tag{34}$$

Then we define the difference across domains as follows

$$q(\mathbf{s}, \mathbf{u}_j) = \log p_{\mathbf{s}|\mathbf{u}}(\mathbf{s}|\mathbf{u}_j) - \log p_{\mathbf{s}|\mathbf{u}}(\mathbf{s}|\mathbf{u}_0). \tag{35}$$

By taking the difference on both sides of Eq. (33) corresponding to $\mathbf{u}_j$ and $\mathbf{u}_0$, we obtain

$$q(\hat{\mathbf{s}}_D, \mathbf{u}_j) = q(\mathbf{s}_D, \mathbf{u}_j), \tag{36}$$

where, for $i \in \{1, \dots, n_I\}$, $\log p_{\hat{s}|\mathbf{u}}(\hat{s}_i|\mathbf{u}_j)$ and $\log p_{s|\mathbf{u}}(s_i|\mathbf{u}_j)$ have been canceled because sources in $\mathbf{s}_I$ are not dependent on $\mathbf{u}_j$. That is, $q(s_i, \mathbf{u}_j) = 0$ if $i \in \{1, \dots, n_I\}$.

Taking the derivatives of both sides of Eq. (36) w.r.t. $\hat{s}_k$ where $k \in \{1, \dots, n_I\}$, we have

$$\frac{\partial q(\hat{\mathbf{s}}_D, \mathbf{u}_j)}{\partial \hat{s}_k} = \sum_{i=n_I+1}^{n} \left( \frac{\partial q(\mathbf{s}_D, \mathbf{u}_j)}{\partial s_i} \frac{\partial s_i}{\partial \hat{s}_k} \right), \tag{37}$$

Clearly, LHS of Eq. (37) equals zero. By considering each $j \in \{1, \dots, n_D\}$ for $\mathbf{u}_j$, we have $n_D$ equations like Eq. (37), which constitute a linear system with a $n_D \times n_D$ coefficient matrix.

According to the assumption, the coefficient matrix of the linear system has full rank. Thus, the only solution of Eq. (37) is $\frac{\partial s_i}{\partial \hat{s}_k} = 0$ for $i \in \{n_I + 1, \dots, n\}$ and $k \in \{1, \dots, n_I\}$.

As $\mathbf{h}^{-1}(\cdot)$ is smooth, its Jacobian can be written as:

$$\mathbf{J}_{\mathbf{h}^{-1}}(\hat{\mathbf{s}}) = \left[ \begin{array}{c|c} \mathbf{A} := \frac{\partial \mathbf{s}_I}{\partial \hat{\mathbf{s}}_I} & \mathbf{B} := \frac{\partial \mathbf{s}_I}{\partial \hat{\mathbf{s}}_D} \\ \hline \mathbf{C} := \frac{\partial \mathbf{s}_D}{\partial \hat{\mathbf{s}}_I} & \mathbf{D} := \frac{\partial \mathbf{s}_D}{\partial \hat{\mathbf{s}}_D} \end{array} \right]. \tag{38}$$

Since $\frac{\partial \mathbf{s}_j}{\partial \hat{\mathbf{s}}_k} = 0$ for $j \in \{n_I + 1, \dots, n\}$ and $k \in \{1, \dots, n_I\}$, entries in the submatrix $\mathbf{C}$ must all be zero. Thus, the submatrix $\mathbf{D}$ must be invertible, otherwise $\mathbf{J}_{\mathbf{h}^{-1}}(\hat{\mathbf{s}})$ will not be invertible, which is a contradiction. Besides, based on Assumption ii, one can show that all entries in the submatrix $\mathbf{B}$ are zero according to part of the proof of Theorem 4.2 in (Kong et al., 2022) (Steps 1, 2, and 3). Therefore, $\hat{\mathbf{s}}_D$ is an invertible transformation of $\mathbf{s}_D$. □

### A.3 Proof of Theorem 4.2

**Theorem 4.2.** *In addition to assumptions in Thm. 4.1, suppose the following assumptions hold:*

*i. For each $i \in \{1, \dots, n_I\}$, there exist $\{\mathbf{s}^{(\ell)}\}_{\ell=1}^{|\mathcal{F}_{i,:n_I}|}$ and a matrix $\mathrm{T} \in \mathcal{T}$ s.t. $\mathrm{span}\{\mathbf{J}_{\mathbf{f}}(\mathbf{s}^{(\ell)})_{i,:n_I}\}_{\ell=1}^{|\mathcal{F}_{i,:n_I}|} = \mathbb{R}_{\mathcal{F}_{i,:n_I}}^{n_I}$ and $[\mathbf{J}_{\mathbf{f}}(\mathbf{s}^{(\ell)})\mathrm{T}]_{i,:n_I} \in \mathbb{R}_{\hat{\mathcal{F}}_{i,:n_I}}^{n_I}$.*

*ii. (Structural Sparsity) For all $k \in \{1, \dots, n_I\}$, there exists $\mathcal{C}_k$ s.t. $\bigcap_{i \in \mathcal{C}_k} \mathcal{F}_{i,:n_I} = \{k\}$.*

*Then $\mathbf{s}_I$ is identifiable up to an element-wise invertible transformation and a permutation.*

*Proof.* Our goal here is to show that $\hat{\mathbf{s}}_I$ is a composition of a permutation and a component-wise invertible transformation of sources in $\mathbf{s}_I$. By using chain rule repeatedly, we have

$$\begin{aligned} \mathbf{J}_{\hat{\mathbf{f}}}(\hat{\mathbf{s}}) &= \mathbf{J}_{\mathbf{f} \circ \mathbf{h}^{-1}}(\hat{\mathbf{s}}) \\ &= \mathbf{J}_{\mathbf{f}}(\mathbf{h}^{-1}(\hat{\mathbf{s}})) \mathbf{J}_{\mathbf{h}^{-1}}(\hat{\mathbf{s}}) \\ &= \mathbf{J}_{\mathbf{f}}(\mathbf{s}) \mathbf{J}_{\mathbf{h}^{-1}}(\hat{\mathbf{s}}). \end{aligned} \tag{39}$$

As $\mathbf{h}^{-1}(\cdot)$ is smooth, its Jacobian can be written as:

$$\mathbf{J}_{\mathbf{h}^{-1}}(\hat{\mathbf{s}}) = \left[ \begin{array}{c|c} \mathbf{A} := \frac{\partial \mathbf{s}_I}{\partial \hat{\mathbf{s}}_I} & \mathbf{B} := \frac{\partial \mathbf{s}_I}{\partial \hat{\mathbf{s}}_D} \\ \hline \mathbf{C} := \frac{\partial \mathbf{s}_D}{\partial \hat{\mathbf{s}}_I} & \mathbf{D} := \frac{\partial \mathbf{s}_D}{\partial \hat{\mathbf{s}}_D} \end{array} \right]. \tag{40}$$

In the proof of Thm. 4.1, we have shown that all entries in the submatrix $\mathbf{C}$ are zero. Then we have

$$\begin{aligned} \mathbf{J}_{\hat{\mathbf{f}}}(\hat{\mathbf{s}})_{:,:n_I} &= \mathbf{J}_{\mathbf{f}}(\mathbf{s})\mathbf{J}_{\mathbf{h}^{-1}}(\hat{\mathbf{s}})_{:,:n_I} \\ &\overset{(\star)}{=} \mathbf{J}_{\mathbf{f}}(\mathbf{s})_{:,:n_I}\mathbf{J}_{\mathbf{h}^{-1}}(\hat{\mathbf{s}})_{:n_I,:n_I}, \end{aligned} \tag{41}$$

where Eq. $(\star)$ is directly from the result that all entries in $\mathbf{C}$, i.e., those in $\mathbf{J}_{\mathbf{h}^{-1}}(\hat{s})_{n_I+1:,:n_I}$, are zero.

Moreover, in the proof of Thm. 4.1, we have also shown that all entries in the submatrix $\mathbf{B}$ are zero. Let $\mathbf{D}(\mathbf{s})$ represent a diagonal matrix and $\mathbf{P}$ represent a permutation matrix. Thus, our goal is equivalent to show that $\mathbf{J}_{\mathbf{h}}(\mathbf{s})_{:n_I,:n_I} = \mathbf{P}_I\mathbf{D}_I(\mathbf{s})$ or $\mathbf{J}_{\mathbf{h}^{-1}}(\hat{\mathbf{s}})_{:n_I,:n_I} = \mathbf{J}_{\mathbf{h}^{-1}}(\hat{\mathbf{s}})_{:n_I,:n_I} = \mathbf{J}_{\mathbf{h}}(\mathbf{s})_{:n_I,:n_I}^{-1} = \mathbf{D}_I(\mathbf{s})^{-1}\mathbf{P}_I^{-1}$. Then we need to prove that

$$\mathbf{J}_{\hat{\mathbf{f}}}(\hat{\mathbf{s}})_{:,:n_I} = \mathbf{J}_{\mathbf{f}}(\mathbf{s})_{:,:n_I}\mathbf{D}_I(\mathbf{s})^{-1}\mathbf{P}_I^{-1}. \tag{42}$$

Additionally, we have

$$\mathbf{J}_{\hat{\mathbf{f}}}(\hat{\mathbf{s}})_{:,:n_I} = \mathbf{J}_{\mathbf{f}}(\mathbf{s})_{:,:n_I}\mathbf{T}(\mathbf{s}), \tag{43}$$

where $\mathbf{T}(\mathbf{s}) \in \mathbb{R}^{n_I \times n_I}$ is a square matrix. Note that we have denoted $\mathcal{F}$ as the support of $\mathbf{J}_{\mathbf{f}}(\mathbf{s})$, $\hat{\mathcal{F}}$ as the support of $\mathbf{J}_{\hat{\mathbf{f}}}(\hat{\mathbf{s}})$ and $\mathcal{T}$ as the support of $\mathbf{T}(\mathbf{s})$. Besides, we have also denoted $\mathrm{T}$ as a matrix with the same support of $\mathcal{T}$. According to Assumption i, we have

$$\mathrm{span}\{\mathbf{J}_{\mathbf{f}}(\mathbf{s}^{(\ell)})_{i,:n_I}\}_{\ell=1}^{|\mathcal{F}_{i,:n_I}|} = \mathbb{R}^n_{\mathcal{F}_{i,:n_I}}. \tag{44}$$

Since $\{\mathbf{J}_{\mathbf{f}}(\mathbf{s}^{(\ell)})_{i,:n_I}\}_{\ell=1}^{|\mathcal{F}_{i,:n_I}|}$ forms a basis of $\mathbb{R}^{n_I}_{\mathcal{F}_{i,:n_I}}$, for any $j_0 \in \mathcal{F}_{i,:n_I}$, we are able to rewrite the one-hot vector $e_{j_0} \in \mathbb{R}^{n_I}_{\mathcal{F}_{i,:n_I}}$ as

$$e_{j_0} = \sum_{\ell \in \mathcal{F}_{i,:n_I}} \alpha_\ell \mathbf{J}_{\mathbf{f}}(\mathbf{s}^{(\ell)})_{i,:n_I}, \tag{45}$$

where $\alpha_\ell$ is the corresponding coefficient. Then

$$\mathrm{T}_{j_0,:} = e_{j_0}\mathrm{T} = \sum_{\ell \in \mathcal{F}_{i,:n_I}} \alpha_\ell \mathbf{J}_{\mathbf{f}}(\mathbf{s}^{(\ell)})_{i,:n_I}\mathrm{T}, \tag{46}$$

According to Assumption i, each term in the above summation belongs to the space $\mathbb{R}^{n_I}_{\hat{\mathcal{F}}_{i,:n_I}}$. Therefore, $\mathrm{T}_{j_0,:}$ itself resides in $\mathbb{R}^{n_I}_{\hat{\mathcal{F}}_{i,:n_I}}$, i.e., $\mathrm{T}_{j_0,:} \in \mathbb{R}^{n_I}_{\hat{\mathcal{F}}_{i,:n_I}}$. Thus

$$\forall j \in \mathcal{F}_{i,:n_I}, \ \mathrm{T}_{j,:} \in \mathbb{R}^{n_I}_{\hat{\mathcal{F}}_{i,:n_I}}. \tag{47}$$

Then the connections between these supports can be established according to Defn. 2.3

$$\forall (i,j) \in \mathcal{F}_{:,:n_I}, \{i\} \times \mathcal{T}_{j,:} \subset \hat{\mathcal{F}}_{:,:n_I}. \tag{48}$$

Since $\mathbf{J}_{\mathbf{f}}(\mathbf{s}^{(\ell)})_{:,:n_I}$ and $\mathbf{J}_{\hat{\mathbf{f}}}(\hat{\mathbf{s}}^{(\ell)})_{:,:n_I}$ have full column rank $n_I$, $\mathbf{T}(\mathbf{s}^{(\ell)})$ must have a non-zero determinant. Otherwise, it would follow that the rank of $\mathbf{T}(\mathbf{s}^{(\ell)})$ is less than $n_I$, which would imply a contradiction that $\mathbf{J}_{\hat{\mathbf{f}}}(\hat{\mathbf{s}}^{(\ell)})_{:,:n_I} = \mathbf{J}_{\mathbf{f}}(\mathbf{s}^{(\ell)})_{:,:n_I}\mathbf{T}(\mathbf{s}^{(\ell)})$ has a column rank less than $n_I$.

The determinant of the matrix $\mathbf{T}(\mathbf{s}^{(\ell)})$ can be represented as its Leibniz formula as

$$\det(\mathbf{T}(\mathbf{s}^{(\ell)})) = \sum_{\sigma \in \mathcal{S}_{n_I}} \left( \mathrm{sgn}(\sigma) \prod_{i=1}^{n_I} \mathbf{T}(\mathbf{s}^{(\ell)})_{i,\sigma(i)} \right) \neq 0, \tag{49}$$

where $\mathcal{S}_{n_I}$ is the set of $n_I$-permutations. Therefore, there is at least one term in the sum that is non-zero, i.e.,

$$\exists \sigma \in \mathcal{S}_{n_I}, \ \forall i \in \{1, \ldots, n_I\}, \ \text{sgn}(\sigma) \prod_{i=1}^{n_I} \mathbf{T}(\mathbf{s}^{(\ell)})_{i,\sigma(i)} \neq 0, \tag{50}$$

which is equivalent to

$$\exists \sigma \in \mathcal{S}_{n_I}, \ \forall i \in \{1, \ldots, n_I\}, \ \mathbf{T}(\mathbf{s}^{(\ell)})_{i,\sigma(i)} \neq 0. \tag{51}$$

Then we can conclude that this $\sigma$ must present in the support of $\mathbf{T}(\mathbf{s})$ since $\mathbf{s}^{(\ell)} \in \mathbf{s}$. Therefore, it follows that

$$\forall j \in \{1, \ldots, n_I\}, \ \sigma(j) \in \mathcal{T}_{j,:}. \tag{52}$$

Together with Eq. (48), we have

$$\forall (i, j) \in \mathcal{F}_{:,:n_I}, (i, \sigma(j)) \in \{i\} \times \mathcal{T}_{j,:} \subset \hat{\mathcal{F}}_{:,:n_I}. \tag{53}$$

Denote

$$\sigma(\mathcal{F}_{:,:n_I}) = \{(i, \sigma(j)) \mid (i, j) \in \mathcal{F}_{:,:n_I}\}. \tag{54}$$

Then we have

$$\sigma(\mathcal{F}_{:,:n_I}) \subset \hat{\mathcal{F}}_{:,:n_I}. \tag{55}$$

Because of the sparsity regularization on the estimated Jacobian, we further have

$$|\hat{\mathcal{F}}_{:,:n_I}| \leq |\mathcal{F}_{:,:n_I}| = |\sigma(\mathcal{F}_{:,:n_I})|. \tag{56}$$

Combined with Eq. (55), we have

$$\sigma(\mathcal{F}_{:,:n_I}) = \hat{\mathcal{F}}_{:,:n_I}. \tag{57}$$

Suppose $\mathbf{T}(\mathbf{s}) \neq \mathbf{D}_I(\mathbf{s})^{-1} \mathbf{P}_I^{-1}$, then

$$\exists j_1 \neq j_2, \ \mathcal{T}_{j_1,:} \cap \mathcal{T}_{j_2,:} \neq \emptyset. \tag{58}$$

Additionally, consider $j_3 \in \{1, \ldots, n_I\}$ for which

$$\sigma(j_3) \in \mathcal{T}_{j_1,:} \cap \mathcal{T}_{j_2,:}. \tag{59}$$

Since $j_1 \neq j_2$, we can assume $j_3 \neq j_1$ without loss of generality. Based on Assumption ii, there exists $\mathcal{C}_{j_1} \ni j_1$ such that $\bigcap_{i \in \mathcal{C}_{j_1}} \mathcal{F}_{i,:n_I} = \{j_1\}$. Because

$$j_3 \notin \{j_1\} = \bigcap_{i \in \mathcal{C}_{j_1}} \mathcal{F}_{i,:n_I}, \tag{60}$$

there must exists $i_3 \in \mathcal{C}_{j_1}$ such that

$$j_3 \notin \mathcal{F}_{i_3,:n_I}. \tag{61}$$

Since $j_1 \in \mathcal{F}_{i_3,:n_I}$, it follows that $(i_3, j_1) \in \mathcal{F}_{:,:n_I}$. Therefore, according to Eq. (48), we have

$$\{i_3\} \times \mathcal{T}_{j_1,:} \subset \hat{\mathcal{F}}_{:,:n_I}. \tag{62}$$

Notice that $\sigma(j_3) \in \mathcal{T}_{j_1,:} \cap \mathcal{T}_{j_2,:}$ implies

$$(i_3, \sigma(j_3)) \in \{i_3\} \times \mathcal{T}_{j_1,:}. \tag{63}$$

Then by Eqs. (62) and (63), we have

$$(i_3, \sigma(j_3)) \in \hat{\mathcal{F}}_{:,:n_I}. \tag{64}$$

This further implies $(i_3, j_3) \in \mathcal{F}_{:,:n_I}$ by Eqs. (54) and (57), which contradicts Eq. (61). Thus, we have proven by contradiction that $\mathbf{T}(\mathbf{s}) = \mathbf{D}_I(\mathbf{s})^{-1} \mathbf{P}_I^{-1}$. By replacing $\mathbf{T}(\mathbf{s})$ with $\mathbf{D}_I(\mathbf{s})^{-1} \mathbf{P}_I^{-1}$ in Eq. (43), we obtain Eq. (42), which is the goal. □

## A.4 Proof of Theorem 4.3

**Theorem 4.3.** *Let the observed data be a large enough sample generated from an undercomplete nonlinear ICA model as defined in Eqs. (2) and (4). Suppose the following assumptions hold:*

i. *For each $i \in \{1, \ldots, n_I\}$, there exist $\{\mathbf{s}^{(\ell)}\}_{\ell=1}^{|\mathcal{F}_{i,:n_I}|}$ and a matrix $\mathrm{T} \in \mathcal{T}$ s.t. $\mathrm{span}\{\mathbf{J_f}(\mathbf{s}^{(\ell)})_{i,:n_I}\}_{\ell=1}^{|\mathcal{F}_{i,:n_I}|} = \mathbb{R}^{n_I}_{\mathcal{F}_{i,:n_I}}$ and $\left[\mathbf{J_f}(\mathbf{s}^{(\ell)})\mathrm{T}\right]_{i,:n_I} \in \mathbb{R}^{n_I}_{\hat{\mathcal{F}}_{i,:n_I}}$.*

ii. *There exist $2n_D + 1$ values of $\mathbf{u}$, i.e., $\mathbf{u}_i$ with $i \in \{0, 1, \ldots, 2n_D\}$, s.t. the $2n_D$ vectors $\mathbf{w}(\mathbf{s}_D, \mathbf{u}_i) - \mathbf{w}(\mathbf{s}_D, \mathbf{u}_0)$ with $i \in \{1, \ldots, 2n_D\}$ are linearly independent, where vector $\mathbf{w}(\mathbf{s}_D, \mathbf{u}_i)$ is defined as follows:*

$$\mathbf{w}(\mathbf{s}_D, \mathbf{u}_i) = \left(\mathbf{v}(\mathbf{s}_{c_1}, \mathbf{u}_i), \cdots, \mathbf{v}(\mathbf{s}_{c_d}, \mathbf{u}_i), \mathbf{v}'(\mathbf{s}_{c_1}, \mathbf{u}_i), \cdots, \mathbf{v}'(\mathbf{s}_{c_d}, \mathbf{u}_i)\right),$$

*where*

$$\mathbf{v}(\mathbf{s}_{c_j}, \mathbf{u}_i) = \left(\frac{\partial \log p(\mathbf{s}_{c_j}|\mathbf{u}_i)}{\partial s_{c_j^{(l)}}}, \cdots, \frac{\partial \log p(\mathbf{s}_{c_j}|\mathbf{u}_i)}{\partial s_{c_j^{(h)}}}\right),$$

$$\mathbf{v}'(\mathbf{s}_{c_j}, \mathbf{u}_i) = \left(\frac{\partial^2 \log p(\mathbf{s}_{c_j}|\mathbf{u}_i)}{(\partial s_{c_j^{(l)}})^2}, \cdots, \frac{\partial^2 \log p(\mathbf{s}_{c_j}|\mathbf{u}_i)}{(\partial s_{c_j^{(h)}})^2}\right).$$

iii. *There exist $\mathbf{u}_1, \mathbf{u}_2 \in \mathbf{u}$, s.t., for any set $A_\mathbf{s} \subseteq \mathcal{S}$ with nonzero probability measure and cannot be expressed as $B_{\mathbf{s}_I} \times \mathcal{S}_D$ for any $B_{\mathbf{s}_I} \subset \mathcal{S}_I$, we have*

$$\int_{\mathbf{s} \in A_\mathbf{s}} p_{\mathbf{s}|\mathbf{u}}(\mathbf{s} \mid \mathbf{u}_1) d\mathbf{s} \neq \int_{\mathbf{s} \in A_\mathbf{s}} p_{\mathbf{s}|\mathbf{u}}(\mathbf{s} \mid \mathbf{u}_2) d\mathbf{s}.$$

iv. *(Structural Sparsity) For all $k \in \{1, \ldots, n_I\}$, there exists $\mathcal{C}_k$ s.t. $\bigcap_{i \in \mathcal{C}_k} \mathcal{F}_{i,:n_I} = \{k\}$.*

*Then $\mathbf{s}_I$ is identifiable up to an element-wise invertible transformation and a permutation, and $\mathbf{s}_D$ is identifiable up to a subspace-wise invertible transformation and a subspace-wise permutation.*

*Proof.* Let $h : \mathbf{s} \to \hat{\mathbf{s}}$ denotes the transformation between the true and estimated sources. By using chain rule repeatedly, we have

$$\begin{aligned}\mathbf{J_f}(\mathbf{s}) &= \mathbf{J_{\hat{f}\circ h}}(\mathbf{s}) \\ &= \mathbf{J_{\hat{f}}}(\hat{\mathbf{s}})\mathbf{J_h}(\mathbf{s}).\end{aligned} \tag{65}$$

Because $\mathbf{J_{\hat{f}}}(\hat{\mathbf{s}})$ and $\mathbf{J_f}(\mathbf{s})$ have full column rank, $\mathbf{J_h}(\mathbf{s})$ must be invertible and have a non-zero determinant. Otherwise, one of them would not be of full column rank, which leads to a contradiction.

Applying the change of variable rule, we have

$$p_{\mathbf{s}|\mathbf{u}}(\mathbf{s}|\mathbf{u})|\det(\mathbf{J_{h^{-1}}}(\hat{\mathbf{s}}))| = p_{\hat{\mathbf{s}}}(\hat{\mathbf{s}}|\mathbf{u}). \tag{66}$$

Taking the logarithm on both sides yields

$$\log p_{\mathbf{s}|\mathbf{u}}(\mathbf{s}|\mathbf{u}) + \log|\det(\mathbf{J_{h^{-1}}}(\hat{\mathbf{s}}))| = \log p_{\hat{\mathbf{s}}|\mathbf{u}}(\hat{\mathbf{s}}|\mathbf{u}). \tag{67}$$

Note that according to the model defined in Eqs. (2) and (3), the joint densities can be factorized as

$$\begin{aligned}p_{\mathbf{s}|\mathbf{u}}(\mathbf{s}|\mathbf{u}) &= \prod_{i=1}^{n_I} p_{s_i}(s_i) \prod_{j=c_1}^{c_d} p_{\mathbf{s}_j|\mathbf{u}}(\mathbf{s}_j|\mathbf{u}), \\ p_{\hat{\mathbf{s}}|\mathbf{u}}(\hat{\mathbf{s}}|\mathbf{u}) &= \prod_{i=1}^{n_I} p_{\hat{s}_i}(\hat{s}_i) \prod_{j=c_1}^{c_d} p_{\hat{\mathbf{s}}_j|\mathbf{u}}(\hat{\mathbf{s}}_j|\mathbf{u}).\end{aligned} \tag{68}$$

Together with Eq. (67), we have

$$\begin{aligned}&\sum_i^{n_I} \log p_{s_i}(s_i) + \sum_{j=c_1}^{c_d} \log p_{\mathbf{s}_j|\mathbf{u}}(\mathbf{s}_j|\mathbf{u}) + \log|\det(\mathbf{J_{h^{-1}}}(\hat{\mathbf{s}}))| \\ &= \sum_i^{n_I} \log p_{\hat{s}_i}(\hat{s}_i) + \sum_{j=c_1}^{c_d} \log p_{\hat{\mathbf{s}}_j|\mathbf{u}}(\hat{\mathbf{s}}_j|\mathbf{u}).\end{aligned} \tag{69}$$

Therefore, for $\mathbf{u} = \mathbf{u}_0, \ldots, \mathbf{u}_{2n_D}$, we have $2n_D + 1$ such equations. Subtracting each equation corresponding to $\mathbf{u}_1, \ldots, \mathbf{u}_{2n_D}$ with the equation corresponding to $\mathbf{u}_0$ results in $2n_D$ equations:

$$
\sum_{i=c_1}^{c_d} \left( \log p_{\mathbf{s}_i|\mathbf{u}_j}(\mathbf{s}_i|\mathbf{u}_j) - \log p_{\mathbf{s}_i|\mathbf{u}_0}(\mathbf{s}_i|\mathbf{u}_0) \right)
$$

$$
= \sum_{i=c_1}^{c_d} \left( \log p_{\hat{\mathbf{s}}_i|\mathbf{u}_j}(\hat{\mathbf{s}}_i|\mathbf{u}_j) - \log p_{\hat{\mathbf{s}}_i|\mathbf{u}_0}(\hat{\mathbf{s}}_i|\mathbf{u}_0) \right). \tag{70}
$$

Then we take the derivatives of both sides of Eq. (70) w.r.t. $\hat{s}_k$ and $\hat{s}_v$ where $k, v \in \{1, \ldots, n\}$ and $k \neq v$. Besides, if both $k > n_I$ and $v > n_I$, then they are not indices of the same subspace. It is clear that the RHS of Eq. (70) equals to zero. For the $i$-th term of the summation on the LHS, we have the following equation after taking the derivatives:

$$
\sum_{l=i^{(l)}}^{i^{(h)}} \left( \left( \frac{\partial^2 \log p_{\mathbf{s}_i|\mathbf{u}_j}(\mathbf{s}_i|\mathbf{u}_j)}{(\partial s_l)^2} - \frac{\partial^2 \log p_{\mathbf{s}_i|\mathbf{u}_0}(\mathbf{s}_i|\mathbf{u}_0)}{(\partial s_l)^2} \right) \cdot \frac{\partial s_l}{\partial \hat{s}_k} \frac{\partial s_l}{\partial \hat{s}_v} \right.
$$

$$
\left. + \left( \frac{\partial \log p_{\mathbf{s}_i|\mathbf{u}_j}(\mathbf{s}_i|\mathbf{u}_j)}{\partial s_l} - \frac{\partial \log p_{\mathbf{s}_i|\mathbf{u}_0}(\mathbf{s}_i|\mathbf{u}_0)}{\partial s_l} \right) \cdot \frac{\partial^2 s_l}{\partial \hat{s}_k \partial \hat{s}_v} \right) = 0, \tag{71}
$$

where $i_l$ and $i_h$ corresponds to the minimum and maximum indices of elements in $\mathbf{s}_i = (s_{i_l}, \ldots, s_{i_h})$. By iterating $i$ from $c_1$ to $c_d$, we are also iterating $l$ from $n_I + 1$ to $n$. Thus by considering all those equations as well as iterating $j$ in $\mathbf{u}_j$ from 0 to $2n_D$, we have a linear system with a $2n_D \times 2n_D$ coefficient matrix.

Then, according to Assumption ii, the coefficient matrix of the linear system has full rank. Thus, the only solution of Eq. (71) is $\frac{\partial s_l}{\partial \hat{s}_k} \frac{\partial s_l}{\partial \hat{s}_v} = 0$ and $\frac{\partial^2 s_l}{\partial \hat{s}_k \partial \hat{s}_v} = 0$.

Note that $k \neq v$ and, if both $k > n_I$ and $v > n_I$, they are not indices of sources in the same subspace. Besides, $\frac{\partial s_l}{\partial \hat{s}_k} \frac{\partial s_l}{\partial \hat{s}_v} = 0$ indicates that it is impossible for both $\frac{\partial s_l}{\partial \hat{s}_k}$ and $\frac{\partial s_l}{\partial \hat{s}_v}$ to be non-zero. Furthermore, because $h$ is invertible, they cannot be both zero, indicating that it is either $\frac{\partial s_l}{\partial \hat{s}_k} = 0$ or $\frac{\partial s_l}{\partial \hat{s}_v} = 0$. Then, because $\mathbf{s}_I$ has invariant distribution w.r.t. $\mathbf{u}$, if $\frac{\partial s_l}{\partial \hat{s}_k} \neq 0$ (i.e., $\frac{\partial s_l}{\partial \hat{s}_v} = 0$), then $\hat{k} \notin \{1, \ldots, n_I\}$. Otherwise, $s_l$ will also be invariant, which is a contradiction. Therefore, $\hat{k}$ can only be the index of an estimated source from one independent subspace, which, together with the invertibility, leads to the conclusion that $\mathbf{s}_D$ is a composition of an invertible subspace-wise transformation and a subspace-wise permutation of $\hat{\mathbf{s}_D}$. So it is the mapping from $\hat{\mathbf{s}}_D$ to $\mathbf{s}_D$ since the subspace-wise transformation is invertible and the inverse of a block-wise permutation matrix is still a block-wise invertible matrix.

Now we have shown the identifiability result for $\mathbf{s}_D = (s_{n_I+1}, \ldots, s_n)$, then we need to show that for the remaining sources $\mathbf{s}_I = (s_1, \ldots, s_{n_I})$.

By using chain rule repeatedly, we have

$$
\begin{aligned}
\mathbf{J}_{\hat{\mathbf{f}}}(\hat{\mathbf{s}}) &= \mathbf{J}_{\mathbf{f} \circ \mathbf{h}^{-1}}(\hat{\mathbf{s}}) \\
&= \mathbf{J}_{\mathbf{f}}(\mathbf{h}^{-1}(\hat{\mathbf{s}})) \mathbf{J}_{\mathbf{h}^{-1}}(\hat{\mathbf{s}}) \\
&= \mathbf{J}_{\mathbf{f}}(\mathbf{s}) \mathbf{J}_{\mathbf{h}^{-1}}(\hat{\mathbf{s}}).
\end{aligned} \tag{72}
$$

Since we have shown that, for every $l \in \{n_I + 1, \ldots, n\}$ and $k \in \{1, \ldots, n_I\}$, $\frac{\partial s_l}{\partial \hat{s}_k} = 0$, all entries of $\mathbf{J}_{\mathbf{h}^{-1}}(\hat{\mathbf{s}})_{n_I+1:n,:n_I}$ must be zero.

Then we have

$$
\begin{aligned}
\mathbf{J}_{\hat{\mathbf{f}}}(\hat{\mathbf{s}})_{:,:n_I} &= \mathbf{J}_{\mathbf{f}}(\mathbf{s}) \mathbf{J}_{\mathbf{h}^{-1}}(\hat{\mathbf{s}})_{:,:n_I} \\
&\overset{(\star)}{=} \mathbf{J}_{\mathbf{f}}(\mathbf{s})_{:,:n_I} \mathbf{J}_{\mathbf{h}^{-1}}(\hat{\mathbf{s}})_{:n_I,:n_I},
\end{aligned} \tag{73}
$$

where Eq. $(\star)$ is directly from the result that all entries in $\mathbf{J}_{\mathbf{h}^{-1}}(\mathbf{s})_{n_I+1:,:n_I}$ are zero.

Based on the proof of Theorem 4.2 in Kong et al. (2022) and Assumption iii, particularly, steps 1, 2, and 3, one can show that $\hat{\mathbf{s}}_I$ does not depend on $\mathbf{s}_D$. Let $\mathbf{D}(\mathbf{s})$ represents a diagonal matrix and $\mathbf{P}$

represent a permutation matrix. Thus, our goal is equivalent to show that $\mathbf{J_h}(\mathbf{s})_{:n_I,:n_I} = \mathbf{P}_I \mathbf{D}_I(\mathbf{s})$ or $\mathbf{J_{h^{-1}}}(\hat{\mathbf{s}})_{:n_I,:n_I} = \mathbf{J_{h^{-1}}}(\hat{\mathbf{s}})_{:n_I,:n_I} = \mathbf{J_h}(\mathbf{s})^{-1}_{:n_I,:n_I} = \mathbf{D}_I(\mathbf{s})^{-1}\mathbf{P}_I^{-1}$. Then we need to prove that

$$\mathbf{J_{\hat{f}}}(\hat{\mathbf{s}})_{:,:n_I} = \mathbf{J_f}(\mathbf{s})_{:,:n_I} \mathbf{D}_I(\mathbf{s})^{-1}\mathbf{P}_I^{-1}. \tag{74}$$

Additionally, we have

$$\mathbf{J_{\hat{f}}}(\hat{\mathbf{s}})_{:,:n_I} = \mathbf{J_f}(\mathbf{s})_{:,:n_I}\mathbf{T}(\mathbf{s}), \tag{75}$$

where $\mathbf{T}(\mathbf{s}) \in \mathbb{R}^{n_I \times n_I}$ is a square matrix. Note that we have denoted $\mathcal{F}$ as the support of $\mathbf{J_f}(\mathbf{s})$, $\hat{\mathcal{F}}$ as the support of $\mathbf{J_{\hat{f}}}(\hat{\mathbf{s}})$ and $\mathcal{T}$ as the support of $\mathbf{T}(\mathbf{s})$. Besides, we have also denoted T as a matrix with the same support of $\mathcal{T}$. According to Assumption i, we have

$$\text{span}\{\mathbf{J_f}(\mathbf{s}^{(\ell)})_{i,:n_I}\}_{\ell=1}^{|\mathcal{F}_{i,:n_I}|} = \mathbb{R}^n_{\mathcal{F}_{i,:n_I}}. \tag{76}$$

Since $\{\mathbf{J_f}(\mathbf{s}^{(\ell)})_{i,:n_I}\}_{\ell=1}^{|\mathcal{F}_{i,:n_I}|}$ forms a basis of $\mathbb{R}^{n_I}_{\mathcal{F}_{i,:n_I}}$, for any $j_0 \in \mathcal{F}_{i,:n_I}$, we are able to rewrite the one-hot vector $e_{j_0} \in \mathbb{R}^{n_I}_{\mathcal{F}_{i,:n_I}}$ as

$$e_{j_0} = \sum_{\ell \in \mathcal{F}_{i,:n_I}} \alpha_\ell \mathbf{J_f}(\mathbf{s}^{(\ell)})_{i,:n_I}, \tag{77}$$

where $\alpha_\ell$ is the corresponding coefficient. Then

$$\mathrm{T}_{j_0,:} = e_{j_0}\mathrm{T} = \sum_{\ell \in \mathcal{F}_{i,:n_I}} \alpha_\ell \mathbf{J_f}(\mathbf{s}^{(\ell)})_{i,:n_I}\mathrm{T} \in \mathbb{R}^{n_I}_{\hat{\mathcal{F}}_{i,:n_I}}, \tag{78}$$

where the final "$\in$" follows from Assumption i that each element in the summation belongs to $\mathbb{R}^{n_I}_{\hat{\mathcal{F}}_{i,:n_I}}$. Thus

$$\forall j \in \mathcal{F}_{i,:n_I}, \ \mathrm{T}_{j,:} \in \mathbb{R}^{n_I}_{\hat{\mathcal{F}}_{i,:n_I}}. \tag{79}$$

Then the connections between these supports can be established according to Defn. 2.3

$$\forall (i,j) \in \mathcal{F}_{:,:n_I}, \{i\} \times \mathcal{T}_{j,:} \subset \hat{\mathcal{F}}_{:,:n_I}. \tag{80}$$

Since $\mathbf{J_f}(\mathbf{s}^{(\ell)})_{:,:n_I}$ and $\mathbf{J_{\hat{f}}}(\hat{\mathbf{s}}^{(\ell)})_{:,:n_I}$ have full column rank $n_I$, $\mathbf{T}(\mathbf{s}^{(\ell)})$ must have a non-zero determinant. Otherwise, it would follow that the rank of $\mathbf{T}(\mathbf{s}^{(\ell)})$ is less than $n_I$, which would imply a contradiction that $\mathbf{J_{\hat{f}}}(\hat{\mathbf{s}}^{(\ell)})_{:,:n_I} = \mathbf{J_f}(\mathbf{s}^{(\ell)})_{:,:n_I}\mathbf{T}(\mathbf{s}^{(\ell)})$ has a column rank less than $n_I$.

The determinant of the matrix $\mathbf{T}(\mathbf{s}^{(\ell)})$ can be represented as its Leibniz formula as

$$\det(\mathbf{T}(\mathbf{s}^{(\ell)})) = \sum_{\sigma \in \mathcal{S}_{n_I}} \left( \text{sgn}(\sigma) \prod_{i=1}^{n_I} \mathbf{T}(\mathbf{s}^{(\ell)})_{i,\sigma(i)} \right) \neq 0, \tag{81}$$

where $\mathcal{S}_{n_I}$ is the set of $n_I$-permutations. Thus, there is at least one non-zero term in the sum, i.e.,

$$\exists \sigma \in \mathcal{S}_{n_I}, \ \forall i \in \{1,\ldots,n_I\}, \ \text{sgn}(\sigma) \prod_{i=1}^{n_I} \mathbf{T}(\mathbf{s}^{(\ell)})_{i,\sigma(i)} \neq 0, \tag{82}$$

which is equivalent to

$$\exists \sigma \in \mathcal{S}_{n_I}, \ \forall i \in \{1,\ldots,n_I\}, \ \mathbf{T}(\mathbf{s}^{(\ell)})_{i,\sigma(i)} \neq 0. \tag{83}$$

Then we can see that this $\sigma$ must present in the support of $\mathbf{T}(\mathbf{s})$ since $\mathbf{s}^{(\ell)} \in \mathbf{s}$. It follows that

$$\forall j \in \{1,\ldots,n_I\}, \ \sigma(j) \in \mathcal{T}_{j,:}. \tag{84}$$

Together with Eq. (80), we have

$$\forall (i,j) \in \mathcal{F}_{:,:n_I}, (i,\sigma(j)) \in \{i\} \times \mathcal{T}_{j,:} \subset \hat{\mathcal{F}}_{:,:n_I}. \tag{85}$$

Denote

$$\sigma(\mathcal{F}_{:,:n_I}) = \{(i,\sigma(j)) \mid (i,j) \in \mathcal{F}_{:,:n_I}\}. \tag{86}$$

Then we have

$$\sigma(\mathcal{F}_{:,:n_I}) \subset \hat{\mathcal{F}}_{:,:n_I}. \tag{87}$$

Because of the sparsity regularization on the estimated Jacobian, we further have

$$|\hat{\mathcal{F}}_{:,:n_I}| \le |\mathcal{F}_{:,:n_I}| = |\sigma(\mathcal{F}_{:,:n_I})|. \tag{88}$$

Together with Eq. (87), we have

$$\sigma(\mathcal{F}_{:,:n_I}) = \hat{\mathcal{F}}_{:,:n_I}. \tag{89}$$

Suppose $\mathbf{T}(\mathbf{s}) \ne \mathbf{D}_I(\mathbf{s})^{-1}\mathbf{P}_I^{-1}$, then

$$\exists j_1 \ne j_2, \ \mathcal{T}_{j_1,:} \cap \mathcal{T}_{j_2,:} \ne \emptyset. \tag{90}$$

Additionally, consider $j_3 \in \{1, \dots, n_I\}$ for which

$$\sigma(j_3) \in \mathcal{T}_{j_1,:} \cap \mathcal{T}_{j_2,:}. \tag{91}$$

Since $j_1 \ne j_2$, we can assume $j_3 \ne j_1$ without loss of generality. Based on Assumption iv, there exists $\mathcal{C}_{j_1} \ni j_1$ such that $\bigcap_{i \in \mathcal{C}_{j_1}} \mathcal{F}_{i,:n_I} = \{j_1\}$. Because

$$j_3 \notin \{j_1\} = \bigcap_{i \in \mathcal{C}_{j_1}} \mathcal{F}_{i,:n_I}, \tag{92}$$

there must exists $i_3 \in \mathcal{C}_{j_1}$ such that

$$j_3 \notin \mathcal{F}_{i_3,:n_I}. \tag{93}$$

Since $j_1 \in \mathcal{F}_{i_3,:n_I}$, we have $(i_3, j_1) \in \mathcal{F}_{:,:n_I}$. Therefore, according to Eq. (80), we have

$$\{i_3\} \times \mathcal{T}_{j_1,:} \subset \hat{\mathcal{F}}_{:,:n_I}. \tag{94}$$

Notice that $\sigma(j_3) \in \mathcal{T}_{j_1,:} \cap \mathcal{T}_{j_2,:}$ implies

$$(i_3, \sigma(j_3)) \in \{i_3\} \times \mathcal{T}_{j_1,:}. \tag{95}$$

Then by Eqs. (94) and (95), we have

$$(i_3, \sigma(j_3)) \in \hat{\mathcal{F}}_{:,:n_I}, \tag{96}$$

which implies $(i_3, j_3) \in \mathcal{F}_{:,:n_I}$ by Eqs. (86) and (89), therefore contradicting Eq. (93). Thus, we have proven by contradiction that $\mathbf{T}(\hat{\mathbf{s}}) = \mathbf{D}_I(\mathbf{s})^{-1}\mathbf{P}_I^{-1}$. By replacing $\mathbf{T}(\mathbf{s})$ with $\mathbf{D}_I(\mathbf{s})^{-1}\mathbf{P}_I^{-1}$ in Eq. (75), we obtain Eq. (74), which is the goal.

Therefore, $\mathbf{s}_I$ is identifiable up to a composition of a component-wise invertible transformation and a permutation, and $\mathbf{s}_D$ is identifiable up to a composition of a subspace-wise invertible transformation and a subspace-wise permutation. $\square$

## A.5 Proof of Theorem 4.4

**Theorem 4.4.** *Let the observed data be a large enough sample generated from an undercomplete nonlinear ICA model as defined in Eqs. (2) and (5), suppose the following assumptions hold:*

*i. For each $i \in \{1, \dots, n_I\}$, there exist $\{\mathbf{s}^{(\ell)}\}_{\ell=1}^{|\mathcal{F}_{i,:n_I}|}$ and a matrix $\mathrm{T} \in \mathcal{T}$ s.t. $\mathrm{span}\{\mathbf{J_f}(\mathbf{s}^{(\ell)})_{i,:n_I}\}_{\ell=1}^{|\mathcal{F}_{i,:n_I}|} = \mathbb{R}^{n_I}_{\mathcal{F}_{i,:n_I}}$ and $[\mathbf{J_f}(\mathbf{s}^{(\ell)})\mathrm{T}]_{i,:n_I} \in \mathbb{R}^{n_I}_{\hat{\mathcal{F}}_{i,:n_I}}$.*

*ii. There exist $2n_D + 1$ values of $\mathbf{u}$, i.e., $\mathbf{u}_i$ with $i \in \{0, 1, \dots, 2n_D\}$, s.t. the $2n_D$ vectors $\mathbf{w}(\mathbf{s}_D, \mathbf{u}_i) - \mathbf{w}(\mathbf{s}_D, \mathbf{u}_0)$ with $i \in \{1, \dots, 2n_D\}$ are linearly independent, where vector $\mathbf{w}(\mathbf{s}_D, \mathbf{u})$ is defined as follows:*

$$\mathbf{w}(\mathbf{s}_D, \mathbf{u}_i) = (\mathbf{v}(\mathbf{s}_D, \mathbf{u}_i), \mathbf{v}'(\mathbf{s}_D, \mathbf{u}_i)),$$

*where*

$$\mathbf{v}(\mathbf{s}_D, \mathbf{u}_i) = \left( \frac{\partial \log p(s_{n_I+1}|\mathbf{u}_i)}{\partial s_{n_I+1}}, \cdots, \frac{\partial \log p(s_n|\mathbf{u}_i)}{\partial s_n} \right),$$

$$\mathbf{v}'(\mathbf{s}_D, \mathbf{u}_i) = \left( \frac{\partial^2 \log p(s_{n_I+1}|\mathbf{u}_i)}{(\partial s_{n_I+1})^2}, \cdots, \frac{\partial^2 \log p(s_n|\mathbf{u}_i)}{(\partial s_n)^2} \right).$$

*iii. There exist $\mathbf{u}_1, \mathbf{u}_2 \in \mathbf{u}$, s.t., for any set $A_{\mathbf{s}} \subseteq \mathcal{S}$ with nonzero probability measure and cannot be expressed as $B_{\mathbf{s}_I} \times \mathcal{S}_D$ for any $B_{\mathbf{s}_I} \subset \mathcal{S}_I$, we have*

$$\int_{\mathbf{s} \in A_{\mathbf{s}}} p_{\mathbf{s}|\mathbf{u}} (\mathbf{s} \mid \mathbf{u}_1) \, d\mathbf{s} \neq \int_{\mathbf{s} \in A_{\mathbf{s}}} p_{\mathbf{s}|\mathbf{u}} (\mathbf{s} \mid \mathbf{u}_2) \, d\mathbf{s}.$$

*iv. (Structural Sparsity) For all $k \in \{1, \ldots, n_I\}$, there exists $\mathcal{C}_k$ s.t. $\bigcap_{i \in \mathcal{C}_k} \mathcal{F}_{i,:n_I} = \{k\}$.*

*Then $\mathbf{s}$ is identifiable up to an element-wise invertible transformation and a permutation.*

*Proof.* Let $h : \mathbf{s} \to \hat{\mathbf{s}}$ denotes the transformation between the true and estimated sources. By using chain rule repeatedly, we have

$$\begin{aligned} \mathbf{J}_{\mathbf{f}}(\mathbf{s}) &= \mathbf{J}_{\hat{\mathbf{f}} \circ h}(\mathbf{s}) \\ &= \mathbf{J}_{\hat{\mathbf{f}}}(\hat{\mathbf{s}}) \mathbf{J}_{\mathbf{h}}(\mathbf{s}). \end{aligned} \tag{97}$$

Because $\mathbf{J}_{\hat{\mathbf{f}}}(\hat{\mathbf{s}})$ and $\mathbf{J}_{\mathbf{f}}(\mathbf{s})$ have full column rank, $\mathbf{J}_{\mathbf{h}}(\mathbf{s})$ must be invertible and have a non-zero determinant. Otherwise, one of them would not be of full column rank, which leads to a contradiction.

Applying the change of variable rule, we have

$$p_{\mathbf{s}|\mathbf{u}}(\mathbf{s}|\mathbf{u})| \det(\mathbf{J}_{\mathbf{h}^{-1}}(\hat{\mathbf{s}}))| = p_{\hat{\mathbf{s}}}(\hat{\mathbf{s}}|\mathbf{u}). \tag{98}$$

Taking the logarithm on both sides yields

$$\log p_{\mathbf{s}|\mathbf{u}}(\mathbf{s}|\mathbf{u}) + \log |\det(\mathbf{J}_{\mathbf{h}^{-1}}(\hat{\mathbf{s}}))| = \log p_{\hat{\mathbf{s}}|\mathbf{u}}(\hat{\mathbf{s}}|\mathbf{u}). \tag{99}$$

Note that according to the model defined in Eqs. (2) and (3), the joint densities can be factorized as

$$\begin{aligned} p_{\mathbf{s}|\mathbf{u}}(\mathbf{s}|\mathbf{u}) &= \prod_{i=1}^{n_I} p_{s_i}(s_i) \prod_{j=n_I+1}^{n} p_{s_j|\mathbf{u}}(s_j|\mathbf{u}) = \prod_{i=1}^{n} p_{s_i|\mathbf{u}}(\hat{s}_i|\mathbf{u}), \\ p_{\hat{\mathbf{s}}|\mathbf{u}}(\hat{\mathbf{s}}|\mathbf{u}) &= \prod_{i=1}^{n_I} p_{\hat{s}_i}(\hat{s}_i) \prod_{j=n_I+1}^{n} p_{\hat{s}_j|\mathbf{u}}(\hat{s}_j|\mathbf{u}) = \prod_{i=1}^{n} p_{\hat{s}_i|\mathbf{u}}(\hat{s}_i|\mathbf{u}). \end{aligned} \tag{100}$$

Together with Eq. (99), we have

$$\sum_{i}^{n} \log p_{s_i|\mathbf{u}}(s_i|\mathbf{u}) + \log |\det(\mathbf{J}_{\mathbf{h}^{-1}}(\hat{\mathbf{s}}))| = \sum_{i}^{n} \log p_{\hat{s}_i|\mathbf{u}}(\hat{s}_i|\mathbf{u}). \tag{101}$$

Then we take the derivatives of both sides of Eq. (101) w.r.t. $\hat{s}_k$ and $\hat{s}_v$ where $k, v \in \{1, \ldots, n\}$ and $k \neq q$. For brevity, we first define the following terms:

$$h'_{i,(k)} := \frac{\partial s_i}{\partial \hat{s}_k}, \tag{102}$$

$$h''_{i,(k,v)} := \frac{\partial^2 s_i}{\partial \hat{s}_k \partial \hat{s}_v}, \tag{103}$$

$$\eta'_i(s_i, \mathbf{u}) := \frac{\partial \log p_{s_i|\mathbf{u}}(s_i|\mathbf{u})}{\partial s_i}, \tag{104}$$

$$\eta''_i(s_i, \mathbf{u}) := \frac{\partial^2 \log p_{s_i|\mathbf{u}}(s_i|\mathbf{u})}{(\partial s_i)^2}. \tag{105}$$

Then we have

$$\sum_{i=1}^{n} \left( \eta''_i(s_i, \mathbf{u}) \cdot h'_{i,(k)} h'_{i,(v)} + \eta'_i(s_i, \mathbf{u}) \cdot h''_{i,(k,v)} \right) + \frac{\partial^2 \log |\det(\mathbf{J}_{\mathbf{h}^{-1}}(\hat{\mathbf{s}}))|}{\partial \hat{s}_k \partial \hat{s}_v} = 0. \tag{106}$$

Therefore, for $\mathbf{u} = \mathbf{u}_0, \ldots, \mathbf{u}_{2n_D}$, we have $2n_D + 1$ such equations. Subtracting each equation corresponding to $\mathbf{u}_1, \ldots, \mathbf{u}_{2n_D}$ with the equation corresponding to $\mathbf{u}_0$ results in $2n_D$ equations:

$$\sum_{i=n_c+1}^{n} \left( (\eta''_i(z_i, \mathbf{u}_j) - \eta''_i(z_i, \mathbf{u}_0)) \cdot h'_{i,(k)} h'_{i,(v)} \right. \tag{107}$$

$$\left. + (\eta'_i(z_i, \mathbf{u}_j) - \eta'_i(z_i, \mathbf{u}_0)) \cdot h''_{i,(k,v)} \right) = 0, \tag{108}$$

where $j = 1, \ldots 2n_D$. Note that for $i \in \{1, \ldots, n_I\}$, $\mathbf{s}_i$ does not depend on $\mathbf{u}$. Thus, we have $\eta_i''(z_i, \mathbf{u}_j) = \eta_i''(z_i, \mathbf{u}_{j'})$ and $\eta_i'(z_i, \mathbf{u}_j) = \eta_i'(z_i, \mathbf{u}_{j'}), \forall j, j'$. Hence only sources dependent on $\mathbf{u}$ (i.e., $\mathbf{s}_D$) remain in Eq. (107).

By considering each $j \in \{1, \ldots, 2n_D\}$ for $\mathbf{u}_j$, we have $2n_D$ equations like Eq. (107), which constitute a linear system with a $2n_D \times 2n_D$ coefficient matrix.

According to Assumption ii, the coefficient matrix of the linear system has full rank. Thus, the only solution of Eq. (107) is $h'_{i,(k)}h'_{i,(q)} = 0$ and $h''_{i,(k,v)} = 0$ for $i = n_c + 1, \ldots, n$ and $k, v \in \{1, \ldots, n\}, k \neq v$.

As $\mathbf{h}^{-1}(\cdot)$ is smooth, its Jacobian can be written as:

$$\mathbf{J}_{\mathbf{h}^{-1}}(\hat{\mathbf{s}}) = \left[ \begin{array}{c|c} \mathbf{A} := \frac{\partial \mathbf{s}_I}{\partial \hat{\mathbf{s}}_I} & \mathbf{B} := \frac{\partial \mathbf{s}_I}{\partial \hat{\mathbf{s}}_D} \\ \hline \mathbf{C} := \frac{\partial \mathbf{s}_D}{\partial \hat{\mathbf{s}}_I} & \mathbf{D} := \frac{\partial \mathbf{s}_D}{\partial \hat{\mathbf{s}}_D} \end{array} \right]. \tag{109}$$

Because $h'_{i,(k)}h'_{i,(v)} = 0, k, v \in \{1, \ldots, n\}, k \neq v$, for each $i = n_I + 1, \ldots, n$, there is at most one index $r \in \{1, \ldots, n\}$ s.t. $h'_{i,(r)} \neq 0$. Therefore, there is at most one non-zero entry in each row indexed by $i = n_D + 1, \ldots, n$ in the Jacobian matrix $\mathbf{J}_{\mathbf{h}^{-1}}(\hat{\mathbf{s}})$. Further, the invertibility of $\mathbf{h}^{-1}(\cdot)$ necessitates $\mathbf{J}_{\mathbf{h}^{-1}}$ to be full-rank which implies that there is exactly one non-zero component in each row of sub-matrices $\mathbf{C}$ and $\mathbf{D}$.

Suppose that non-zero component lies in the sub-matrix $\mathbf{C}$, then $\mathbf{s}_D$ is not dependent on $\mathbf{u}$. Thus, the only non-zero component must lie in $\mathbf{D}$. Because $\mathbf{J}_{\mathbf{h}}$ is of full-rank and $\mathbf{C}$ is a zero sub-matrix, $\mathbf{D}$ must have full rank. Hence, $\hat{\mathbf{s}}_D$ must be a composition of a component-wise invertible transformation and a permutation of $\mathbf{s}_D$. Moreover, according to part of the proof of Theorem 4.2 in Kong et al. (2022) (Steps 1, 2, and 3), the submatrix $B$ is zero if Assumption iii holds.

Now we have shown that, for the matrix $\mathbf{J}_{\mathbf{h}^{-1}}$, its submatrix $D$ is a generalized permutation matrix and both submatrices $B$ and $C$ are zero-matrices). Because $\mathbf{h}^{-1}$ is smooth and invertible, the sub-matrices of the corresponding positions of $\mathbf{J}_{\mathbf{h}}$ have the same properties. Then, we need to show the identifiability for the remaining sources $\mathbf{s}_I = (s_1, \ldots, s_{n_I})$, i.e., $\hat{\mathbf{s}}_I$ is a permutation with component-wise invertible transformation of $\mathbf{s}_I$.

Let $\mathbf{D}(\mathbf{s})$ represents a diagonal matrix and $\mathbf{P}$ represent a permutation matrix. Thus, our goal is equivalent to show that $\mathbf{J}_{\mathbf{h}}(\mathbf{s})_{:n_I,:n_I} = \mathbf{P}_I \mathbf{D}_I(\mathbf{s})$ or $\mathbf{J}_{\mathbf{h}^{-1}}(\hat{\mathbf{s}})_{:n_I,:n_I} = \mathbf{J}_{\mathbf{h}^{-1}}(\hat{\mathbf{s}})_{:n_I,:n_I} = \mathbf{J}_{\mathbf{h}}(\mathbf{s})_{:n_I,:n_I}^{-1} = \mathbf{D}_I(\mathbf{s})^{-1}\mathbf{P}_I^{-1}$. By using chain rule repeatedly, we have

$$\begin{aligned} \mathbf{J}_{\hat{\mathbf{f}}}(\hat{\mathbf{s}}) &= \mathbf{J}_{\mathbf{f} \circ \mathbf{h}^{-1}}(\hat{\mathbf{s}}) \\ &= \mathbf{J}_{\mathbf{f}}(\mathbf{h}^{-1}(\hat{\mathbf{s}}))\mathbf{J}_{\mathbf{h}^{-1}}(\hat{\mathbf{s}}) \\ &= \mathbf{J}_{\mathbf{f}}(\mathbf{s})\mathbf{J}_{\mathbf{h}^{-1}}(\hat{\mathbf{s}}). \end{aligned} \tag{110}$$

Then we have

$$\begin{aligned} \mathbf{J}_{\hat{\mathbf{f}}}(\hat{\mathbf{s}})_{:,:n_I} &= \mathbf{J}_{\mathbf{f}}(\mathbf{s})\mathbf{J}_{\mathbf{h}^{-1}}(\hat{\mathbf{s}})_{:,:n_I} \\ &\overset{(\star)}{=} \mathbf{J}_{\mathbf{f}}(\mathbf{s})_{:,:n_I}\mathbf{J}_{\mathbf{h}^{-1}}(\hat{\mathbf{s}})_{:n_I,:n_I}, \end{aligned} \tag{111}$$

where Eq. $(\star)$ is directly from the result that all entries in $\mathbf{C}$, i.e., those in $\mathbf{J}_{\mathbf{h}^{-1}}(\hat{s})_{n_I+1:,:n_I}$, are zero.

Therefore our goal is equivalent to show that

$$\mathbf{J}_{\hat{\mathbf{f}}}(\hat{\mathbf{s}})_{:,:n_I} = \mathbf{J}_{\mathbf{f}}(\mathbf{s})_{:,:n_I}\mathbf{D}_I(\mathbf{s})^{-1}\mathbf{P}_I^{-1}. \tag{112}$$

Additionally, we have

$$\mathbf{J}_{\hat{\mathbf{f}}}(\hat{\mathbf{s}})_{:,:n_I} = \mathbf{J}_{\mathbf{f}}(\mathbf{s})_{:,:n_I}\mathbf{T}(\mathbf{s}), \tag{113}$$

where $\mathbf{T}(\mathbf{s}) \in \mathbb{R}^{n_I \times n_I}$ is a square matrix. Note that we have denoted $\mathcal{F}$ as the support of $\mathbf{J}_{\mathbf{f}}(\mathbf{s})$, $\hat{\mathcal{F}}$ as the support of $\mathbf{J}_{\hat{\mathbf{f}}}(\hat{\mathbf{s}})$ and $\mathcal{T}$ as the support of $\mathbf{T}(\mathbf{s})$. Besides, we have also denoted T as a matrix with the same support of $\mathcal{T}$. According to Assumption i, we have

$$\text{span}\{\mathbf{J}_{\mathbf{f}}(\mathbf{s}^{(\ell)})_{i,:n_I}\}_{\ell=1}^{|\mathcal{F}_{i,:n_I}|} = \mathbb{R}_{\mathcal{F}_{i,:n_I}}^n. \tag{114}$$

Since $\{\mathbf{J_f}(\mathbf{s}^{(\ell)})_{i,:n_I}\}_{\ell=1}^{|\mathcal{F}_{i,:n_I}|}$ forms a basis of $\mathbb{R}^{n_I}_{\mathcal{F}_{i,:n_I}}$, for any $j_0 \in \mathcal{F}_{i,:n_I}$, we are able to rewrite the one-hot vector $e_{j_0} \in \mathbb{R}^{n_I}_{\mathcal{F}_{i,:n_I}}$ as

$$e_{j_0} = \sum_{\ell \in \mathcal{F}_{i,:n_I}} \alpha_\ell \mathbf{J_f}(\mathbf{s}^{(\ell)})_{i,:n_I}, \tag{115}$$

where $\alpha_\ell$ is the corresponding coefficient. Then

$$\mathrm{T}_{j_0,:} = e_{j_0} \mathrm{T} = \sum_{\ell \in \mathcal{F}_{i,:n_I}} \alpha_\ell \mathbf{J_f}(\mathbf{s}^{(\ell)})_{i,:n_I} \mathrm{T} \in \mathbb{R}^{n_I}_{\hat{\mathcal{F}}_{i,:n_I}}, \tag{116}$$

where the final "$\in$" follows from Assumption i that each element in the summation belongs to $\mathbb{R}^{n_I}_{\hat{\mathcal{F}}_{i,:n_I}}$. Thus

$$\forall j \in \mathcal{F}_{i,:n_I}, \ \mathrm{T}_{j,:} \in \mathbb{R}^{n_I}_{\hat{\mathcal{F}}_{i,:n_I}}. \tag{117}$$

Then the connections between these supports can be established according to Defn. 2.3

$$\forall (i,j) \in \mathcal{F}_{:,:n_I}, \{i\} \times \mathcal{T}_{j,:} \subset \hat{\mathcal{F}}_{:,:n_I}. \tag{118}$$

Since $\mathbf{J_f}(\mathbf{s}^{(\ell)})_{:,:n_I}$ and $\mathbf{J_{\hat{f}}}(\hat{\mathbf{s}}^{(\ell)})_{:,:n_I}$ have full column rank $n_I$, $\mathbf{T}(\mathbf{s}^{(\ell)})$ must have a non-zero determinant. Otherwise, it would follow that the rank of $\mathbf{T}(\mathbf{s}^{(\ell)})$ is less than $n_I$, which would imply a contradiction that $\mathbf{J_{\hat{f}}}(\hat{\mathbf{s}}^{(\ell)})_{:,:n_I} = \mathbf{J_f}(\mathbf{s}^{(\ell)})_{:,:n_I} \mathbf{T}(\mathbf{s}^{(\ell)})$ has a column rank less than $n_I$.

The determinant of the matrix $\mathbf{T}(\mathbf{s}^{(\ell)})$ can be represented as its Leibniz formula as

$$\det(\mathbf{T}(\mathbf{s}^{(\ell)})) = \sum_{\sigma \in \mathcal{S}_{n_I}} \left( \mathrm{sgn}(\sigma) \prod_{i=1}^{n_I} \mathbf{T}(\mathbf{s}^{(\ell)})_{i,\sigma(i)} \right) \neq 0, \tag{119}$$

where $\mathcal{S}_{n_I}$ is the set of $n_I$-permutations. Thus, there is at least one term in the sum that is non-zero, i.e.,

$$\exists \sigma \in \mathcal{S}_{n_I}, \ \forall i \in \{1, \ldots, n_I\}, \ \mathrm{sgn}(\sigma) \prod_{i=1}^{n_I} \mathbf{T}(\mathbf{s}^{(\ell)})_{i,\sigma(i)} \neq 0, \tag{120}$$

which is equivalent to

$$\exists \sigma \in \mathcal{S}_{n_I}, \ \forall i \in \{1, \ldots, n_I\}, \ \mathbf{T}(\mathbf{s}^{(\ell)})_{i,\sigma(i)} \neq 0. \tag{121}$$

Then we can conclude that this $\sigma$ is in the support of $\mathbf{T}(\mathbf{s})$ since $\mathbf{s}^{(\ell)} \in \mathbf{s}$. Therefore, it follows that

$$\forall j \in \{1, \ldots, n_I\}, \ \sigma(j) \in \mathcal{T}_{j,:}. \tag{122}$$

Combined with Eq. (118), we have

$$\forall (i,j) \in \mathcal{F}_{:,:n_I}, (i, \sigma(j)) \in \{i\} \times \mathcal{T}_{j,:} \subset \hat{\mathcal{F}}_{:,:n_I}. \tag{123}$$

Denote

$$\sigma(\mathcal{F}_{:,:n_I}) = \{(i, \sigma(j)) \mid (i,j) \in \mathcal{F}_{:,:n_I}\}. \tag{124}$$

Then we have

$$\sigma(\mathcal{F}_{:,:n_I}) \subset \hat{\mathcal{F}}_{:,:n_I}. \tag{125}$$

Because of the sparsity regularization on the estimated Jacobian, we further have

$$|\hat{\mathcal{F}}_{:,:n_I}| \leq |\mathcal{F}_{:,:n_I}| = |\sigma(\mathcal{F}_{:,:n_I})|. \tag{126}$$

Together with Eq. (125), it follows that

$$\sigma(\mathcal{F}_{:,:n_I}) = \hat{\mathcal{F}}_{:,:n_I}. \tag{127}$$

Suppose $\mathbf{T}(\mathbf{s}) \neq \mathbf{D}_I(\mathbf{s})^{-1} \mathbf{P}_I^{-1}$, then

$$\exists j_1 \neq j_2, \ \mathcal{T}_{j_1,:} \cap \mathcal{T}_{j_2,:} \neq \emptyset. \tag{128}$$

Additionally, consider $j_3 \in \{1, \dots, n_I\}$ for which

$$\sigma(j_3) \in \mathcal{T}_{j_1,:} \cap \mathcal{T}_{j_2,:}. \tag{129}$$

Since $j_1 \neq j_2$, we can assume $j_3 \neq j_1$ without loss of generality. Based on Assumption iv, there exists $\mathcal{C}_{j_1} \ni j_1$ such that $\bigcap_{i \in \mathcal{C}_{j_1}} \mathcal{F}_{i,:n_I} = \{j_1\}$. Because

$$j_3 \notin \{j_1\} = \bigcap_{i \in \mathcal{C}_{j_1}} \mathcal{F}_{i,:n_I}, \tag{130}$$

there must exists $i_3 \in \mathcal{C}_{j_1}$ such that

$$j_3 \notin \mathcal{F}_{i_3,:n_I}. \tag{131}$$

Since $j_1 \in \mathcal{F}_{i_3,:n_I}$, we have $(i_3, j_1) \in \mathcal{F}_{:,:n_I}$. Therefore, according to Eq. (118), we have

$$\{i_3\} \times \mathcal{T}_{j_1,:} \subset \hat{\mathcal{F}}_{:,:n_I}. \tag{132}$$

Note that $\sigma(j_3) \in \mathcal{T}_{j_1,:} \cap \mathcal{T}_{j_2,:}$ implies

$$(i_3, \sigma(j_3)) \in \{i_3\} \times \mathcal{T}_{j_1,:}. \tag{133}$$

Then by Eqs. (132) and (133), we have

$$(i_3, \sigma(j_3)) \in \hat{\mathcal{F}}_{:,:n_I}. \tag{134}$$

This implies $(i_3, j_3) \in \mathcal{F}_{:,:n_I}$ by Eqs. (124) and (127), which contradicts Eq. (131). Therefore, we have proven by contradiction that $\mathbf{T}(\mathbf{s}) = \mathbf{D}_I(\mathbf{s})^{-1}\mathbf{P}_I^{-1}$. By replacing $\mathbf{T}(\mathbf{s})$ with $\mathbf{D}_I(\mathbf{s})^{-1}\mathbf{P}_I^{-1}$ in Eq. (113), we obtain Eq. (112), which is the goal. □

# B  Experiments

In this section, we describe the experimental settings as well as some additional results.

## B.1  Supplementary experimental settings

To produce observational data that meets the required assumptions for different models, we simulate the sources and mixing process as follows:

***UCSS***.  To ensure that the true nonlinear mixing process adheres to the *Structural Sparsity* condition (Assumption ii in Thm. 3.1), as per previous work (Zheng et al., 2022), we generate observed variables in a structured way: Each observed variable is only a nonlinear mixture of its direct ancestors. For instance, if the observed variable $\mathbf{x}_1$ has parents $\mathbf{s}_1$ and $\mathbf{s}_2$, then $\mathbf{x}_1 = \mathbf{f}_1(\mathbf{s}_1, \mathbf{s}_2)$. We use Generative Flow (GLOW) (Kingma and Dhariwal, 2018) with a projection layer as the nonlinear function $\mathbf{f}_i$. The difference between GLOW and GIN is that GLOW does not impose a constraint on the determinant of the Jacobian, thus being more suitable for the general nonlinear function since it has less inductive bias. The implementation of GLOW is a part of FrEIA[1] (Ardizzone et al., 2018-2022).

The ground-truth sources are sampled from a multivariate Gaussian, with zero means and variances sampled from a uniform distribution on $[0.5, 3]$, which are of the same values as in previous works (Khemakhem et al., 2020a; Sorrenson et al., 2020; Zheng et al., 2022). It is worth noting that we sample sources from a single multivariate Gaussian so that all sources are marginally independent, unlike from most previous works assuming conditional independence given auxiliary variables.

***Mixed***.  For the *Mixed* model, we partition sources into $\mathbf{s}_I$ and $\mathbf{s}_D$. For sources in $\mathbf{s}_I$, we sample them in the same way as that for *UCSS*. For sources in $\mathbf{s}_D$, we sample them from $2n_D + 1$ multivariate Gaussian distributions as required by Assumption ii in Thm. 4.4. Similarly, these multivariate Gaussian distributions are of zero means and variances sampled from a uniform distribution on $[0.5, 3]$. For sources in $\mathbf{s}_I$, we generate their influences on the observed variables in a structured way described above to satisfy the partial sparsity assumption (Assumption iv in Thm. 4.4); for sources in $\mathbf{s}_D$, we remove the constraint on the structure and permit each source to affect all observed variables.

***Base***.  For the *Base* model, following (Sorrenson et al., 2020), we use GLOW (Kingma and Dhariwal, 2018) as the mixing function to generate the data. The sources are from a single multivariate Gaussian distribution with zero means and variances uniformly sampled from the interval $[0.5, 3]$. No constraints on the structure have been imposed for the *Base* model.

In the evaluation of our model, we utilize the Mean Correlation Coefficient (MCC) as a metric for assessing the correspondence between the ground-truth and recovered latent sources. The MCC is calculated by first determining the pair-wise correlation coefficients between the true sources and the recovered sources after a nonlinear component-wise transformation learned by regression. Subsequently, an assignment problem is solved to match each recovered source with the corresponding ground-truth source that exhibits the highest correlation. MCC is a widely accepted metric in the literature for measuring the degree of identifiability, accounting for component-wise transformations (Hyvärinen and Morioka, 2016). Our results are all based on 20 trials, each with a different random seed.

For the synthetic datasets used in our experiments, the sample size is 2000. The parameters used for training include a learning rate of $0.01$ and a batch size of 200. Additionally, the number of coupling layers for both GIN and GLOW is set as 10. In regard to the "Triangles" dataset, it comprises $60,000$ $32 \times 32$ images of drawn triangles. The statistics of the dataset are described in (Yang et al., 2022). For the experiments conducted on this dataset, the learning rate is set at $3 \times 10^{-4}$ and the batch size is 100. Concerning the EMNIST dataset, it includes $240,000$ $28 \times 28$ images of real-world handwritten digits. The learning rate and batch size used for these experiments are $3 \times 10^{-4}$ and 240, respectively. The experiments are conducted directly using the official implementation of GIN[2] (Sorrenson et al., 2020) with an additional sparsity regularization term on the Jacobian of the estimated mixing function.

---

[1]https://github.com/vislearn/FrEIA
[2]https://github.com/VLL-HD/GIN

## B.2   Supplementary experimental results

In this section, we delve deeper into the applicability of our results by offering additional empirical studies that further illuminate the implications of the proposed theory. Specifically, we focus on the following aspects: (1) the effect of different regularization terms on the performance of identification; (2) the applicability of the theory as illustrated by additional real-world examples.

**Regularization.**   For the regularization term, directly utilizing the $\ell_0$ penalty may be computationally infeasible as it results in a discrete optimization problem. To overcome this issue, we adopt the $\ell_1$ regularizer, which has been extensively studied in the literature for high-dimensional support recovery, particularly for variable selection (Wainwright, 2009) and Gaussian graphical model selection (Ravikumar et al., 2008). The usage of $\ell_1$ regularizer induces sparsity in the solution; however, it may also introduce bias which can negatively affect the performance (Fan and Li, 2001; Breheny and Huang, 2011). This is because the $\ell_1$ norm penalty also

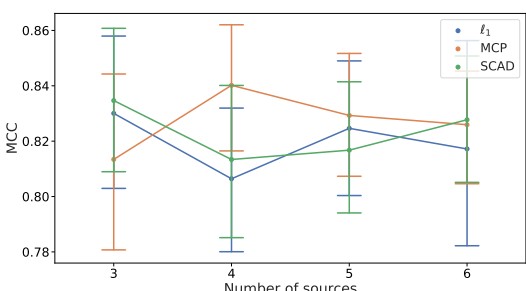

Figure 8: MCC of *UCSS* w.r.t. different sparsity regularizations and numbers of sources.

penalizes both small and large entries, unlike the $\ell_0$ norm which remains constant for nonzero entries. To remedy this bias issue, we explore alternative penalties, such as the smoothly clipped absolute deviation (SCAD) penalty (Fan and Li, 2001) and minimax concave penalty (MCP) (Zhang, 2010), which can be interpreted as hybrids of $\ell_0$ and $\ell_1$ penalties. Additionally, we note that the support recovery of the $\ell_1$ penalty is based on the incoherence conditions in various cases (Wainwright, 2009; Ravikumar et al., 2008, 2011), which may be restrictive in practice, whereas the SCAD and MCP penalties do not rely on such conditions (Loh and Wainwright, 2017). Based on our experimental results (Fig. 8), we adopt the MCP penalty as the regularization term.

**EMNIST.**   To further demonstrate the generalizability of the proposed identifiability results, we present identification results for all digits on the EMNIST dataset. As previously mentioned, we show the recovered attributes with the top-4 singular values. From Fig. 9, it is clear that these attributes are highly interpretable and appear to be the underlying concepts that influence the process of writing digits by hand. This indicates the potential applicability of our assumptions in real-world scenarios.

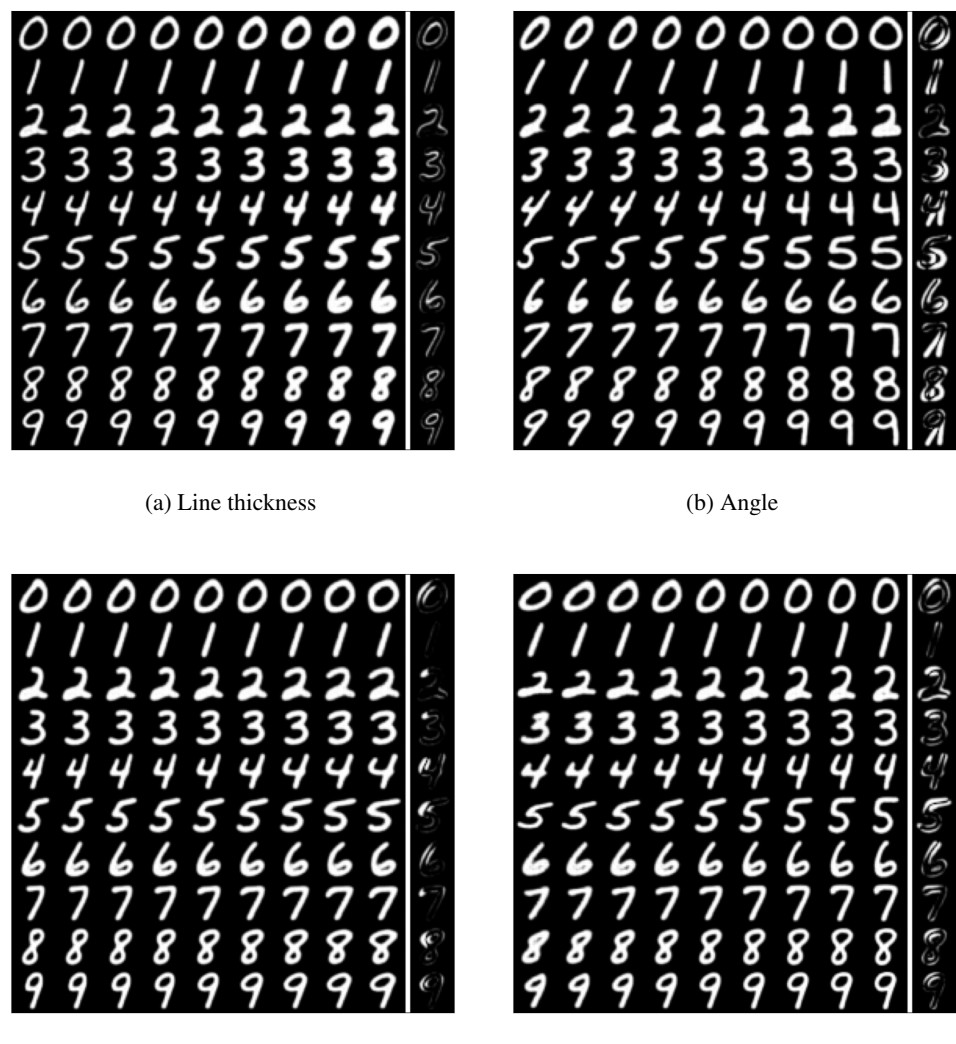

(a) Line thickness

(b) Angle

(c) Upper width

(d) Height

Figure 9: Results for all digit classes within the EMNIST dataset. We present the identified sources with the top-4 standard deviations SDs. Each sub-figure represents a source identified by our model, with its value varying from $-4$ to $+4$ SDs to illustrate its influence. The rightmost column presents a heat map given by the absolute pixel difference between the $-1$ and $+1$ SDs. The interpretation of these sources may correspond to line thickness, angle, upper width, and height, respectively.

