# OpenReview forum: "Generalizing Nonlinear ICA Beyond Structural Sparsity"
_NeurIPS.cc/2023/Conference — NeurIPS 2023 oral_

### Official Review · Reviewer_qJ69 · 2023-07-05

**Soundness:** 3 good
**Presentation:** 3 good
**Contribution:** 3 good
**Rating:** 7
**Confidence:** 3

**Summary:**

This paper utilizes the structural sparsity assumption on the support of the Jacobian matrix of the mixing function to extend the identifiability of nonlinear ICA in more settings including under-completeness, partial sparsity and source dependence, flexible grouping structures. It is a technically solid paper supported by theorems, proofs and experiments.

**Strengths:**

1. Overall, the manuscript is well written with clear organization, comprehensive literature review, technically solid theorems, detailed proofs and promising experiment results.

2. This work addressed some limitations of theorems about identifiability with Structural Sparsity in Zheng et al. 2022 and extended nonlinear ICA with Structural Sparsity to more general settings. The proposed theorems could be more practically useful in real-world datasets.

3. The notations, theorems and proofs are clear in general.

**Weaknesses:**

1. This work is interesting, and it would be great if code is provided to replicate the results. Please consider making the code publicly available.

2. The meanings of some notations are not clear. See Questions.

3. Ablation study. The author(s) only evaluated MCCs w.r.t. the number of sources. In Figure 3, the MCCs for 8 or 10 sources are a bit low so I wonder if more samples can help to improve MCCs. It would be more informative and convincing if more experiment configurations are considered (e.g., number of samples, various grouping structures) to demonstrate the effectiveness of proposed Theorems.

4. Ablation study. Though the result comparison seems obvious visually, the authors should consider performing statistical tests to compare results between proposed methods and baseline method.

5. Minor: "exits" should be "exists" at line 236.

**Questions:**

1. Theorem 3.1: Is $|\mathcal{F}_{i,:}|$ the $L_0$ or $L_1$ norm of $\mathcal{F}$? Is $\mathcal{C}_k$ a minimal set of sample indices to uniquely identify source $k$? How does the assumption ii show Structural Sparsity? The regularization constraint $|\hat{\mathcal{F}}| \leq |\mathcal{F}|$, which induces sparsity, should be included in Theorems. Also, I note that the author(s) tried to explain the assumptions in the following paragraphs, but I would suggest to describe the Theorems, at least Theorem 3.1, in plain words so that readers can better understand the Theorems. Or at least explain the notations (e.g., $\mathcal{C}_k$) which are not explained in Section 2 Preliminaries.

2. Theorem 3.1: Zheng et al. 2022 also proposed a Theorem on the undercomplete case. Could you kindly clarify the novelty between your proposed Theorem and that proposed in Zheng et al. 2022?

3. Theorem 4.1: The author(s) claimed that we do not need to know the dependence structures or the number of dependent sources, but it is not intuitive to me how Theorems 4.1 and 4.2 uncover the dependence structures and the number of dependent sources. Could you please clarify?

4. Theorem 4.2: What are $u_1$ and $u_2$? Are they two different sets of auxiliary labels?

5. Lines 281 - 283: The claim on multi-modal data is unclear. Could you please clarify how to identify linkage across multiple modalities?

---

> ### Author Rebuttal · Authors · 2023-08-10
>
> We sincerely appreciate your detailed reading and insightful questions. All of these constructive suggestions have further improved the quality of the updated manuscript. Please find our point-by-point response below.
>
> **Q1:** Implementation availability.
>
> **A1:** Thanks for finding our work interesting. We are more than happy to make the scripts publicly available soon. For quick utilization, kindly note that all experiments are conducted using public GitHub repositories (FrEIA and GIN), detailed in Section B.1. Please feel free to let us know if you have any questions about the experiments.
>
> **Q2:** Clarification on some notations.
>
> **A2:** We are very grateful for your detailed reading, which helps improve our manuscript's clarity. We have emphasized all these points in the updated manuscripts:
>
> - **Q2(a):** Is $|\mathcal{F}|$ the $\ell_0$ or $\ell_1$ norm of $\mathcal{F}$?
>
> - **A2(a):** It is the $\ell_0$ norm of $\mathcal{F}$.
>
> - **Q2(b):** Is $\mathcal{C}_{k}$ a minimal set of sample indices to uniquely identify source $k$?
>
> - **A2(b):** We do not necessitate the set $\mathcal{C}\_{k}$ to be minimal, only that it uniquely identifies $k$. Of course, it is equivalent to consider $\mathcal{C\}_{k}$ as a minimal set.
>
> - **Q2(c):** What are $\mathbf{u}\_1$ and $\mathbf{u}\_2$?
>
> - **A2(c):** These are two distinct values of the auxiliary variable $\mathbf{u}$.
>
> **Q3:** More experiment configurations.
>
> **A3:** Thanks for your suggestion. In light of it, we have conducted additional experiments with different sample sizes, of which the results are shown in the attached PDF in the global response. From the results, it is clear that increasing the sample size could improve MCCs and further stabilize the performance.
>
> **Q4:** Statistical tests to compare results between proposed methods and baseline methods.
>
> **A4:** Thanks a lot for the suggestion. Accordingly, we have conducted statistical tests on all comparisons, and all p-values are less than $0.01$, which are consistent with the visual differences in the violin graphs. We have emphasized this in the updated manuscript.
>
> **Q5:** Word "exits" should be "exists" at L236.
>
> **A5:** Thanks! It has been corrected.
>
> **Q6:** How does assumption ii of Thm. 3.1 show Structural Sparsity?
>
> **A6:** We are grateful for that insightful question. If the connective structure between sources and observed variables is extremely dense (e.g., no zero entry in the Jacobian matrix), assumption ii cannot be satisfied, emphasizing its role in promoting structural sparsity. However, it is worth noting that, after extending from the bijective setting in [1], where the assumption of Structural Sparsity is originally proposed, to the undercomplete case in our manuscript, this assumption can be met even with a relatively dense structure, provided there are enough observed variables and the underlying graph is not fully connected. Thus, it leans more toward a "structural diversity" assumption in our generalization, and the name “Structural Sparsity” is chosen primarily to acknowledge its root and maintain continuity with its original name.
>
> **Q7:** Better include the regularization constraint during estimation in theorems.
>
> **A7:** Thank you for the suggestion. We have now incorporated it into all related theorems to avoid potential misunderstanding. Initially, we had just emphasized it following the theorem since it was not an assumption about the data-generating process, but including it directly in the theorem indeed improves the clarity.
>
> **Q8:** Better explanation of theorems and some notations (e.g., using plain words) so that readers can understand them more easily.
>
> **A8:** Thanks a lot for the kind suggestion. We have added descriptions in plain words of all theorems in the updated manuscripts. For example, Thm. 3.1 states that with sufficient sample size and structural sparsity (detailed later) on the connective structure between sources and observed variables, component-wise identifiability can be achieved even under undercomplete cases with sparsity regularization. We have also highlighted some notations used in the theorems. For instance, $\mathcal{C}_{k}$ in the assumption of Structural Sparsity denotes a set of source indices.
>
> **Q9:** Could you clarify the novelty between your proposed theorem and that proposed in [1] about the undercomplete case?
>
> **A9:** We appreciate the great question. As mentioned in L159-160, [1] only removes the rotational indeterminacy while our theorem removes all major indeterminacies and only preserves the component-wise transformation and permutation. In other words, [1] only gets rid of specific spurious solutions due to the rotational indeterminacy (e.g., the ‘rotated-Gaussian’ MPA) while we prove the full identifiability of the undercomplete case.
>
> **Q10:** How do Thms. 4.1 and 4.2 uncover the dependence structures and the number of dependent sources?
>
> **A10:** Thanks for your insightful question. For the set of dependent sources, i.e., $\mathbf{s}\_D$, Thms. 4.1 and 4.2 only provide the subspace-wise identifiabliity up to an invertible transformation, and thus not being able to uncover the structures and the number of dependent sources. The identifiability of $\mathbf{s}\_D$ up to an invertible transformation means that $\mathbf{s}\_D$ will not be mixed with sources in $\mathbf{s}\_I$ after estimation, i.e., the block-wise identifiability (e.g., Thm 4.2 in [2]), which does not mean the component-wise identifiability of sources in $\mathbf{s}\_D$.
>
> **Q11:** How to identify linkages across multiple modalities.
>
> **A11:** Thank you for the great question. Since we stil assume the (conditional) independence between different subspaces $\mathbf{s}\_{c\_j}$ (Eq. 4), we do not allow dependencies across subspaces and thus cannot identify linkages across multiple modalities.
>
> ---
>
>
>
> [1] Zheng et al. "On the identifiability of nonlinear ICA: sparsity and beyond."
>
> [2] Kong et al. “Partial identifiability of domain adaptation.”

---

### Official Review · Reviewer_m3QW · 2023-07-05

**Soundness:** 2 fair
**Presentation:** 2 fair
**Contribution:** 3 good
**Rating:** 6
**Confidence:** 2

**Summary:**

The article serves as an extension to the work of Zheng et al. 2022, which posited the identifiability of nonlinear ICA based on specific structural sparsity assumptions related to the mapping of sources and mixtures. This current article expands on that by addressing the undercomplete case—where the number of mixtures exceeds the number of sources—and furthermore, it relaxes both the sparsity and source independence assumptions to yield more general identifiability results.	In the end authors provide some numerical examples to illustrate the applications of their identifiability theorems.

**Strengths:**

The article tackles the foundational issue of the nonlinear inverse problem, making significant assertions regarding fundamental identifiability theorems. These are premised on assumptions of partial independence and structural sparsity.The problem under investigation is a fundamental problem and the article offers some important results for this problem.

**Weaknesses:**

The most significant shortcoming of the article lies in its presentation, particularly in its explanation of the core assumptions underpinning the theorems, as well as the motivation for the conditions applied in these assumptions. Absent a solid grasp of these theorems, it becomes challenging to properly evaluate the paper's contributions and ascertain its potential impact.

In terms of specific issues:

* Concerning Theorem 3.1: This theorem seems to aim at generalizing Theorem 1 from Zheng et al., 2022 for a complete (m=n) case to an undercomplete (m>n) case. The identifiability results for ICA setups typically do not depend on a specific estimator choice or estimation algorithm. However, the statement of Theorem 3.1 seems rather unclear in this context. According to the article's notation, \hat{f} refers to a specific estimate of the mixing function. Both the set of support matrices \mathcal{T} and the support \hat{\mathcal{F}} depend on this particular estimate. The assumption (i) used in Theorem 3.1 is based on \mathcal{T} and \hat{\mathcal{F}}, hence, this condition appears to be linked to a specific choice of the estimator for mixing. It would be beneficial if the authors could clarify whether the assumption (i) must hold on a particular \hat{f}, a specific set of functions, etc. This clarification will likely impact the proof in the supplementary material and the explanation given between lines 129-136.


* Line 97: Traditional or linear ICA does not necessarily require m=n.

* Line 114: Should \mathcal{S} be \mathcal{A}?

* Line 111 vs Line 536: The symbol \mathcal{T} has two differing definitions - the set of matrices sharing the same support as T(s) and the support of T(s) itself.

* Theorem 4.1: The vectors 'w' - whose independence implies identifiability - have not been sufficiently motivated or explained.

* Similar comments can be made for other identifiability theorems.


**Questions:**

* Line 190: Could you clarify what is meant by "changing" sources?

* Line 195-196: 6, the phrase "For sources s_D, they do not need to be mutually independent as long as they are dependent on the variable u" is somewhat confusing. Subsequent equation (3) implies that S_D and S_I are conditionally independent when conditioned on u, and that the components of s_I are independent. What is the necessity for this latent variable u? Perhaps the authors could shed some light on this.

* How does Theorem 4.1’s contribution compare with those from Khemakhem et al. (2020a) and Sorenson et al. (2020)?

* Lines 215-217: Could you elucidate what condition (i) in Theorem 4.2 represents? What do s_d and B_{s_I} signify? Perhaps the discussions on lines 229-247 should precede Theorem 4.2 to better contextualize its contents and results, potentially with more lucid explanations.

* Regarding Figure 4 and 5, could you expound on how these examples pertain to the nonlinear ICA setup (what are the sources, which appear to be images, and what are the nonlinear mixings)? How are the interpretations in the captions of these figures derived? Could you also elucidate how these examples relate to the identifiability theorems presented in the article and the conditions stipulated in these theorems?


**Limitations:**

Yes, the authors adequately addressed the limitations.

---

> ### Author Rebuttal · Authors · 2023-08-10
>
> We appreciate the reviewer for the time dedicated and constructive feedback, which has greatly improved the quality of the updated manuscript. Please find the responses to all your comments below.
>
> **Q1:** More discussion on the assumption (i) in Thm. 3.1.
>
> **A1:** Thanks so much for the suggestion. The assumption (i) does not necessitate a particular $\hat{\mathcal{F}}$, and it is typically satisfied asymptotically as detailed in L129-136 (it only necessitates the **existence** of one $\mathrm{T}$ in the entire space). Except for the required sparsity regularization (L125-128), there are no specific functional classes to constrain $\hat{\mathcal{F}}$ during estimation. We have further highlighted it in the updated manuscript.
>
> **Q2:** L97: Traditional or linear ICA does not necessarily require m=n.
>
> **A2:** Thanks a lot. We have modified it to: “Different from settings where m=n …”
>
> **Q3:** L114: Should $\mathcal{S}$ be $\mathcal{A}$?
>
> **A3:** Yes, you are totally right. We have corrected the typo in the updated manuscript.
>
> **Q4:** L111 vs L536: $\mathcal{T}$ has two differing definitions.
>
> **A4:** Thanks for rasing this point. We have corrected the denotation of $\mathcal{T}$ in L536 to a set of matrices with the same support of $\mathbf{T}(\mathbf{s})$, and correspondingly, $\mathrm{T} \in  \mathcal{T}$ in the following sentence.
>
> **Q5:** Thm. 4.1: the linearly independent vectors '$w$'  have not been sufficiently motivated.
>
> **A5:** Thanks for the suggestion. The linear independence of $w$ requires that the conditional distribution varies sufficiently across different values of $\mathbf{u}$ (L229-230). The motivation of it is similar to the common assumption of variability used in [1, 2, 3]. Even though the assumption of variability is almost surely fulfilled as discussed in [1, 2, 3], our assumption is still strictly weaker from various perspectives as detailed in L229-241. For example, our definition of $w$ is only the first half of Eq. 8 in [1] and we require much fewer distinct values of $\mathbf{u}$. We have further highlighted it in the updated manuscript.
>
> **Q6:** L190: The meaning of "changing" sources.
>
> **A6:** Thanks for the question. The "changing" sources ($\mathbf{s}\_D$) mean that the distribution of these sources changes across different values of the auxiliary variable $\mathbf{u}$. For instance, styles of images change across domains while their content stays invariant.
>
> **Q7:** L195-196: More clarification on "For sources $\mathbf{s}\_D$, they do not need to be mutually independent as long as they are dependent on the variable $\mathbf{u}$".
>
> **A7:** Thank you for your suggestion. It means that sources in $\mathbf{s}\_D$ (i.e., those with indices $\\{n\_{I+1},\cdots,n\\}$) do not need to be (mutually) conditionally independent given the auxiliary variable $\mathbf{u}$; they only need to be influenced by (dependent on) $\mathbf{u}$. The variable $\mathbf{u}$ only provides changes on the distributions of these sources, like a domain label or time index.
>
> **Q8:** How does Thm. 4.1’s contribution compare with those from [2, 3]?
>
> **A8:** Thanks for raising this. We only require sources in $\mathbf{s}\_D$​ to depend on an auxiliary variable $\mathbf{u}$ with $n\_D​+1$ values, intuitively fitting scenarios where fewer changes (smaller $n\_D$) correspond to easier identifiability (fewer required values, i.e., $n\_D+1$). In contrast, prior works like [2, 3] assume all sources to depend on $\mathbf{u}$ with $nk+1$ values ($k$ denotes the order of the distribution), limiting identifiability to ideal situations without any degree of violations on either the number of sources dependent on $\mathbf{u}$ or the number of values of $\mathbf{u}$. This can restrict practical application, as often only subsets of sources are influenced by auxiliary variables, or those variables lack sufficient variability. Of course, as a trade-off, Thm. 4.1 itself does not provide component-wise identifiability. More details can be found in L224-247.
>
> **Q9:** L215-217: More explanation on the motivation and some notations ($\mathbf{s}\_d$ and $B\_{\mathbf{s}\_I}$) of condition (i) in Thm. 4.2.
>
> **A9:** We appreciate these insightful questions and suggestions. Condition (i) in Thm. 4.2 originates from [4], which intuitively means there exist two values of the auxiliary variable such that their influences on the sources are different. $\mathbf{s}_d$ is a typo and should be $\mathbf{s}\_D$, which denotes sources that dependent on $\mathbf{u}$ (sorry for the confusion). $B\_{\mathbf{s}\_I}$ is a subspace of $\mathcal{S}\_I$. In light of your suggestions, we have moved the discussion (L229-L247) to the paragraph directly following Thm. 4.2 and further emphasized these notations.
>
> **Q10:** More explanation on Figs. 4 and 5.
>
> **A10:** For these images (triangles in Fig. 4 and hand-written digits in Fig. 5), we assume that they are generated by hidden sources (e.g., angle, height, etc.), and try to recover these from their observed mixtures (i.e., images). Each row represents a source we identified, and we vary its value with the rightmost column showing a heatmap of the absolute pixel difference to visualize its influence. We interpret the estimated sources’ potential semantics from their influences, as listed in the captions. We only deal with single classes (e.g., zero in EMNIST), without auxiliary variables (e.g., digit labels). Thus, prior identifiability theories relying on auxiliary variables cannot support these seemingly reasonable results, but our generalized theorems could probably underpin them.
>
>
> ---
>
>
>
> [1] Hyvarinen et al. "Nonlinear ICA using auxiliary variables and generalized contrastive learning."
>
> [2] Khemakhem et al. "Variational autoencoders and nonlinear ica: A unifying framework."
>
> [3] Sorrenson et al. "Disentanglement by nonlinear ica with general incompressible-flow networks (gin)."
>
> [4] Kong et al. “Partial identifiability of domain adaptation."

---

> > ### Comment · Reviewer_m3QW · 2023-08-12
> >
> > I would like to thank the authors for their clarifications.

---

> > > ### Author Response · Authors · 2023-08-16
> > >
> > > Thanks so much for your further feedback.

---

### Official Review · Reviewer_nmMP · 2023-07-07

**Soundness:** 3 good
**Presentation:** 3 good
**Contribution:** 3 good
**Rating:** 6
**Confidence:** 3

**Summary:**

The paper extends identifiability theory of nonlinear ICA (NICA), and deep latent variable models in general, by utilizing structural sparsity. In particular, previous works have shown that NICA can be identified if there is some observed auxiliary data or latent dependencies that essentially capture the inductive biases in the data generative process. The approach of structural sparsity (Zheng, '22) instead takes an alternative approach, namely constraining the nonlinear mixing function and its Jacobian. In this paper the authors extend that work as follows i.e. assuming structural sparsity and some additional assumptions:

1. the authors show identifiability for a situation in which the mixing function is injective rather than bijective i.e. undercomplete case / i.e. smaller latent dimension than observed dimension. This is important for e.g learning low dimensional, semantically meaningful, interpretable latent features

2. the authors show identifiability for a situation in which the structural sparsity principle applies only to some of the independent components

3. further identifiability is shown for situation where not all the latent components are independent, rather components form independent subspaces. In fact some components may be dependent, conditionally independent or have some grouping structures.

**Strengths:**

First, this paper is in general of good quality in that it is well organized and, in general, clearly written.

Main strengths:
a.) most significantly, authors remove several strong limitations of previous works and extend identifiability of structural sparsity to undercomplete and case where not all latent components are structurally sparse (in those situations the remaining sources are shown to be identifiable). As a result these ideas are now more applicable to realistic data and scenarios.

b.) These results have been reached, mostly, without too strong additional assumptions. For instance, it is shown that the necessary assumptions are more likely to hold in this new undercomplete case which is encouraging!

c.) authors bridge gap between structural sparsity and the previous works that assume auxiliary variables. in particular, this work allows unconditionally independent components to follow structural sparsity and components which are conditionally independent given auxiliary variables. Whilst arguably to be expected, it is important to show this result (but see below for potential related weakness)

**Weaknesses:**

In general the weakness of this paper is that provides only few theoretical advances (albeit important; as mentioned above) but provides little beyond that. In particular, this is the results of:

1. Contribution of the paper is not as significant as the authors describe, or at least there is limited coverage of relevant works
2. Potential problems in some of the identifiability theorems
3. Novelty is limited to identifiabiltiy theorems -- no new algorithms
4. Experiments are lacking

I will expand on each of these points below:

More detail for 1.):  In particular the authors state that "Therefore, we establish, to the best of our knowledge, one of the first general frameworks for uncovering latent variables with appropriate identifiability guarantees in a principled manner". I think this is too vague and general and fails to acknowledge the generality of some other works -- your work can be novel whilst admitting the generality of some other works too. First, Kivva '22 show a very general framework for identifiability by making, arguably, less strong assumptions on the mixing functions -- currently the work of Kivva '22 is only mentioned later on in section 3 (and even there in a problematic manner as ill point out below). Due to the generality of the results in Kivva, I would expect their result to be discussed in the introduction / early on in the text and tell the reader why yours is better or at least different. For example Kivva make different type of assumptions on the mixing function (piecewise affine) etc, while you on the sparsity. Second the work of Halva '21 (disentangling identifiable features) provides another very general framework and unlike what you claim, it is not limited to time-series but to any dependencies of arbitrary order, and also does not require condition independence on some auxiliary variables but rather also assumes unconditional independence.

More detail for 2.): In Theorems 4.1 and 4.3  $S_d$ is "identified up to an invertible transformation". Surely if something is identified just up to invertible (vector-valued) function then we are not doing any better than nonlinear ICA i.e. we are essentially where one started and thus we have not identified anything. To me this is misleading and not a publishable identifiability result (if I have understood correctly -- please correct me if I'm wrong and I'll adjust my score accordingly). Authors do acknowledge this point but rather than talking about it they make a vague remark that "Thm. 4.1 may be helpful for some tasks that do not necessitate the recovery of each individual source, such as domain adaptation." This does not suffice in my opinion. And similarly about Thm 4.3 they say: "there exists an invertible transformation $h_{c_i}$ which is analogous to the previous element-wise indeterminacy. Consequently, even when dealing with mixtures of high and one-dimensional sources, like in the case of multi-modal data, we can still recover the hidden generating process to some extent." Again I think this is bit generous and hiding the fact that $\mathbf{s}_{c_i}$ is fully unidentifiable in the sense of nonlinear ICA. At least this limitation must be admitted more clearly -- preferably its usefulness would be shown empirically.

More detail for 3.):
There is a simple regularization term of the jacobian added -- but this is heurestic (vs. mle methods) from previous work. Undoubtedly there could be work done towards what is the best way to estimate a model that assumes structural sparsity but such is not done here.

More detail for 4.)
An important question is whether structural sparsity is a valid assumption. I think this can indeed be the case for many types of generative processes.  But the question then is then do the experiments strengthen that intuition. I feel not. It is not clear to me why e.g. EMNIST experiment there would be structural sparsity. EMNIST is also a very simple data set. I would expect the experiments to show that the learned independent components are useful in practical applications (see the brain signal experiments in Halva '21 for instance or in Khemakhem (iVAE) '20). I'm not saying specifically this type of real data experiments need to be introduced, but something to further highlight the strenght of this method would be helpful. Another example is to evaluate the method more thoroughly on some benchmarks from the disentanglement learning literature. There are also some claims in the paper that would be good to justify experimentally for instance you claim on line 311-313 that "This is particularly helpful in the context of self-supervised learning 311 (Von Kügelgen et al., 2021) or transfer learning (Kong et al., 2022), where latent representations are 312 modeled as a changing part and an invariant part." If this indeed the case, why not show that on data? Indeed 311 to 324 gives nice discussion and its a great shame this has not been shown experimentally as it would really take this paper to the next level.

**Questions:**

I will use this space for further suggestions and questions:

- Could you please introduce structural sparsity bit more clearly and intuitively -- if one has not read the original Zheng'22 paper then it is difficult to follow.

- In estimation, please clarify: is it required that we know which groups of latent variables are independent, and which are potentially dependent? How is this exactly established in practice?

- please explain in more detail how your algorithm, in practice, allows dimension reduction without assuming observation noise, and how jacobian can be computed for non-bijective transformation

- What is the level of nonlinearity in the mixing functions? I dont believe this is mentioned anywhere, e.g. number of layers or similar.

- ". Since the proposed condition is on the connective structure from sources to observed variables, i.e., the support of the Jacobian matrix of the mixing function, it does not require the mixing function to be of any specific algebraic form.". Please make this sentence bit more precise or explain better what does 'specific algebraic form' mean. Because structural sparsity does still limit the form of the function -- f can not longer be any arbitrary function.

-  "Most of these methods require auxiliary variables to be observable, such as class labels and domain indices (Hyvärinen and Morioka, 2016, 2017". H&M 2017, really only require the previous data so it's not really a big limitation and it's arguable whether this really constitutes of having auxiliary variables...I would consider moving that reference to the next sentence since it's a time-series model : "with the exceptions being those for time series...[move H&M'17 reference here]"

- "The most obvious  one arises from the fact that it may fail in a number of situations where the generating processes are heavily disentangled." Please explain in more detail why this may be?

- "We first present the result on removing one of the major assumptions in ICA, i.e., the number of observed variables m must be equal to that of hidden sources n." This makes it sound like it hasn't been done previously in general, which of course it has been done many times previously in linear ICA (e.g eriksson and koivunen '03) and nonlinear ICA (e.g. khemakhem '20, halva '21 etc etc). so rather than saying removing majort assumption in ICA, make it specific to sparsity

- as for the title: are you really "generalizing beyond structural sparsity"? as I feel structural sparsity is still the fundamental building block here. I would say you are generalizing structural sparsity in nonlinear ICA.

- "This is similar to Independent Subspace Analysis (ISA) (Theis, 2006)"  Either explain why you cite Theis, or cite an earlier ISA work (e.g. Hyvarinen & Hoyer, 2000)?

**Limitations:**

Authors should discuss limitations in more detail:

- what are possible limitations of structural sparsity assumptions should be discussed more e.g. any scenarios where you expect it to be a poor assumption?

- the point discussed in the "Weaknesses" section about the limitations in the identifiability theorems of 4.1 and 4.3 must be addressed and justified much more clearly or else those theorems should be removed

- authors should note the heurestic approach of their estimation algorithm -- does GIN even have universal _function_ approximation capability?

- related, discuss limitations of estimation algorithm in general. Is the algorithm guaranteed to find the correct sparsity for example? If not, then that should be pointed out as need for future works.

- "However, our setting is more flexible in the sense that we do not assume all sources to be influenced by the auxiliary variable. Specifically, sources in $s_I$ are mutually independent as in the original ICA setting, while only  sources in $s_D$ have access to the side information from the conditional independence given u,". This is true and a nice result of theorem 4.4, but there is the limitation that should be discussed, namely now you are making restricting assumption on _both_ the mixing function as well as on the auxiliary variables -- in some sense this is worst of both worlds (but still a nice theoretical result with possible practical uses).

---

> ### Author Rebuttal · Authors · 2023-08-10
>
> Thanks so much for your time and insightful comments. We are glad that you find our results important and the paper of good quality. Please find our point-by-point response below.
>
> **Q1:** Limited coverage of some works.
>
> **A1:** Thanks for the suggestions. We have now detailed the discussion in the introduction. Specifically, in [1], there are assumptions on both source distribution and mixing functions: (1) the sources are assumed to be a Gaussian mixture, with an unobserved state $\mathbf{u}$ (L158-161); (2) the mixing function is assumed to be piece-wise affine. These allow identifiability of sources up to an affine transformation where mixtures remain, with more assumptions needed (e.g., conditional independence given $\mathbf{u}$) for component-wise identifiability.
>
> Thus, we differ by (1) having no distributional assumptions, and (2) allowing general nonlinear functions if the assumption on connective structures is met. We do not claim that our assumption on the mixing functions is better than those in [1]. Structural sparsity allows general nonlinearity with sparse connections, while piece-wise affine functions allow dense structures.
>
> In addition, we fully agree that [2] should be further emphasized for its highly significant contributions. We have highlighted its generalization to arbitrary dependency order (e.g., spatial dependencies) without assuming conditional independence. Meanwhile, additional information (e.g., time or spatial index) may not always be available, which is one of the motivations of our work.
>
> **Q2:** Implication of Thms. 4.1 and 4.3.
>
> **A2:** Thanks for the question. We have added more descriptions to clarify this. In Thm. 4.1, the identifiability of $\mathbf{s}\_D$ up to an invertible transformation means that $\mathbf{s}\_D$ will not be mixed with sources in $\mathbf{s}\_I$ after estimation. In practice, it implies that the **subspace** of the changing part (the distribution of $\mathbf{s}\_D$ changes w.r.t. $\mathbf{u}$) can be disentangled from the mixture of both changing and invariant parts ($\mathbf{s}$), which aids tasks like domain adaptation, e.g., the changing style (as a whole) can be disentangled from different images with invariant content. Similarly, in Thm. 4.3, the subspace-wise identifiability means sources in $\mathbf{s}\_{c\_i}$ will not be mixed with sources outside $\mathbf{s}\_{c\_i}$, which disentangles $\mathbf{s}\_{c\_i}$ as an individual high-dimensional component.
>
> **Q3:** Novelty appears limited to theorems, and whether sparsity penalty and GIN are just heuristics.
>
> **A3:** According to our theory, the additional sparsity penalty on the MLE objective during estimation (L339-341), together with assumptions on the data-generating process, is needed to guarantee the correct identification (L125-128). Moreover, according to [3], coupling-based flows (e.g., GIN) are universal diffeomorphism approximators. The volume-preserving nature of GIN does not hinder it from validating our theorems, as rescaling is one of the allowed indeterminacies after identification. Of course, there exists some approximation of $\ell_0$ norm for gradient-based optimization (MCP penalty), and more work could be done for further improvement.
>
> **Q4:** Experiments on more tasks.
>
> **A4:** Indeed, there are various tasks that benefit from our theory, and more applications would be intriguing. Meanwhile, we have emphasized in introduction (L88-89) and experiment (L329-334) with additional citations (>9) that prior research shows latent variable models are likely identifiable in complex scenarios, possibly involving undercompleteness and violations of sparsity and independence. Our theory may interpret these empirical results, and our ablation studies and experiments on both the synthetic and real-world datasets provide further validations, complementing previous works.
>
> **Q5:** Discuss structural sparsity and failure cases more.
>
> **A5:** Thanks, we have added more discussion on it. For failure, one may consider recording in a very crowded room, where every microphone records the mixture of signals from most sources at the same time.
>
> **Q6:** Is it necessary to know which latent variables are independent/dependent?
>
> **A6:** No. In practice, each data point can be assigned a class corresponding to the value of the auxiliary variable of dependent variables (L335-337). These labels do not provide extra information for independent variables since they do not need auxiliary variables.
>
> **Q7:** How to reduce dimensions and compute the Jacobian for non-bijective transformations?
>
> **A7:** As in prior works (e.g., [4]) and noted in L346-347, we concatenate latent sources with independent Gaussian noises for flow-based estimation's dimensionality needs and Jacobian computation.
>
> **Q8:** Level of nonlinearity, e.g., number of layers.
>
> **A8:** 10 layers (L881).
>
> **Q9:** What does 'specific algebraic form' mean?
>
> **A9:** Great question. We meant to refer to "specific" function classes like conformal mappings mentioned above but have clarified this by changing the term to 'the above-mentioned classes' to avoid confusion.
>
> **Q10:** Explain: "The most … are heavily disentangled."
>
> **A10:** Apologies for the typo. It should be "heavily **entangled**", like a crowded room where sources heavily influence each other, resulting in a dense Jacobian.
>
> **Q11:** More suggested updates:
>
> - (1) relocate a reference;
>
> - (2) specify undercompleteness claim to sparsity;
>
> - (3) a new title;
>
> - (4) cite an earlier ISA work;
>
> - (5) further highlight a limitation of Thm. 4.4.
>
> **A11:** We are grateful for all the constructive suggestions. All points have been updated accordingly.
>
>
> ---
>
> [1] Kivva et al. "Identifiability …"
>
> [2] Hälvä et al. "Disentangling …"
>
> [3] Teshima et al. "Coupling-based invertible neural networks are universal diffeomorphism approximators."
>
> [4] Sorrenson et al. "Disentanglement by nonlinear ICA with general incompressible-flow networks (gin)."

---

> > ### Author Response · Authors · 2023-08-16
> >
> > Thank you once more for your time and suggestions. Since the discussion window narrows, might we kindly ask if our clarification has resolved the potential confusion, especially in **Q2 & A2**? Your further feedback is deeply appreciated.

---

> > > ### Author Response · Authors · 2023-08-19
> > >
> > > Sorry for the repeated reminders. As the discussion will end in **48 hours**, would you mind kindly checking if you have any further questions? For instance, we have clarified the **implication of Thms. 4.1 and 4.3**, and you have mentioned in the second point of weakness that
> > >
> > > > "if I have understood correctly -- please correct me if I'm wrong and I'll adjust my score accordingly"
> > >
> > > As all reviewers noted, our results are important to the community in a timely manner. Thus, we would like to try our best to address any potential confusion given the opportunity for discussion.

---

> > > > ### Author Response · Authors · 2023-08-19
> > > > **Mistake in updating the rating?**
> > > >
> > > > We have been eagerly looking forward to your feedback on our detailed response.  However, without seeing your feedback, we noticed that the rating by you was changed from 4 (Borderline reject) to 3 (Reject).  We therefore are wondering whether it was intended to be changed this way or just a mistake.  If you have any further questions or concerns, please kindly let us know.  Your feedback will be appreciated.

---

> > > > > ### Comment · Reviewer_nmMP · 2023-08-19
> > > > >
> > > > > I am confused -- can you not see my comment to your rebuttal I made above?

---

> > > > > > ### Author Response · Authors · 2023-08-19
> > > > > >
> > > > > > Thanks for your reply. We didn't find any feedback from you except the score change. Would you mind kindly checking if we are the readers of that comment?

---

> > > > > > > ### Comment · Reviewer_nmMP · 2023-08-19
> > > > > > >
> > > > > > > I am not sure why but it did indeed look like you were missing from 'readers'. I also couldnt edit it. So here I am re-sending it:
> > > > > > >
> > > > > > > I have now read all the reviews and rebuttals carefully. I feel like I have no choice but to reduce the rating by one point from '4' to '3' as it better reflects my current opinion of the paper. Overall the paper makes interesting theoretical contribution but the results are not as novel as the author claims, and a lot of them are to be expected as combinations of previous results (e..g sparsity and auxiliary variables combined). The experimental validation lacks breadth. Additionally the following are further problems that justify my new rating:
> > > > > > >
> > > > > > > - One of the other reviews made me realize that undercompleteness was already discussed in [1]. This result is not properly acknowledged in this paper at all. The 'introduction' is written as if the authors are presenting the combination of undercompleteness and structural sparsity for the first time. For example, in the introduction you write: *"Unfortunately, Zheng et al. (2022) require Structural Sparsity to hold for all sources in order to provide any identifiability guarantee...[paragraph change] Besides partial sparsity, identifiability with Structural Sparsity also fails with the undercompleteness (more observed variables than sources) and/or partial source dependence (potential dependence among some hidden sources)."*. This makes it look like the structural sparsity in Zheng would fail in undercomplete case and does not at all acknowledge that they in fact *do* provide result for undercomplete case. This is not good, and the introduction **must** be changed to properly attribute for this. Currently you only make a comment on Zheng's work on undercompleteness at the end of section 3 (l. 158). To raise my score, I expect to see exact changes the author will implement.
> > > > > > >
> > > > > > > - **with regards to author's A2**: You write *"In Thm. 4.1, the identifiability of $S_d$ up to an invertible transformation means that it will not be mixed with sources in $S_I$ after estimation."*. Yes I agree that your are identifiably disentangling $S_d$ and $S_I$ from each other however saying you 'identify $S_d$ up to invertible transformation' is non-sensical in that that the $\hat{S}_d$ that you estimate have no guarantee of being related in any reasonable way to the ground-truth $S_d$ (so completely unidentified). Further, you don't even say anything about $S_I$ in Thm. 4.1. But merely: "Then $s_D$ is identifiable up to an invertible transformation.". I expect to hear of the exact changes you would make to Thms 4.1 - 4.3 so that the reader gets a better picture of what these theorems *really* mean.
> > > > > > >
> > > > > > > - **wrt. A3:** You should include a short explanation of this in the paper (the part about universal approximation and GIN), if it's not already there.
> > > > > > >
> > > > > > > - **wrt. A4:** If I understand correctly, you essentially claim here that you don't need more experiments because previous literature has shown identifiability in all kinds of situations and you provide explanation of that. I think you are being far too generous to yourselves here. There are million different possible inductive biases that could potentially explain these observations (ofc. including your work too) but it is not good enough reason in my opinion for the lacking empirical side.
> > > > > > >
> > > > > > > I am happy to revise my score if these issues have good resolution.
> > > > > > >
> > > > > > > ***
> > > > > > > [1] Zheng et al. (2022), "On the Identifiability of Nonlinear ICA: Sparsity and Beyond"

---

> > > > > > > > ### Author Response · Authors · 2023-08-19
> > > > > > > > **Thanks so much for your further suggestions (1/2)**
> > > > > > > >
> > > > > > > > Thank you very much for your further comment. We respectfully believe that there exist some misunderstandings that could be addressed. We are very glad that you provide us the opportunity to further clarify and highlight them. In light of your suggestions, we have further incorporated changes in the updated version:
> > > > > > > >
> > > > > > > > **Q12:** The novelty compared to the result about undercompleteness in [1].
> > > > > > > >
> > > > > > > > **A12:** Thanks a lot for your suggestion, which let us realize again the necessity of clarification on that earlier in the introduction. We must emphasize that **[1] did not give any identifiability result on the undercomplete nonlinear ICA at all**. As discussed in (1) A2 in the response to reviewer jekt, (2) A9 in the response to reviewer qJ69, and (3) L159-160, [1] did not give an identifiability result but only provide a way to distinguish spurious solutions due to rotated-Gaussian MPA (Thm. 3 in [1]). Thus, identifiability with Structural Sparsity and undercompleteness was not established in [1].
> > > > > > > >
> > > > > > > > Since this misunderstanding can be clearly avoided by highlighting the difference between proving identifiability and distinguishing a specific spurious solution, we have added additional detailed discussions **earlier in the introduction and abstract** to avoid potential confusion, as mentioned in the responses to other reviewers. Specifically, the related sentences in the **introduction (after L81)** have been added as follows:
> > > > > > > >
> > > > > > > > > ”We would like to highlight that [1] provided guarantees to avoid the spurious solution of rotated-Gaussian MPA in the undercomplete case with structural sparsity. However, it is not an identifiabilty result since there exist numerous other spurious solutions in nonlinear ICA (see, e.g., the Darmois construction). Thus, the identifiability with undercompleteness and partial sparsity is still an open question, which is one of the motivations of our work.”
> > > > > > > >
> > > > > > > > Furthermore, we have added the following sentence in the **abstract (after L9)**:
> > > > > > > >
> > > > > > > > > ”Structural Sparisty has been introduced before to avoid the spurious solution of “rotated-Gaussian MPA” in the undercomplete case, but the identifiability has not been provided.”
> > > > > > > >
> > > > > > > > We hope the updated text could clarify the potential confusion. Thanks again for highlighting the necessity of that. If you have other related concerns, please kindly let us know.
> > > > > > > >
> > > > > > > >
> > > > > > > > **Q13:** The exact changes on Thms 4.1 and 4.3.
> > > > > > > >
> > > > > > > > **A13:** We sincerely appreciate your valuable suggestions on improving the presentation and your agreement on the implications of these theorems. Regarding the presentation, we have replaced the related sentences with *“$\mathbf{s}_D$ is block-wise identifiable”*. We would like to note that the usage of the term “block-wise identifiable” originates from previous works on identifiably disentangling the changing style from images across domains (e.g., Thm. 4.2 in [2], Thm. 4.2 in [3]), which is similar to our definition of $\mathbf{s}_D$ since its distribution does change across different values of $\mathbf{u}$. We have also highlighted these works for specifying the reference of the definition.
> > > > > > > >
> > > > > > > >
> > > > > > > > **Q14:** Include a short explanation of the universal approximation and GIN in the paper.
> > > > > > > >
> > > > > > > > **A14:** Thanks so much for the suggestion. It has been added, specifically after L345, as follows:
> > > > > > > >
> > > > > > > > > ”It is worth noting that,  according to [4], coupling-based flows (e.g., GIN) are universal diffeomorphism approximators. The volume-preserving nature of GIN does not hinder it from validating our theorems, as rescaling is one of the allowed indeterminacies after identification.”

---

> > > > > > > > > ### Author Response · Authors · 2023-08-19
> > > > > > > > > **Thanks so much for your further suggestions (2/2)**
> > > > > > > > >
> > > > > > > > > **Q15:** Comments on the breath of real-world experiments.
> > > > > > > > >
> > > > > > > > > **A15:** We fully agree with you that there are *millions of different possible inductive biases* for the true hidden generating process of real-world datasets. We also agree that, besides the visual disentanglement task, there are various other exciting applications that could potentially benefit from our theory and worth exploring (e.g., the brain's signals as you mentioned). The lack of more applications is indeed a limitation of our work (as highlighted in the conclusion, L396-397), and we are very grateful for your kind suggestions on what could be the next steps.
> > > > > > > > >
> > > > > > > > > At the same time, since it is impossible to know the structure of the real-world unknown data-generating process, our theory can only be rigorously **validated** by ablation studies (via simulated data), which we have done as part of the experiments. The asymptotic guarantee has also been further validated by the experiments varying the sample size in the attached PDF in the global response. The experiments on the image datasets are indeed not very complicated, and we aim to use these results as potential, explainable,  illustrations of the application scenarios, complementing our validations of the theory.
> > > > > > > > >
> > > > > > > > > Apologies if there is any potential confusion in our previous response regarding that. We do agree with you that there are more exciting real-world applications that could take this paper to the next level, and we are very grateful for your specific suggestions on them.
> > > > > > > > >
> > > > > > > > > ---
> > > > > > > > >
> > > > > > > > > Last but not least, we genuinely appreciate the time and effort you dedicated to reviewing our manuscript. We greatly value this opportunity for clarification and are thankful for the chance you provided to further improve the presentation.
> > > > > > > > >
> > > > > > > > > ---
> > > > > > > > >
> > > > > > > > > [1] Zheng et al. "On the identifiability of nonlinear ICA: sparsity and beyond."
> > > > > > > > >
> > > > > > > > > [2] Kong et al. “Partial identifiability of domain adaptation.”
> > > > > > > > >
> > > > > > > > > [3] Von Kügelgen et al. "Self-supervised learning with data augmentations provably isolates content from style."
> > > > > > > > >
> > > > > > > > > [4] Teshima et al. "Coupling-based invertible neural networks are universal diffeomorphism approximators."

---

> > > > > > > > > > ### Comment · Reviewer_nmMP · 2023-08-20
> > > > > > > > > >
> > > > > > > > > > **With respect to A12:** Yes that's much better and helps to avoid confusion with the Zheng paper -- it also makes your contribution and its novelty much clearer!
> > > > > > > > > >
> > > > > > > > > > **Wrt. A13:** Yes I am aware of the term 'block-wise identifiable' and I do think this is better but I do find it bit weird when there is no mention of $S_I$. Would you consider perhaps adding explanation under the theorem just to explain what is meant by this block-wise identifiability and explain that it does *not* mean that $S_D$ are identified element-wise?
> > > > > > > > > >
> > > > > > > > > > **Wrt. A14+A15:** Thanks.

---

> > > > > > > > > > > ### Author Response · Authors · 2023-08-20
> > > > > > > > > > > **We sincerely appreciate your further feedback**
> > > > > > > > > > >
> > > > > > > > > > > Thank you so much for your reply. Wrt. A13, in addition to the definition of “block-wise identifiability” before Thm. 4.1, in light of your suggestions, we have now also added the following text after L208 (directly following Thm. 4.1):
> > > > > > > > > > >
> > > > > > > > > > > > “We would like to highlight that **the block-wise identifiability of $\mathbf{s}_D$ does not mean that sources in $\mathbf{s}_D$ are element-wise identifiable** (i.e., identifiable up to an element-wise invertible transformation and a permutation). Instead, it only guarantees that sources in $\mathbf{s}_D$ will not be mixed with sources outside of $\mathbf{s}_D$ (e.g., $\mathbf{s}_I$ in Eq. 3), which might be helpful in scenarios such as disentangling changing styles across images from different domains for the purpose of finding the block of style variables, but not necessarily each individual style variable.”
> > > > > > > > > > >
> > > > > > > > > > > We hope that, by incorporating this, readers can get a clearer picture of the results. Thanks again for your constructive suggestion! We are looking forward to your kind feedback.

---

> > > > > > > > > > > > ### Comment · Reviewer_nmMP · 2023-08-20
> > > > > > > > > > > >
> > > > > > > > > > > > Thank you -- I appreciate your patience and efforts. I have revised my score.

---

> > > > > > > > > > > > > ### Author Response · Authors · 2023-08-20
> > > > > > > > > > > > >
> > > > > > > > > > > > > We are very grateful for all of your constructive and insightful suggestions! We believe that the manuscript has been improved a lot with your help.

---

### Official Review · Reviewer_gBCZ · 2023-07-10

**Soundness:** 4 excellent
**Presentation:** 4 excellent
**Contribution:** 4 excellent
**Rating:** 8
**Confidence:** 4

**Summary:**

This paper extends a recent result from Zheng et al 2022, which introduces an assumption they call “structural sparsity” to induce identifiability in nonlinear ICA without relying on a common (but arguably unrealistic) assumption that the observed variables are conditionally independent given observed auxiliary information. Whereas Zheng et al 2022 gave identifiability results only in the setting where the structural sparsity assumption holds perfectly and there are an equal number of sources and observed variables, this paper relaxes these assumptions in several interesting ways and gives identifiability or partial identifiability results in these more general settings.

The first theoretical contribution shows identifiability under structural sparsity in the undercomplete setting, where there are more observed variables than sources. This lets them relax the usual assumption that the mixing function must be bijective, and instead only requires that the mixing function be injective.

The second theoretical contribution relaxes the structural sparsity assumption to the setting where you have partial structural sparsity (it holds for a subset of sources) or partial independence of sources and shows partial identifiability under these assumptions. Here the partial dependence of sources does not need to be known.

The third theoretical contribution assumes that the dependence between sources is known, and the fourth theoretical contribution assumes the sources with dependencies are conditionally independent given auxiliary variables (which is distinct from existing work because they don’t assume all sources are influenced by the auxiliary variable, just the dependent sources).

They use an estimation method using a sparsity regularizer (that encourages a sparse estimated mixing function) with a flow-based generative model. They perform experiments on two simple visual datasets (Triangles and EMNIST) and perform ablations where they generate data that satisfy two combinations of assumptions for their theory, compared to a base setting that does not satisfy their assumptions. Following existing work, they use MCC as their metric and their models achieve higher MCC when the assumptions are satisfied.

**Strengths:**

- Overall, this is a very interesting paper and makes novel contributions in what I think is an interesting setting: using sparsity to induce identifiability in nonlinear ICA.
- They clearly motivate why relaxing each assumption makes the assumptions more realistic.
- I agree that the “conditional independence given auxiliary information” assumption that is common in the literature is not a great assumption, and I’m happy to see recent work removing or reducing this assumption.
- They don’t require distributional assumptions.
- It is well-written and well-structured overall. It is very clear what the prior work accomplishes and what the contributions are.

**Weaknesses:**

- There could be more experiments in realistic settings. (However, given the strength of the theoretical contributions in this paper, I think the paper should be accepted as is.)

Minor comments on the writing (did not affect score):
- Line 84-85: You say “part of the sources can be grouped into irreducible independent subgroups…”, but “irreducible subgroup” is a term in algebra with a specific meaning. You could avoid this “collision” by saying “irreducible independent subgroupings” or something similar.
- Line 156: You start a sentence with the word “Differently, …” which sounds strange. You could say “In contrast, …” instead.
- Line 167: “While this removes the restriction of bijectivity between sources and observed variables, it remains uncertain as to whether Structural Sparsity holds in general, particularly for all sources in a universal way.”
    - This sentence is confusing - you are saying it is uncertain whether Structural Sparsity holds in general, but Structural Sparsity is one of your assumptions. Are you saying it is uncertain whether Structural Sparsity is a reasonable assumption, based on whether it is likely to be satisfied on real-world data?
- Line 177: It’s also weird to start this sentence with “Differently”.
- Multiple lines: You start a handful of sentences with “Besides, …” and each time that is not really the word you mean. You should rethink how each of these sentences connects to the previous sentences and find the appropriate word for each case.

**Questions:**

- Are you aware of Lachapelle et al 2022b? See https://arxiv.org/pdf/2207.07732.pdf. Lachapelle et al 2022a uses mechanism sparsity to induce permutation identifiability, but Lachapelle et al 2022b extends this approach to the partial identifiability setting. It would be (1) worth mentioning in the Introduction section that Lachapelle et al 2022a introduced the idea to use sparsity to induce identifiability, which inspired the approach of Zheng et al. 2022 (as stated in the text of Zheng et al 2022, see Section 3.1 of that paper), and (2) to cite Lachapelle et al 2022b as prior work using sparsity for partial identifiability (though in a distinct setting from your results as it relies on conditional independence given observed auxiliary variables).

**Limitations:**

- Their experiments are only on visual disentanglement tasks and there are many other interesting disentanglement or other tasks that would be interesting to see in future work.
- No concerns about negative societal impacts.

---

> ### Author Rebuttal · Authors · 2023-08-10
>
> We are very grateful for your detailed reading and insightful suggestions. Your comments on the quality of our paper and the strength of our contributions mean a lot to us. Please kindly find our point-by-point response below.
>
> **Q1:** There are many other interesting real-world tasks that would be interesting to see in future work.
>
> **A1:** Thanks a lot for your great suggestions. Indeed, there are various tasks that could benefit from our theory, and, as you suggested, more applications would be intriguing. We have shown some positive results on the visual tasks, and perhaps natural language is also a promising field. At the same time, various empirical research have shown that latent variables are likely identifiable in complex scenarios, possibly involving undercompleteness and violations of sparsity and independence (L88-89, L329-334). Complementing previous works, our theory may interpret these empirical results, and our ablation studies and experiments on both the synthetic and real-world datasets provide further validations. Of course, validating our theory in even more tasks is exciting and will be constantly done in future works.
>
> In addition, we have also included new experimental results in the PDF attached to the global response. These results demonstrate that the quality of identification can be enhanced by increasing the sample size, further validating our theorems.
>
> **Q2:** L84-85: “Irreducible subgroup” is a term in algebra with a specific meaning. You could avoid this “collision” by saying “irreducible independent subgroupings” or something similar.
>
> **A2:** Thank you so much for the constructive suggestion. We have replaced “subgroup” with “subgrouping” in the updated manuscript.
>
> **Q3:** L156: “Differently, …” could be replaced with “In contrast, …”
>
> **A3:** Thanks. We have updated it accordingly.
>
> **Q4:** L167: “While this removes the restriction of bijectivity between sources and observed variables, it remains uncertain as to whether Structural Sparsity holds in general, particularly for all sources in a universal way.” This sentence is confusing.
>
> **A4:** Thanks for the insightful question. In this sentence, we are trying to motivate the importance of dealing with potential partial violations of Structural Sparsity among a subset of sources. In light of your suggestion, we have revised it to “... it remains uncertain as to whether Structural Sparsity always holds for all sources in a universal way”. We hope it could help to avoid potential confusion.
>
> **Q5:** L177: It is weird to start this sentence with “Differently”.
>
> **A5:** Yes, we fully agree with you. We have removed this in the updated manuscript.
>
> **Q6:** A handful of sentences are started with “Besides, ..”, which could be replaced with more appropriate words.
>
> **A6:** We are very grateful for the great suggestion. We have carefully modified the related connecting words in the updated manuscript as follows:
>
> - **L37:** Replaced “Besides” with “Moreover”.
>
> - **L64:** Replaced “Besides” with “In addition to”.
>
> - **L100:** Replaced “Besides” with “Furthermore”.
>
> - **L110:** Replaced “Besides” with “Additionally”.
>
> - **L212:** Replaced “Besides” with “Furthermore”.
>
> - **L278:** Replaced “besides” with “in addition to”.
>
> We hope these modifications could improve the transition between related sentences. Thanks again!
>
> **Q7:** Discuss [1] in the introduction and cite [2] as prior work.
>
> **A7:** Thanks so much for sharing these excellent works. In the updated manuscript, we have highlighted in the introduction that [1] introduced the idea of proving identifiability with sparsity, which then inspired [3]. Moreover, we have emphasized in both the introduction and theory that [2] also uses sparsity for partial identifiability.
>
> ---
>
>
> [1] Sébastien, et al. "Disentanglement via mechanism sparsity regularization: A new principle for nonlinear ICA."
>
> [2] Lachapelle, Sébastien, and Simon Lacoste-Julien. "Partial disentanglement via mechanism sparsity."
>
> [3] Zheng et al. "On the identifiability of nonlinear ICA: sparsity and beyond."

---

> > ### Comment · Reviewer_gBCZ · 2023-08-19
> > **Reviewer response to rebuttal**
> >
> > Thanks to the authors for the thorough response to my comments and for including the additional plot in the one-page pdf. You've addressed all the questions and suggestions for improvements in my review. I will keep my score of 8.

---

> > > ### Author Response · Authors · 2023-08-19
> > >
> > > Thanks so much for your effort! We are very grateful for your encouragement.

---

### Official Review · Reviewer_jekt · 2023-07-27

**Soundness:** 4 excellent
**Presentation:** 3 good
**Contribution:** 3 good
**Rating:** 7
**Confidence:** 3

**Summary:**


The paper introduces more flexible ways to perform nonlinear independent component analysis (nonlinear ICA).  Nonlinear ICA involves identifying the sources s from the observed x when both s and x are related by x = f(s) and f is a nonlinear function.  Previous work has developed a method for this problem under a strict structural sparsity assumption that the s's and x's are one-to-one and onto, and all the s's are independent of each other.  Current work provides theorems that relax the assumption in various ways including: (1) undercompletness--there can be more observed variables x than sources s; (2) partial sparsity--only a subset of all the s's may map to x's; (3) source dependence--all sources s do not have to be statistically dependent on each other; and (4) flexible grouping structures--the possibility that some of the sources can be partitioned into independent subgroups of sources.  There are experiments on synthetic and real world datasets that show the effectivness of their approach.




**Strengths:**


This paper is original because it introduces novel approaches, as far as I know, that extend the situations where nonlinear ICA can be applied.

The paper exhibits good quality in various ways.  First, there are various theoretical results included in the paper that extend the cases where nonlinear ICA can be applied.  Second, there are also results in several experimental settings that back up the theory.

The paper is mostly clear in its explanations.

In terms of significance, extending the situations where nonlinear ICA can be applied is an accomplishment.


**Weaknesses:**

While in theory extending the cases where nonlinear ICA can be applied is a strength, because there wasn't any empirical qualitative evaluation of how this approach compares to other approaches in disentangling the sources, it is not clear how significant this work is.  It does not have to be a comparison of how well it disentangles sources; it could be comparing them on some other application, such as how well they extract features that are useful for classification, for example.  It does not even have to be comparing this paper's approach to previous approaches; it could be comparing the different extensions of nonlinear ICA presented in this paper.

Also, as pointed out by the authors, another limitation of this work is that the experimental results were only on visual datasets but not on other modalities.


**Questions:**

While the undercompleteness result appears to me to be unique to this paper, (Zheng et al 2022) also has an undercompletness result.  This paper is written so that it sounds like (Zheng et al 2022) has no undercompletness result.  It would be nice if this situation could be explained or clarified.

It was a bit confusing that on line 113, A is defined as a set of natural number tuples but on line 103 A is defined as a subset of natural numbers.

I think on line 114 that A_{:,j} := { i | (i,j) \in S } should really be A_{:,j} := { i | (i,j) \in A }.


**Limitations:**

Yes.

---

> ### Author Rebuttal · Authors · 2023-08-10
>
> We sincerely thank the reviewer’s time on carefully reading our manuscript and providing insightful suggestions. These have undoubtedly further improved the manuscript. Please kindly find our detailed, point-by-point response below.
>
> **Q1:** Suggestions on expanding empirical comparisons.
>
> **A1:** We are very grateful for these constructive suggestions. It is worth noting that, instead of proposing a better model to disentangle the sources, we focus on providing a theoretical guarantee for uncovering generating processes under certain conditions. Our result could be one of the lacking interpretations of many previous empirical studies showing that latent variable models are likely identifiable in complex scenarios, as mentioned in L88-89 and L329-344. The various extensions proposed for nonlinear ICA focus on the assumptions regarding the ground-truth data-generating process, rather than the estimation methods. Therefore, they can only be rigorously validated through ablation studies conducted on different data-generating processes. Complementing various previous empirical studies, we believe that our ablation studies and experiments on both the synthetic and real-world datasets provide further validations.
>
> In addition, we have also included new experimental results in the PDF attached to the global response. These results demonstrate that the quality of identification can be enhanced by increasing the sample size, further validating our theorems.
>
> **Q2:** Clarify the theorem proposed in [1] about the undercomplete case.
>
> **A2:** We appreciate the great suggestion. As mentioned in L159-160, [1] only removes the rotational indeterminacy while our theorem removes all major indeterminacies and only preserves the component-wise transformation and permutation. In other words, [1] only gets rid of specific spurious solutions due to the rotational indeterminacy (e.g., the ‘rotated-Gaussian’ MPA) while we prove the full identifiability of the undercomplete case. We have added additional detailed discussion earlier in the introduction to avoid potential confusion.
>
> **Q3:** Notations:
>
> - **(1):** It was a bit confusing that on L113, $\mathcal{A}$ is defined as a set of natural number tuples but on L103 $\mathcal{A}$ is defined as a subset of natural numbers;
>
> - **(2):** there is a typo on L114, i.e., $\mathcal{S}$ should be $\mathcal{A}$.
>
> **A3:** Thank you so much for reminding us, and sorry for any potential confusion. We have modified L113-114 as: “For any set of indices $\mathcal{B} \subset \\{1, \ldots, m\\} \times \\{1, \ldots, n\\}$, analogously, we have $\mathcal{B}\_{i,:}\coloneqq\\{j \mid(i, j) \in \mathcal{B}\\}$ and $\mathcal{B}_{:,j}\coloneqq\\{i \mid(i, j) \in \mathcal{B}\\}$." This modification also corrects the typo. Thanks again!
>
>
> ---
>
>
>
> [1] Zheng et al. "On the identifiability of nonlinear ICA: sparsity and beyond."

---

> > ### Comment · Reviewer_jekt · 2023-08-16
> >
> >
> > I have read your response.  Thank you for preparing it.  It has clarified the meaning of certain passages in the paper.  Maybe it would be an even better paper if the theory could tell you whether to use a certain identifiability approach given a particular set of empirical data, rather than having to perform ablation studies, but the current paper as it is does break new ground.

---

> > > ### Author Response · Authors · 2023-08-16
> > >
> > > Thanks a lot for your encouragement. We are very grateful for all of your insightful suggestions.

---

### Author Rebuttal · Authors · 2023-08-10

We extend our sincere thanks to each of the reviewers for their thoughtful insights and the time devoted to reviewing our manuscript. We are encouraged that all of the reviewers have found our paper of good quality in various ways. With appreciation, we have provided detailed, point-by-point responses to each reviewer's comments in the individual replies. In this global response, we take the opportunity to present additional experimental results, which have been summarized in the attached PDF. Specifically, the quality of the identification improves w.r.t. the increasing of the sample size.

---

### Decision · Program_Chairs · 2023-09-21

**Decision:**

Accept (oral)

**Comment:**

This article extends the previous work on nonlinear Independent Component Analysis (ICA) identifiability, which focused on structural sparsity assumptions. This paper builds on that by handling the undercomplete scenario and relaxing sparsity and source independence assumptions. The article includes theoretical advancements and practical examples to illustrate their identifiability theorems.

Overall, the paper is intriguing, offering fresh insights into inducing identifiability in nonlinear ICA through sparsity. Theoretical advancements are well-presented, expanding the applicability of nonlinear ICA. The relaxation of assumptions is motivated logically, supported by experimental results. The paper is well-written and structured, effectively delineating prior work, contributions, and theoretical foundations.

The level of novelty and contribution to the community is deemed substantial.